# Effective Scalar Potential in Asymptotically Safe Quantum Gravity

**Christof Wetterich**

Institut für Theoretische Physik, Universität Heidelberg, Philosophenweg 16, D-69120 Heidelberg, Germany;
C.Wetterich@thphys.uni-heidelberg.de

**Abstract:** We compute the effective potential for scalar fields in asymptotically safe quantum gravity. A scaling potential and other scaling functions generalize the fixed point values of renormalizable couplings. The scaling potential takes a non-polynomial form, approaching typically a constant for large values of scalar fields. Spontaneous symmetry breaking may be induced by non-vanishing gauge couplings. We strengthen the arguments for a prediction of the ratio between the masses of the top quark and the Higgs boson. Higgs inflation in the standard model is unlikely to be compatible with asymptotic safety. Scaling solutions with vanishing relevant parameters can be sufficient for a realistic description of particle physics and cosmology, leading to an asymptotically vanishing "cosmological constant" or dynamical dark energy.

**Keywords:** quantum gravity; asymptotic safety; effective scalar potential; inflation; Higgs inflation; Higgs boson mass

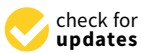



## 1. Introduction

The effective potential for scalar fields is the key ingredient for spontaneous symmetry breaking by the Higgs mechanism in the standard model or for grand unified theories. It determines the properties of inflationary cosmology as well as dynamical dark energy. We compute here the influence of quantum gravity on the shape of the potential, motivated by the following issues:

*Clash between mass of the Higgs boson and Higgs inflation.* Within asymptotically safe quantum gravity [1,2] the value of the Higgs boson mass has been predicted to be 126 GeV with a few GeV uncertainty [3]. This prediction relies on two assumptions. The first is a positive and substantial gravity-induced anomalous dimension $A$ that renders the quartic scalar coupling $\lambda_H$ an irrelevant parameter. Then $\lambda_H$ is predicted to have a very small value at and near the ultraviolet (UV) fixed point. The second assumes that once the metric fluctuations decouple at momenta sufficiently below the effective Planck mass the running of $\lambda_H$ is given by the standard model with, at most, small modifications. First indications for a positive $A$ have been seen in early investigations how matter couples to gravity in asymptotic freedom [4]. Physical gauge fixing, or a gauge invariant flow equation for a single metric [5], show a graviton domination of $A$ [6] and establish a positive $A$ [6–8], substantiating the prediction of the mass of the Higgs boson.

The prediction of the Higgs boson mass concerns the properties of the effective scalar potential at field values much smaller than the effective Planck mass $M$. In contrast, models of Higgs inflation [9,10] explore instead properties of the potential at field values somewhat below $M$, or even exceeding $M$. Usually only particle fluctuations are included in the computation of the effective potential, while the contributions of metric fluctuations are neglected. It has been argued [11] that asymptotically safe quantum gravity may substantially influence the behavior of the Higgs potential at large fields. In this paper we aim for a more global view of the effective scalar potential, ranging from small field values to large ones exceeding $M$.

A global view in field space is also necessary because of a potential clash between Higgs inflation and the prediction for the mass of the Higgs boson. The prediction for the mass of the Higgs boson is based on the quantum gravity prediction of a very small quartic scalar coupling for a re-normalization scale near the Planck mass. It has been obtained under the assumption of a minimal coupling of the Higgs boson to gravity.

In the presence of a non-minimal coupling of the type $\xi h^\dagger h R$ between the Higgs doublet $h$ and the curvature scalar $R$, the gravitational fluctuations contribute to the flow of the quartic coupling:

$$k\partial_k \lambda \approx A\lambda + B_\xi \frac{k^4 \xi^2}{M^4}, \tag{1}$$

with $k$ the re-normalization scale. The fixed point behavior of quantum gravity that is responsible for the prediction of the mass of the Higgs boson, concerns a range where $k^2 \gtrsim M^2$, such that the effects of the metric fluctuations are relevant. These metric fluctuations are the key for the prediction, since they are responsible for the substantial positive anomalous dimension $A$. This anomalous dimension is universal in the sense that it does not depend on the representation of the scalar field or on interactions beyond its gravitational interactions.

A modification of the gravitational contribution by the presence of the non-minimal coupling $\xi$ could lead to an important quantitative change for the prediction of the Higgs-boson mass. Indeed, the fixed point for flow (1) occurs for:

$$\lambda_* = -\frac{B_\xi k^4}{A M^4} \xi^2, \tag{2}$$

and ceases to be very small for large $\xi$. In our approximation we find:

$$B_\xi = \frac{5v}{12\pi^2 (1-v)^3}, \tag{3}$$

with $v = 2U/(M^2 k^2)$ involving the value of the scalar potential $U$ (or cosmological constant). In the relevant range of $k$ one has for the standard model $M^2/k^2 \approx 0.01$, $v \approx -10$, and $A \approx 0.05$. Insertion into Equations (2) and (3) yields $\lambda_* \approx 100\xi^2$. Due to the effects of other couplings, the flow of the quartic coupling is slightly more complicated than Equations (1) and (3). The detailed form of $B_\xi$ also depends on details for the setup of the flow equations. Nevertheless, it is clear that a large value of $\xi$ is not compatible with a fixed point at a small value of $\lambda$. We will find that for $\xi_0$ larger than about 0.01, asymptotic safety still predicts the mass of the Higgs boson, but this prediction ends outside the observed range.

For Higgs inflation, a rather large non-minimal coupling is usually assumed, say $\xi_\infty \gtrsim 10$. This is several orders of magnitude larger than the value $\xi_0$ allowed by the observed mass of the Higgs boson. The non-minimal coupling $\xi_\infty$ for Higgs inflation concerns large values of the Higgs field, while the vacuum mass of the Higgs boson concerns values of $h^\dagger h$ many orders of magnitude smaller than $M^2$. Since $\xi$ may be a function of $h^\dagger h$, an overall view of a whole coupling function is needed, similar to the need of an overall view of the effective scalar potential. We will find that $\xi_\infty$ is typically more than a factor 10 smaller than $\xi_0$, exacerbating the clash.

For asymptotically safe quantum gravity, the ultraviolet fixed point needs scaling solutions both for the effective potential and the coupling of scalar fields to the curvature scalar. It is on these scaling solutions that we concentrate in the present paper. For the scaling solutions found, $\xi$ turns out to be rather small over the whole range of the Higgs doublet field. These solutions are compatible with a successful prediction of the mass of the Higgs boson. On the other hand, for the pure standard model coupled to gravity Higgs inflation with a large non-minimal coupling, $\xi$ is not compatible with asymptotic safety. It remains to be seen if Higgs inflation with small $\xi$ is viable.

*Link between inflation and dynamical dark energy.* For cosmology, a global view on the effective potential for a scalar singlet field $\chi$ is also needed for models of cosmon inflation [12–14] and dynamical dark energy or quintessence [15]. In these models the scalar singlet plays the role of the inflation or the cosmon as a quintessence field, or both simultaneously. It has been found [11] that the effective potential for a singlet $\chi$ shows a rather rich structure, due to a crossover between different fixed points. While "gravity scale symmetry" associated to the UV fixed point is responsible for the almost scale invariant primordial fluctuation spectrum, an infrared (IR) fixed point [6,16–18] is reached for large values of $\chi$. The "cosmic scale symmetry" associated to the IR fixed point is spontaneously broken by any nonzero $\chi$. The associated pseudo-Goldstone boson (cosmon) has a very small mass for large $\chi$. It is responsible for dynamical dark energy [15].

The present paper addresses this issue as well. The parts concentrating on the fluctuations of a scalar singlet and the metric, with all other particles treated as massless (sects 3, 6) can be seen as a computation of the effective potential $U(\chi)$ for the scalar singlet $\chi$. We reproduce features found earlier in the context of dilaton quantum gravity [11,17,18]. Our rather simple approach helps to understand these features. It also puts the candidate scaling solutions found earlier in a wider context of possible scaling solutions.

Concerning the properties of the effective potential for non-singlet scalar fields as the Higgs doublet, we do not distinguish here between quantum Einstein gravity [2], where the Planck mass $\bar{M}$ corresponds to a relevant parameter and constitutes an intrinsic mass scale, breaking quantum scale symmetry explicitly, and dilaton quantum gravity [17,18], where the effective Planck mass depends monotonically on $\chi$, such that for a suitable normalization of $\chi$ one has $M = \chi$ for large $\chi$. In the latter case, quantum scale symmetry can be preserved, being only spontaneously broken by $\chi \neq 0$. Whenever we use $M^2$, the reader may substitute it by a function $F(\chi)$.

*Regimes of the re-normalization flow and predictivity of quantum gravity.* The re-normalization flow describes the change of the effective scalar potential for increasing length scales, as more and more fluctuation effects are included. It is characterized by different regimes. The "quantum gravity regime" is associated to re-normalization scales exceeding $M$, corresponding to length scales smaller than the Planck length. In this regime the fluctuations of the metric play an important role. The quantum gravity regime is associated to the UV fixed point defining quantum gravity as a non-perturbatively re-normalizable quantum field theory (asymptotic safety). At the UV fixed point one has a scaling behavior;

$$M^2(k) = 2w_* k^2 \,, \tag{4}$$

with $w_*$ the fixed point value of the dimensionless coupling $w(k) = M^2(k)/(2k^2)$ (in case of additional scalar fields $\chi$ we may replace $w_*$ by a scaling function depending also on $\chi^2/k^2$). In the quantum gravity regime, the effective scalar potential takes a scaling form where the dimensionless potential $u = U/k^4$ only depends on dimensionless field ratios as $\tilde{\rho} = \rho/k^2$, with $\rho$ a typical quadratic invariant formed from scalar fields (for the Higgs doublet one has $\rho = h^\dagger h$, with $h$ the re-normalized scalar doublet, while for a scalar singlet $\chi$ we use $\rho = \chi^2/2$). The main emphasis of the present paper is the computation of the "scaling potential" $u_*(\tilde{\rho})$.

A second "particle regime" concerns the flow for $k \ll M$. In this regime the metric fluctuations decouple effectively, up to the flow of an overall constant in $U$, e.g. the cosmological constant. The flow equation for the field dependence of $U$ is governed by the effective particle theory for momenta below the Planck mass. This flow can be computed in perturbation theory. It obviously depends on the precise particle content of the the effective low energy theory. The flow in the particle regime may again be characterized by an approximate fixed point, and the associated "particle scale symmetry". For a standard model as effective low energy theory, this fixed point is associated to the (almost) second order character of the vacuum electroweak phase transition. A similar fixed point may exist for grand unified theories (GUT). The present paper will not deal with the flow in the

particle regime which has to be added for $k \ll M$. The transition from the quantum gravity regime to the particle regime is modeled by a simple behavior for the $k$-dependent Planck mass,

$$M^2(k) = M^2 + 2w_*k^2 \,, \tag{5}$$

where $M^2$ is associated to the observed Planck mass, either a constant or given by a scalar field, $M^2 = \chi^2$.

For extremely large field values, $\rho/k^2$, one finally reaches the infrared regime. There graviton fluctuations may become again important due to a potential instability in the graviton propagator. A "graviton barrier" [6] prevents the potential to rise for large field values stronger than the field dependent squared Planck mass. We will not be concerned very much with the infrared regime in the present investigation.

The present paper concentrates on the quantum gravity regime. We are mainly interested in general characteristics of the scaling form of the effective potential, as the location of the minimum $\tilde{\rho}_0$ at $\tilde{\rho}_0 = 0$ or at $\tilde{\rho}_0 \neq 0$, and the general behavior as $\tilde{\rho}$ vanishes or increases beyond $\tilde{\rho}_0$. We put emphasis on the dependence on gauge couplings and Yukawa couplings that we treat here as constants. This covers two scenarios. The first is that the fixed point values of these couplings may be at nonzero values. In this case the gauge couplings and Yukawa couplings typically correspond to irrelevant parameters that can be predicted by quantum gravity [19]. Or, the second, the UV fixed point corresponds to zero values of these couplings, which are relevant parameters. The flow away from the fixed point is, however, very slow in the vicinity of the fixed point. For their observed small values the gauge and Yukawa couplings only increase rather slowly with decreasing $k$. To a good approximation they can be treated as constants in this regime. Our investigation of scaling solutions for constant gauge and Yukawa couplings describes then approximate scaling solutions in the vicinity of the UV fixed point.

*Scaling solutions*. The main emphasis of the present paper concerns scaling solutions, in particular the scaling potential. We will briefly discuss some aspects of the flow away from the scaling potential. For models with fundamental scale invariance, the scaling solutions are all what is needed. For a computation of the scaling potential in asymptotically safe quantum gravity, we first treat $w_*$ as an unknown parameter. Our computation needs therefore to be supplemented by a computation of $w_*$. The latter depends on the precise particle content of the model. In Section 6 we extend this to a fixed scaling function $w_*(\tilde{\rho}) = w_0 + \xi \tilde{\rho}/2$, with free parameters $w_0$ and $\xi$. Finally, in Sections 7 and 8 we extend the truncation to simultaneous solutions of flow equations for both $u(\tilde{\rho})$ and $w(\tilde{\rho})$. This establishes a system of combined scaling functions $u_*(\tilde{\rho})$ and $w_*(\tilde{\rho})$. This stepwise procedure helps to organize the rather complex issue in a way that important features can be treated separately.

For vanishing gauge and Yukawa couplings there exists a "constant scaling solution" for which $u(\tilde{\rho})$ and $w(\tilde{\rho})$ are independent of $\tilde{\rho}$. This is the extended Reuter fixed point. We are interested in the possible existence of other fixed points, for which the scaling functions $u(\tilde{\rho})$ and $w(\tilde{\rho})$ are independent of $k$, but show a non-trivial dependence, a $\tilde{\rho}$. This is typically induced by non-zero gauge and Yukawa couplings, but it could also occur for vanishing gauge and Yukawa couplings. We consider first the regime where non-minimal couplings of the scalar field to gravity $\sim \xi \rho R$ can be neglected (here $R$ is the curvature scalar and $\xi$ the non-minimal coupling). In this case our main findings for the global scaling form for a possible non-constant dimensionless effective scalar potential $u(\tilde{\rho})$ are the following. For zero gauge and Yukawa couplings, the potential interpolates between two constants,

$$u(\tilde{\rho} \to 0) = u_0 \,, \quad u(\tilde{\rho} \to \infty) = u_\infty \,, \quad u_\infty > u_0 \,. \tag{6}$$

The minimum is situated at the origin $\tilde{\rho} = 0$.

This behavior occurs also for nonzero Yukawa couplings $y$ and zero gauge couplings. In contrast, nonzero gauge couplings $g$ can induce a potential minimum at $\tilde{\rho}_0 \neq 0$. The asymptotic behavior (6) remains valid. While for vanishing gauge and Yukawa cou-

plings particular "constant scaling solutions" exist, with $\tilde{\rho}$-independent $u_*(\tilde{\rho}) = u_0$ or $u_*(\tilde{\rho}) = u_\infty$, this possibility is no longer given in our truncation for nonzero gauge or Yukawa couplings.

For a non-vanishing non-minimal gravitational coupling, $\xi \neq 0$, the asymptotic behavior of $u$ for $\tilde{\rho} \to \infty$ can change. We still find scaling solutions with a constant $u_\infty$. Alternatively, for asymptotically large $\tilde{\rho}$, the "IR-behavior" $u(\tilde{\rho} \to \infty) = \xi \tilde{\rho}/2$ is reached. The intermediate behavior can be rather complex. In particular, we find for $g = y = 0$ that the scaling potential can develop a minimum at $\tilde{\rho}_0 \neq 0$. For $\xi \neq 0$, no constant scaling solution exists.

For scaling solutions of the combined flow equations for $u(\tilde{\rho})$ and $w(\tilde{\rho})$, we focus on a family of candidate scaling solutions that depend on a continuous parameter $\xi_\infty$. For these solutions one has the asymptotic behavior:

$$w(\tilde{\rho} \to \infty) = \frac{1}{2}\xi_\infty \tilde{\rho}, \qquad\qquad u(\tilde{\rho} \to \infty) = u_\infty. \qquad (7)$$

For the particle content of the standard model, the minimum of the scaling potential occurs for $\tilde{\rho} = 0$. As $\xi_\infty \to 0$, the constant scaling solution is approached smoothly. For $\xi_\infty \gtrsim 10^{-3}$ the existence of the solution becomes questionable since the $\tilde{\rho}$-dependence of $u/w$ becomes strong, with a rather irregular behavior of $\partial u/\partial\tilde{\rho}$ and $\partial v/\partial\tilde{\rho}$ in an intermediate region. For the solutions with $\xi_\infty < 10^{-3}$, more elaborate numerical solutions should establish if these solutions exist for all $\xi_\infty$ in this range or not.

*Breakdown of polynomial approximation.* For perturbative computations in particle physics, the effective scalar potential is usually well approximated by a polynomial. Quantum gravity effects modify this property profoundly. As a general feature, the scaling solutions for the effective scalar potential cannot be approximated by polynomials. There is a basic reason why quantum gravity is rather different from perturbatively re-normalizable quantum field theories as gauge theories or Yukawa-type theories. Small gauge and Yukawa couplings are in the vicinity of a Gaussian fixed point for a non-interacting theory. In this case, the re-normalizability of couplings is directly related to their canonical dimension. Different powers of scalar fields in a polynomial expansion have a different canonical dimension. Above critical power four, the higher powers in an expansion of $U$ are typically suppressed. This reasoning is no longer valid for asymptotic safety for which interactions play a role at the fixed point.

For example, a crossover between two constants as for Equation (6) can well happen with a positive mass term $\tilde{m}_0^2$ at the origin, but a negative quartic coupling $\lambda_0$. The negative quartic coupling does not indicate any instability of the potential, but merely a decrease of $\tilde{m}^2(\tilde{\rho})$ as $\tilde{\rho}$ increases. This is rather typical for a crossover between constants for $\tilde{\rho} \to 0$ and $\tilde{\rho} \to \infty$. The perturbative experience that a negative quartic coupling $\lambda$ indicates an instability or the presence of another potential minimum for larger field values is misleading in the context of quantum gravity.

*Spontaneous symmetry breaking for scaling potentials.* We observe that the interplay of gravitational fluctuations with fluctuations of gauge fields often leads to a scaling potential with a minimum at $\tilde{\rho}$ different from zero. This points to spontaneous symmetry breaking around the Planck scale by a type of gravitational Coleman–Weinberg mechanism. The symmetry breaking is induced by fluctuations.

The scaling potential $u(\tilde{\rho})$ is a function of the scale invariant variable $\tilde{\rho} = \rho/k^2$. In particular, a minimum at $\tilde{\rho}_0$ corresponds to a "sliding minimum" of the effective potential $U = uk^4$, at $\rho_0 = \tilde{\rho}_0 k^2$. The question arises as to which range of $\tilde{\rho}$ is relevant for observations. For a rough estimate we make the simple ansatz that the scaling solution is valid for $k > k_t$, with transition scale $k_t$ determined by $2w(\tilde{\rho}) k_t^2 = M^2 + \xi\rho$ and $M$ the observed Planck

mass. We further assume that for $k < k_t$, the metric fluctuations decouple and the effective low energy theory becomes valid. This approximation determines at $k_t$ the field $\rho = \tilde{\rho} \, k_t^2$ as:

$$\frac{\rho}{M^2} = \frac{\tilde{\rho}}{2w_0} \, . \tag{8}$$

A minimum of the scaling potential at $\tilde{\rho}_0$ corresponds at $k_t$ to $\rho_0 = M^2 \tilde{\rho}_0/(2w_0)$. Typically, $\rho_0$ continues to change in the low energy effective theory. Nevertheless, for $2w_0 \approx 0.1$ a characteristic field $\tilde{\rho}$ can be associated with field values $\rho \sim 10\tilde{\rho}M^2$. A typical GUT scale $\rho \sim (10^{16}\,\text{GeV})^2$ corresponds to $\tilde{\rho} \approx 10^{-5}$ or $x = \ln(\tilde{\rho}) \approx -11.5$. We often find the location of a minimum at $x$ around $-2$ which corresponds to $\rho$ around $M^2$.

For GUT models, an important part of the spontaneous symmetry breaking is due to scalar fields in representations that do not allow for Yukawa couplings to the fermions. For $SU(5)$-theories this could be the 24-representation, and for $SO(10)$-theories the 45 or 54 representations. In the presence of quantum gravity and for a non-zero gauge coupling, we find that the candidate scaling solutions have a minimum at non-zero field values, indicating indeed spontaneous breaking of the grand unified gauge group. For the investigated examples, the scale of spontaneous symmetry breaking is typically found close to the Planck mass. A more systematic investigation will be needed in order to see under which circumstances the GUT-scale can be substantially below the Planck mass.

*Overview.* The present paper is organized such that the effects of different couplings are described separately. In Section 2, we present the flow equation for the effective scalar potential, following closely references [6,7,11]. The specific physical gauge fixing, equivalent in our truncation to the gauge invariant flow equation [5], makes the structure very apparent. The general features are similar to earlier investigations [4,20–36]. In Section 3 we concentrate on the scaling solution for "matter freedom", which describes a situation where gauge and Yukawa couplings, as well as the non-minimal coupling $\zeta$, can be neglected. In this limit all particles are free except for their gravitational interactions. We find candidate scaling solutions characterized by a crossover from a fixed point with constant $u = u_0$ for $\tilde{\rho} \to 0$ to another one with constant $u = u_\infty$ to $\tilde{\rho} \to \infty$. Improvement of the numerical treatment would be needed in order to decide definitely if this truncation admits scaling solutions different from the constant scaling solutions. Section 4 addresses the flow in the vicinity of the scaling solution for matter freedom, supplemented in Appendix C by a discussion of the scalar mass term and quartic coupling.

In Section 5, we take a first step beyond matter freedom by discussing non-vanishing gauge couplings, still keeping an approximation with constant $w_*$. Typical scaling potentials show a minimum near $\tilde{\rho} = 1$. A similar discussion in Appendix D for non-vanishing Yukawa couplings shows that in this case, the minimum of the scaling potential occurs for $\tilde{\rho} = 0$. For scalars with both gauge and Yukawa couplings, the competition between the opposite tendencies for gauge and Yukawa couplings will be important. In Section 6 we include a non-minimal coupling $\zeta$ of the scalar field to the curvature tensor. This changes the behavior for $\tilde{\rho} \to \infty$.

In Sections 7 and 8 we extend the truncation by investigating solutions to the combined flow equations for $u(\tilde{\rho})$ and $w(\tilde{\rho})$. For the derivation of the flow equations we will follow reference [8]. In Section 7 we discuss general features and turn to the standard model coupled to quantum gravity in Section 8. There we discuss in particular the issue of Higgs inflation and the prediction for the Higgs boson or the top quark mass. Section 9 contains our conclusions.

## 2. Flow Equation for Effective Potential

The present work is based on the flow equation for the effective average action [37–41]. Instead of a flow with a changing UV-cutoff in earlier formulations [42,43], the flow of the effective average action considers the variation of an infrared cutoff. The effective average action corresponds to the quantum effective action (generating functional of one-particle-

irreducible correlation functions) in the presence of an infrared cutoff $k$ which suppresses the fluctuations with momenta $q^2 < k^2$. The quantum effective action is obtained in the limit $k \to 0$. The flow equation involves only a momentum range $q^2 \approx k^2$. It is ultraviolet finite such that no ultraviolet cutoff needs to be introduced. The microscopic physics is specified by the "initial values" of the flow for very large $k$. The simple one-loop form of the exact flow equation permits for successful non-pertubative approximations. Reviews on functional re-normalization are in [44–51], and for its applications to quantum gravity see [52–59].

Let us consider scalar fields $\phi_a$, belonging to various representations of some symmetry group, and investigate the flow of the effective scalar potential $U(\phi_a)$. Our truncation for the effective average action involves up to two derivatives:

$$\mathcal{L} = \sqrt{g} \left\{ -\frac{F(\phi_a)}{2} R + U(\phi_a) + \sum_a \frac{Z_a}{2} D^\mu \phi_a D_\mu \phi_a + \dots \right\}, \tag{9}$$

where the dots denote parts involving gauge fields and fermions. We are interested in the "flow" or dependence on $k$ of the functions $U(\phi)$ and $F(\phi)$, and work in an approximation for which the flow of the wave functions $Z_a(\phi)$ is neglected, setting $Z_a(\phi) = 1$. This corresponds to an incomplete first order in a derivative expansion.

The flow equation for $U$ has contributions from fluctuations of various fields,

$$\partial_t U = k \partial_k U = \overline{\zeta} = \tilde{\pi}_{\text{grav}} + \tilde{\pi}_{\text{s}} + \tilde{\pi}_{\text{gauge}} + \tilde{\pi}_{\text{f}}, \tag{10}$$

namely metric fluctuations $(\tilde{\pi}_{\text{grav}})$, scalar fluctuations $(\tilde{\pi}_{\text{s}})$, gauge boson fluctuations $(\tilde{\pi}_{\text{gauge}})$, and fermion fluctuations $(\tilde{\pi}_{\text{f}})$. We will specify the various contributions step by step. The concrete form of Equation (10) is based on [6–8,11], with explicit form given in [7].

For a physical gauge fixing or the gauge invariant flow equation, the gravitational contribution takes a rather simple form [6,7]:

$$\tilde{\pi}_{\text{grav}} = \frac{k^4}{24\pi^2} \left( 1 - \frac{\eta_g}{8} \right) \left( \frac{5}{1-v} + \frac{1}{1-v/4} \right) - \frac{k^4}{8\pi^2}. \tag{11}$$

The gravitational contribution depends on $U$ and the coefficient $F$ in front of the curvature scalar via the combination:

$$v = \frac{2U}{Fk^2} = \frac{u}{w}, \tag{12}$$

with dimensionless functions $u$ and $w$ depending on the scalar fields $\phi_a$,

$$u = \frac{U}{k^4}, \quad w = \frac{F}{2k^2}. \tag{13}$$

Equation (11) is a central equation for this work. It describes the universal contribution of gravitational fluctuations to the flow of the scalar potential. It is the same for all scalar fields, involving only the combined potential $u$ for all scalar fields though Equation (12). The second ingredient is the effective field-dependent strength of the gravitational interaction encoded in $w$. The various factors in Equation (11) are rather easy to understand. The overall scale is set by $k^4$, as appropriate for the dimension of $U$, and $1/(32\pi^2)$ is a typical loop factor from the momentum integration. The first term in Equation (11) arises from the fluctuations of the graviton (five components of the traceless transversal tensor), the second from the physical scalar fluctuations contained in the metric. For a computation of the flow of $U$, the effect of these fluctuations is evaluated in flat space. The minus sign in the denominator $(1-v)^{-1}$ reflects the negative mass-like term in the flat space graviton propagator for a positive $U$. Indeed, the graviton propagator is proportional to $(Fq^2 + 2U)$,

and the squared momentum $q^2$ is replaced effectively by $k^2$. This holds similarly for the second term which is due to the fluctuations of the physical scalar mode in the metric. The third "measure contribution" accounts for the gauge modes in the metric fluctuations and ghosts. It is independent of the scalar fields. For the specific form of the "threshold functions" appearing in Equation (11) we have employed a Litim cutoff function [60].

We neglect the mixing between the physical scalar mode in the metric and the scalars $\phi_a$, which only plays a very small role for our investigation. Finally, the gravitational anomalous dimension:

$$\eta_g = -\partial_t \ln(w) \tag{14}$$

reflects the choice of the IR-cutoff function proportional to $F$. At the UV-fixed point, $\eta_g$ vanishes if $w$ is field independent.

The contribution from scalar fluctuations $\tilde{\pi}_s$ reads: [61]

$$\tilde{\pi}_s = \frac{k^4}{32\pi^2} \sum_A \left(1 - \frac{\eta_A}{6}\right)\left(1 + \tilde{m}_A^2\right)^{-1}, \tag{15}$$

where the sum runs over $N_S$ scalar fields. The index $A$ labels the eigenvalues $M_A^2$ of the (re-normalized) scalar mass matrix:

$$M_{ab}^2 = (Z_a Z_b)^{-\frac{1}{2}} \frac{\partial^2 U}{\partial\phi_a \partial\phi_b}, \quad \tilde{m}_A^2 = \frac{M_A^2}{k^2}. \tag{16}$$

Here $Z_a$ are the scalar wave functions, given by the coefficient of the kinetic term for $\phi_a$. The factor $(1 + \tilde{m}_A^2)^{-1}$ is a threshold function that accounts for the suppression of contributions of particles with mass terms larger than $k^2$, ensuring decoupling automatically. The anomalous dimension $\eta_A = -\partial_t \ln(Z_A)$ reflects the choice of an IR-cutoff function for the scalar proportional to $Z_A$, with $Z_A$ connected suitably to $Z_a$ (in the case of scalars in a single representation, one uses the same $Z$ for the cutoff function and the definition of all re-normalized fields). Through the threshold function in the scalar contribution, the flow equation for $U$ involves field-derivatives of $U$.

The contributions from gauge bosons $\tilde{\pi}_{\text{gauge}}$ and the contributions from fermions $\tilde{\pi}_f$ do not depend on the scalar fields in the limit of zero gauge couplings or Yukawa couplings, respectively. They will be specified later. The flow of mass terms and quartic couplings obtains by differentiating Equation (10) twice or four times with respect to $\phi$.

The flow Equation (10) holds for fixed values of $\phi_a$. For the investigation of the scaling solution relevant for a fixed point one transforms this to a flow equation for $u = U/k^4$ at fixed dimensionless renormalized fields,

$$\tilde{\phi}_a = \frac{Z_a^{\frac{1}{2}} \phi_a}{k}, \tag{17}$$

where

$$\partial_t u = -4u + \sum_a \left(1 + \frac{\eta_a}{2}\right)\tilde{\phi}_a \frac{\partial u}{\partial\tilde{\phi}_a} + \frac{\overline{\zeta}}{k^4}. \tag{18}$$

Derivatives of $u$ with respect to $\tilde{\phi}_a$ define dimensionless re-normalized couplings. For the scaling solution characterizing a fixed point, the r.h.s. of Equation (18) has to vanish, resulting in a system of differential equations for $u$.

In Sections 3–5, we focus on an approximation for which $w$ is taken as a constant, independent of scalar fields and independent of $k$. In Section 6 we extend this to an ansatz $w = w_0 + \xi\phi^2/k^2$. In Sections 7 and 8 we discuss the full system of flow equations for $u$ and $w$. This supplements Equation (10) by a flow equation for $F$. In Appendix A we provide a summary of the flow of the calculations which should help the reader to identify the most important formula in a simple way.

### 3. Scaling Solutions for Matter Freedom

We first discuss an approximation for which all matter interactions are neglected. This approximation reveals some characteristic features of the effects of gravitational fluctuations on the scalar effective potential. We approximate here $F$ by a field-independent running squared Planck mass. At the UV-fixed point it scales $\sim k^2$, with fixed dimensionless parameter $w_0$,

$$F = M_p^2(k) = 2w_0 k^2 . \tag{19}$$

This basic result [2,4,62] of the use of functional re-normalization for asymptotically safe quantum gravity reflects directly the dimension of $F$ or the effective squared Planck mass. At the UV-fixed point the dimensionless ratio $w$ must be constant.

#### 3.1. Flow Equation for Matter Freedom

Let us first consider a situation where the values of gauge couplings, Yukawa couplings, dimensionless scalar mass terms, and quartic scalar couplings are sufficiently small such that the contribution of these fluctuations only matters for the flow of the field-independent part of $u$. We call this approximation "matter freedom" since the interactions between matter components are neglected. Approximating further $\eta_g = 0$, as valid for the scaling solution, and $\eta_A = 0$, corresponding to our neglection of the running of scalar wave functions, one finds:

$$\tilde{\zeta} = \frac{\overline{\zeta}}{k^4} = \frac{1}{24\pi^2} \left( \frac{5}{1-v} + \frac{1}{1-v/4} \right) + 4b_{\mathrm{U}} \tag{20}$$

with constant:

$$b_{\mathrm{U}} = \frac{N-4}{128\pi^2} . \tag{21}$$

The contribution of gravitational fluctuations can be directly inferred from Equation (11). The additional part $\sim N$ arises from matter fluctuations, with an effective number of degrees of freedom given by:

$$N = N_{\mathrm{S}} + 2N_{\mathrm{V}} - 2N_F . \tag{22}$$

Here $N_{\mathrm{S}}$ denotes the number of real scalars, $N_{\mathrm{V}}$ the number of gauge bosons ($N_{\mathrm{V}} = 45$ for $SO(10)$, $N_{\mathrm{V}} = 24$ for $SU(5)$ and $N_{\mathrm{V}} = 12$ for the standard model), and $N_F$ the number of Weyl fermions ($N_F = 48$ for $SO(10)$, $N_F = 45$ for $SU(5)$, and the standard model). For the standard model one has $N_{\mathrm{S}} = 4$, with much larger numbers of scalars for GUT models. For the standard model, $N = -62$ is negative, while GUT models typically have positive $N$. In the counting-only particles with masses much smaller than $k$ are included and approximated by massless particles.

We concentrate on a particular $N_{\mathrm{S}}'$-dimensional scalar representation and a potential $u(\tilde{\rho})$ depending only on the invariant:

$$\tilde{\rho} = \frac{1}{2} \sum_{a=1}^{N_{\mathrm{S}}'} \tilde{\phi}_a^2 . \tag{23}$$

The other $N_{\mathrm{S}} - N_{\mathrm{S}}'$ scalar fields may be set to zero (alternatively, one may consider fixed values for the dimensionless ratio as $\chi^2/k^2$ of some other scalar singlet field $\chi$, and consider $u(\tilde{\rho}, \chi^2/k^2)$ at fixed $\chi^2/k^2$). Our interest is the $\tilde{\rho}$-dependence of the potential. Neglecting the anomalous dimension $\eta_a$ the flow equation for the potential reads:

$$\partial_t u = \beta_u = -4u + 2\tilde{\rho}\, \partial_{\tilde{\rho}} u + 4c_{\mathrm{U}}(u), \tag{24}$$

with

$$c_{\mathrm{U}}(u) = \frac{1}{96\pi^2} \left( \frac{5}{1-v} + \frac{1}{1-v/4} \right) + b_{\mathrm{U}} \tag{25}$$

a non-linear function of $u$ through the dependence on $v = u/w$.

### 3.2. Constant Scaling Solutions

We are interested in the scaling solution at the UV-fixed point for which $\partial_t u$ vanishes. For any given $w(\tilde{\rho})$ this scaling solution for $u(\tilde{\rho})$ has to obey the nonlinear differential equation:

$$2\tilde{\rho}\frac{\partial u}{\partial \tilde{\rho}} = 4u - \frac{1}{24\pi^2}\left(\frac{5}{1 - u/w} + \frac{1}{1 - u/4w}\right) - 4b_U. \tag{26}$$

In general, $w$ depends on $\tilde{\rho}$. We first consider the case where the scaling form can be approximated by a constant $w = w_0$ and generalize this setting in Sections 6–8. A simple scaling solution is a constant potential,

$$u_*(\tilde{\rho}) = u_0. \tag{27}$$

For a given $w_0$, the value of $u_0$ obtains by setting the r.h.s. of Equation (26) to zero.

For more general scaling solutions we still may consider for $\tilde{\rho} \to 0$ the limit of $u$ approaching a constant,

$$u_*(\tilde{\rho} \to 0) = u_0. \tag{28}$$

If $\partial u/\partial \tilde{\rho}$ remains finite for $\tilde{\rho} \to 0$ (or does not diverge too strongly), the constant $u_0$ obtains again by setting the r.h.s. of Equation (26) to zero. Simplifying by approximating $(1 - v/4)^{-1}$ by $(1 - v)^{-1}$ yields a quadratic equation for $v_0 = u_0/w_0$, namely:

$$(v_0 + (N_0 - 4)z)(1 - v_0) = 8z, \quad z = \frac{1}{128\pi^2 w_0}, \tag{29}$$

with $N_0$ the effective particle number for $\tilde{\rho} = 0$.

Fixed point solutions for $v_0$ obey:

$$v_\pm = \frac{1}{2}\left\{1 + (N_0 - 4)z \pm \sqrt{(1 - (N_0 - 4)z)^2 - 32z}\right\}. \tag{30}$$

They exist provided $z$ is in a range where the argument of the square root is positive. For the special case $N_0 = -4$, the argument of the square root is positive for all $z$. The two solutions are $v_+ = 1 - 8z$, $v_- = 0$. For $N_0 < -4$, the argument of the square root is again always positive and one finds $v_+ > 1 - 8z$, $v_- < 0$. Restrictions on $z$ can arise for $N_0 > -4$. In this case $z$ has to be outside the interval $[z_-, z_+]$, given for $N_0 \neq 4$ by:

$$z_\pm = \frac{N_0 + 12 \pm 4\sqrt{2N_0 + 8}}{(N_0 - 4)^2}. \tag{31}$$

For $N_0 = 4$, the condition reads $z < 1/32$. For $N_0 \to 4$, the lower boundary $z_-$ approaches $1/32$ while $z_+$ diverges.

The precise relation between $v_0$ and $w_0$ or $z$ according to the solution of Equation (24) for $\partial_t u = 0$, $\tilde{\rho}\,\partial_{\tilde{\rho}} u = 0$ is algebraically less simple, but qualitatively and quantitatively similar [7]. We can infer it from Figure 1 which plots the relation between $v$ and $w$ in the form of a function $w(v)$. The latter follows from Equation (24),

$$w = \frac{c_U(v)}{v} = \frac{1}{96\pi^2 v}\left(\frac{5}{1 - v} + \frac{1}{1 - v/4}\right) + \frac{b_U}{v}. \tag{32}$$

Acceptable scaling solutions require $v < 1$, $w > 0$. For $N > -4$, the function $w(v)$ is positive for the interval $0 < v < 1$, diverging at both ends of the interval. This is the only allowed range. There is a minimum of $w(v)$ at $v_c$, with critical value $w_c = w(v_c)$. For $N > -4$, scaling solutions exist only for $w > w_c$. For $N < -4$, one finds positive $w$ for

negative $v$, with $w(v \to -\infty) \to 0$. A second solution with positive $w$ corresponds to a range of positive $v$ sufficiently close, but still smaller than the pole at $v = 1$.

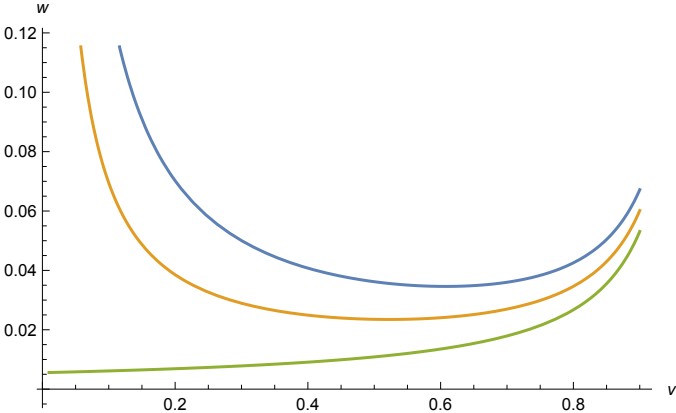

**Figure 1.** Relation between $v$ and $w$ for three values $N = 12$ (upper curve), $N = 4$ (middle curve), and $N = -4$ (lower curve).

For an appropriate range of $w$ one finds two solutions $v_+$ and $v_-$. They correspond to the two solutions $v_\pm$ of the approximation (28), (29). For $N > -4$ this requires $w > w_c$, corresponding to the restriction on $z$ given by Equation (31). We conclude that acceptable scaling solutions exist in our truncation for matter freedom, except for very strong gravity ($w_0 < w_c$) for $N > -4$.

We may interpret $u_0$ and $v_0$ as the limiting behavior of the scaling solution for $\tilde{\rho} \to 0$. For any given allowed value of $w_0$ or $z$, there are two possible values for $v_0$ and therefore two possible solutions for $u_0$. For $w(\tilde{\rho}) = w_0$ independent of $\rho$ Equation (26) actually admits a solution with constant $u(\tilde{\rho}) = u_0$ for all values of $\tilde{\rho}$. For this simple solution the effective potential is completely flat:

$$U(\rho) = u_0 k^4 . \tag{33}$$

Solutions with $\tilde{\rho}$-independent $u$ are called "constant scaling solutions". We will see in Section 7 that one of these constant scaling solutions corresponds to the extended Reuter fixed point.

*3.3. Crossover Scaling Solutions*

Since Equation (26) is a first order differential equation one may ask if there exist other scaling solutions with $\partial u / \partial \tilde{\rho} \neq 0$. For these solutions, the boundary conditions $u(\tilde{\rho} \to 0) = u_0$ should be obeyed. A numerical solution of Equation (26) indeed finds a family of scaling solutions that interpolate between the constant values $v_\pm$, as shown in Figures 2 and 3. For all values of $w_0$ compatible with the presence of two fixed points $v_+$ and $v_-$, the generic scaling solution is a crossover from $u(\tilde{\rho} \to 0) = v_- w$ to $u(\tilde{\rho} \to \infty) = v_+ w$. We show the numerical solutions of Equation (26) for different initial conditions (chosen arbitrarily at $\tilde{\rho} = 1$) in Figure 2. The different curves can be obtained by a shift in $x = \ln(\tilde{\rho})$. The crossover trajectory between the two fixed points is universal. The initial conditions only specify at which $\tilde{\rho}$, a given value of $u$, on the crossover trajectory is reached. The possible shifts in $x = \ln(\tilde{\rho})$ are arbitrary. Limiting cases are the constant scaling solutions $u(\tilde{\rho}) = v_- w$ or $u(\tilde{\rho}) = v_+ w$. For initial conditions with $u(\tilde{\rho}_{in})$ outside the interval $[v_- w, v_+ w]$, no scaling solution exists. Local solutions of the differential equation do not reach finite values for $\tilde{\rho} \to 0$ and $\tilde{\rho} \to \infty$. They typically diverge at some finite $\tilde{\rho}$.

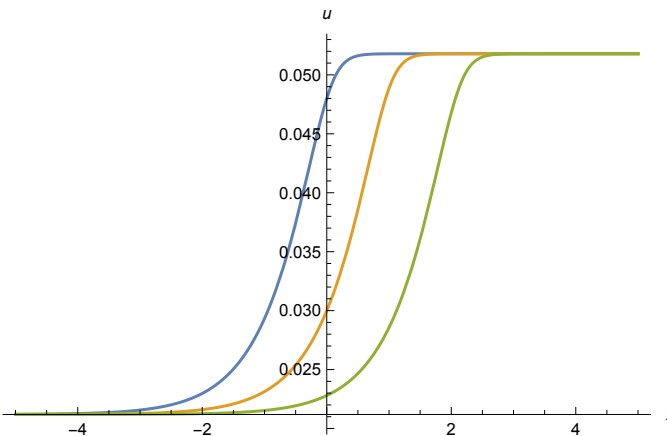

**Figure 2.** Scaling potential $u$ as a function of $x = \ln(\tilde{\rho})$. The three curves correspond to different initial conditions, which may be specified by $u(x = 0)$. The parameters are $N = 12$ and $w = 0.06$.

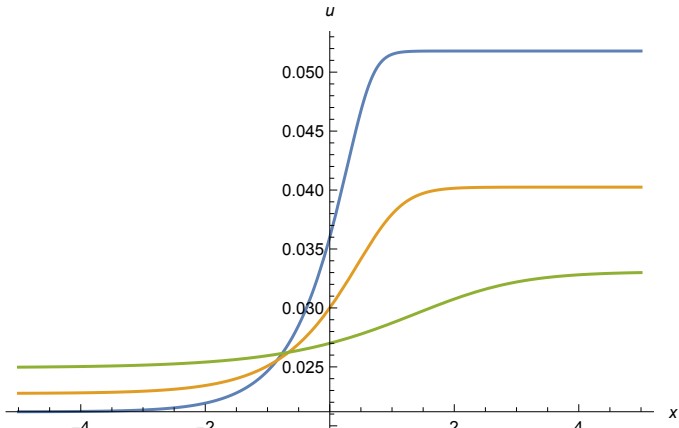

**Figure 3.** Scaling potential $u$ as a function of $x = \ln(\tilde{\rho})$ for different values of $w = 0.06$, $w = 0.05$, and $w = 0.045$ (upper, middle, and lower curve for large $x$, respectively). We use $N = 12$.

As $w$ is lowered, the interval $[v_-, v_+]$ shrinks. This is reflected by a shrinking of the distance between the boundary values $u(\tilde{\rho} \to 0)$ and $u(\tilde{\rho} \to \infty)$, as depicted in Figure 3. This shrinking continues until one reaches $w_c$ where $v_-(w_c) = v_+(w_c) = v_c$. For $w = w_c$ the unique scaling solution is a constant $u(\tilde{\rho}) = v_c w_c$. For $w < w_c$, the scaling solution ceases to exist.

### 3.4. Scalar Anomalous Dimension

The gravity-induced scalar anomalous dimension $A$ plays an important role for many aspects of the effective potential. We encounter it here first by an investigation of scaling solutions in the vicinity of the constant scaling solution. We will see later that it also governs the gravity induced flow of the scalar mass term and quartic coupling. Let us therefore investigate small deviations $\Delta u$ from the constant scaling solution. For the crossover solutions shown in Figures 2 and 3 they describe the onset of the crossover region. A linear approximation in $\Delta u$ will always become valid for $\tilde{\rho} \to 0$ and $\tilde{\rho} \to \infty$. In the vicinity of a constant scaling solution we expand:

$$u(\tilde{\rho}) = u_0 + \Delta u(\tilde{\rho}), \tag{34}$$

where $\Delta u(\tilde{\rho})$ obeys the linear differential equation:

$$2\tilde{\rho}\frac{\partial \Delta u}{\partial \tilde{\rho}} = (4 - A)\Delta u, \tag{35}$$

with

$$A = 4\frac{\partial c_U}{\partial u} = \frac{4}{w}\frac{\partial c_U}{\partial v}.$$ (36)

From Equation (25) one infers:

$$A = \frac{1}{96\pi^2 w}\left(\frac{20}{(1-v)^2} + \frac{1}{(1-v/4)^2}\right).$$ (37)

The gravity-induced anomalous dimension $A$ is a key quantity for the discussion of the gravitational effects on the scalar potential. For all $v$ and positive $w$ one finds $A(v,w) \geq 0$. The first term in Equation (37) is generated by the graviton fluctuations, while the second term originates from the physical scalar fluctuations in the metric. For positive $v$, the first term in Equation (37) dominates by more than a factor of 20, justifying the "graviton approximation" which keeps only the transversal traceless metric fluctuations [6].

We plot $A$ as a function of $v$ in Figure 4. For this purpose we use $w(v)$ according to the constant scaling solution (32), as shown in Figure 1. Inversion leads to two values $A_{\pm}$ for a given $w_0$, corresponding to the solutions $v_{\pm}$. For $N = -4$, one finds values $A < 4$ for negative $v$, not shown in Figure 4. For $v \to -\infty$, one reaches $A = 0$ (for $N \leq -4$). Generically, $A$ increases for decreasing $N$ and fixed $v$, and for increasing $v$ for fixed $N$. Away from the constant scaling, solution $A(\tilde{\rho})$ depends on the two functions $w(\tilde{\rho})$ and $v(\tilde{\rho})$ separately. For the region of small $\tilde{\rho}$ we have to evaluate $A(w,v)$ for $\tilde{\rho} \to 0$, i.e. $A_0 = A(w_0, v_0)$.

What is apparent already for the simple case of a constant scaling solution in Figure 4 is that $A$ is typically not a very small quantity. Generally, $A$ is positive and not very small as compared to one. It can exceed value one for a suitable range of $w$ and $v$.

In Appendix B, we discuss the solution of Equation (35), as well as the general form of candidate scaling solutions for matter freedom. We find that for $A_0 \neq 2$ higher order derivatives of the effective potential, as the quartic scalar coupling, $\lambda(\tilde{\rho}) = \partial^2 u/\partial\tilde{\rho}^2$, diverge for $\tilde{\rho} \to 0$ for the crossover scaling solution. The neglection of the scalar masses in the scalar fluctuation contribution $\tilde{\pi}_s$ in Equation (15) is no longer satisfied in the region $\tilde{\rho} \to 0$.

### 3.5. Scalar Mass Term

In Appendix C, we discuss in detail the influence of the scalar mass terms which is due to a more complete treatment of $\tilde{\pi}_s$. The scalar mass term is typically found to be small for many of our solutions, including the following sections. For the crossover solution of matter freedom we show the scalar mass term $\tilde{m}^2(\tilde{\rho}) = \partial u/\partial\tilde{\rho}$ in Figure 5. As the location of the crossover, which may be associated with the maximum of $\tilde{m}^2$, moves to a larger $x$ the height of the maximum decreases. Matter domination could therefore provide for a rather accurate picture for the sub-family of scaling solutions where the crossover happens at large $x = \ln(\tilde{\rho})$. Even for a crossover at somewhat negative $x$ we find $\tilde{m}^2 \ll 1$, such that at first sight a neglection of the mass term $\tilde{m}^2$ in $\tilde{\pi}_s$, and therefore the approximation of matter freedom, seem justified for large regions in $\tilde{\rho}$.

An exception is the region around the origin at $\tilde{\rho} = 0$. Adding even a small mass term will dominate the small deviations from the constant scaling solution. We present in Appendix C a detailed discussion of the influence of the scalar mass term on the flow equation and scaling solutions for the effective potential. For models of scalars coupled to gravity we find that it matters in a region of very small $\tilde{\rho}$, typically $\tilde{\rho} \lesssim N_S/(64\pi^2)$. It cures the otherwise divergent behavior of the quartic and higher order scalar couplings. This will also become apparent in the next section.

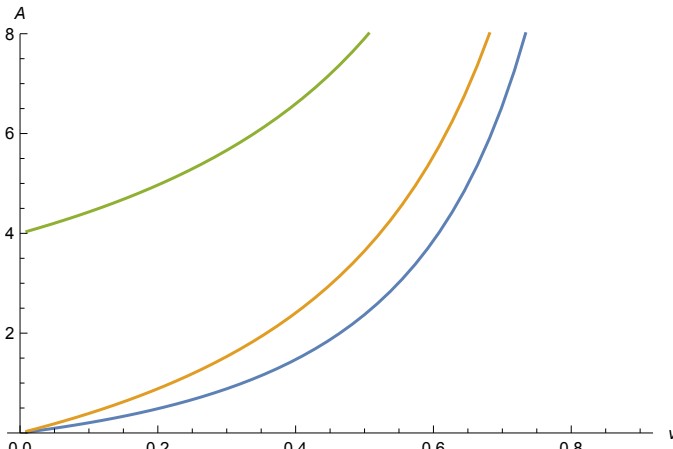

**Figure 4.** Dimensionless mass term $\tilde{m}^2$ as a function of $x = \ln(\tilde{\rho})$. The plot is for $w = 0.06$, using the parameters of Figure 2. For a location of the crossover at larger $x$, the height of the maximum decreases.

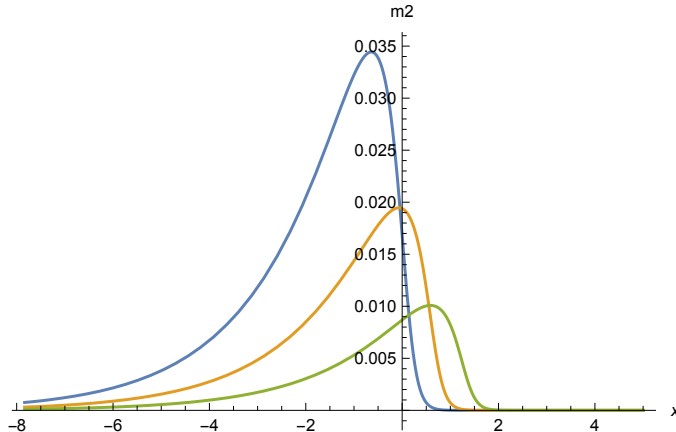

**Figure 5.** Relation between $A$ and $v$, for $v$ being a fixed point solution for an appropriate $w$ as given by the relation (32). We show three values of $N = 12$ (lower curve), $N = 4$ (middle curve), and $N = -4$ (upper curve).

In models of scalars coupled to gravity, with vanishing gauge and Yukawa couplings, there still remain some unsettled issues in the transition region around $\tilde{\rho}_t = N_S/(64\pi^2)$. Their resolution would require numerical solutions beyond the simplified approaches employed in Appendix C. We will mainly be interested in the following in situations with non-zero gauge couplings, Yukawa couplings, or non-minimal scalar-gravity couplings. All these couplings generate non-zero mass terms and change the behavior for $\tilde{\rho} \to 0$, removing potential singularities for $\tilde{\rho} \to 0$ even if the modifications of $\tilde{\pi}_s$ due to $\tilde{m}^2 \neq 0$ are omitted. In Sections 5–7, we will neglect the modifications of $\tilde{\pi}_s$ by non-zero scalar mass terms. We may consider this approximation as justified whenever $\tilde{m}^2(\tilde{\rho}) < 0.1$ for the whole range of $\tilde{\rho}$. In other words, we approximate scalar fluctuation contributions as arising from massless scalars. This procedure simplifies the discussion of scaling solutions considerably, since the right-hand side of Equation (26) and its generalizations depend on $\tilde{\rho}$ and $u(\tilde{\rho})$, but no longer on derivatives of $u(\tilde{\rho})$ with respect to $\tilde{\rho}$. For the numerically found solutions, caution is required. The numerical approach may be blind to spiky behavior of $u(\tilde{\rho})$ in very small regions of $\tilde{\rho}$, which could lead to larger values of $\tilde{m}^2$ in these regions.

## 4. Flow in the Vicinity of the Scaling Solution

We next turn to the flow with $k$ at fixed $\tilde{\rho}$. This is an alternative way to discuss properties of scaling solutions. For a scaling solution, the flow with $k$ for constant $\tilde{\rho}$ has to

stop. This also holds for all particular couplings that may be defined by some $\tilde{\rho}$-derivatives of $u$ at fixed $\rho_0$. Examples are the scalar mass term or quartic coupling at the origin,

$$\tilde{m}_0^2 = \left.\frac{\partial u}{\partial \tilde{\rho}}\right|_{\tilde{\rho}=0}, \quad \lambda_0 = \left.\frac{\partial^2 u}{\partial \tilde{\rho}^2}\right|_{\tilde{\rho}=0}. \tag{38}$$

Within suitable approximations one may obtain a closed system of flow equations for a finite number of couplings. This may be solved without the need to solve a differential equation for all values of $\tilde{\rho}$. One may then look for fixed points in a system of flow equations for a finite number of couplings.

Beyond the fixed point solution, the flow with $k$ also tells if a fixed point is approached for decreasing $k$ (irrelevant couplings) or if the flow trajectories move away from the fixed points (irrelevant couplings). We discuss this issue in Appendix C.3.

Defining $\tilde{m}^2(\tilde{\rho})$ and $\lambda(\tilde{\rho})$ by the first and second $\tilde{\rho}$-derivatives of $u$:

$$\tilde{m}^2(\tilde{\rho}) = \frac{\partial u}{\partial \tilde{\rho}}, \quad \lambda(\tilde{\rho}) = \frac{\partial^2 u}{\partial \tilde{\rho}^2}, \tag{39}$$

we first derive the flow equations in the limit of matter freedom where the contributions of vector bosons, fermions, and scalars can be approximated by a constant $b_U$. They follow from Equation (24) by first and second differentiation with respect to $\tilde{\rho}$.

The flow of the mass term obeys:

$$\partial_t \tilde{m}^2(\tilde{\rho}) = (A(\tilde{\rho}) - 2)\tilde{m}^2(\tilde{\rho}) + 2\tilde{\rho}\lambda(\tilde{\rho}). \tag{40}$$

We observe the appearance of the gravity-induced anomalous dimension $A(\tilde{\rho})$ given by Equation (37) in terms of $w_0$ and $v(\tilde{\rho}) = u(\tilde{\rho})/w_0$. The appearance of $A$ both for small deviations from a given scaling solutions and for the flow with $k$ is no accident. Both problems concern small changes of a given potential $u(\tilde{\rho})$. Taking a further $\tilde{\rho}$-derivative yields:

$$\partial_t \lambda(\tilde{\rho}) = A(\tilde{\rho})\lambda(\tilde{\rho}) + 2\tilde{\rho}\frac{\partial\lambda(\tilde{\rho})}{\partial\tilde{\rho}} + \frac{1}{w}\frac{\partial A}{\partial v}\tilde{m}^4(\tilde{\rho}). \tag{41}$$

Here $\partial A/\partial v$ obtains from Equation (37), again evaluated for $w = w_0$ and $v = v(\tilde{\rho}) = u(\tilde{\rho})\, w_0$.

Consider first the limit where $\tilde{\rho}\,\partial\lambda/\partial\tilde{\rho}$ can be neglected. In this case the fixed point of the flow occurs for:

$$\tilde{m}_*^2 = 0, \quad \lambda_* = 0. \tag{42}$$

This is realized for the scaling solution for $\tilde{\rho} \to \infty$. Indeed, for an asymptotic behavior of the scaling solution for $\tilde{\rho} \to \infty$,

$$u_*(\tilde{\rho}) = u_\infty + c_\infty \tilde{\rho}^{\frac{4-A_\infty}{2}}, \tag{43}$$

and $A_\infty > 4$ one finds that:

$$\tilde{m}_*^2(\tilde{\rho}) = c_\infty\left(2 - \frac{A_\infty}{2}\right)\tilde{\rho}^{1-\frac{A_\infty}{2}}, \tag{44}$$

and

$$\lambda_*(\tilde{\rho}) = c_\infty\left(2 - \frac{A_\infty}{2}\right)\left(1 - \frac{A_\infty}{2}\right)\tilde{\rho}^{-\frac{A_\infty}{2}}. \tag{45}$$

Both approach zero for $\tilde{\rho} \to \infty$ (in this section, and more generally if needed, we denote by stars the scaling solutions or fixed points).

Linearizing the flow in the vicinity of the fixed point (42) yields (in the approximation $\partial \lambda / \partial \tilde{\rho} = 0$):

$$\partial_t \tilde{m}^2 = (A - 2) \tilde{m}^2 + 2 \tilde{\rho} \lambda \, ,$$
$$\partial_t \lambda = A \lambda \, . \tag{46}$$

These flow equations hold strictly for $\tilde{\rho} \to \infty$. Replacing $\tilde{m}^2$ by $\Delta \tilde{m}^2 = \tilde{m}^2 - \tilde{m}_*^2(\tilde{\rho})$, and $\lambda$ by $\Delta \lambda = \lambda - \lambda_*(\tilde{\rho})$, they also hold for $\Delta m^2$ and $\Delta \lambda$ at finite large $\tilde{\rho}$ to a very good approximation. The solution for $\Delta \lambda$:

$$\Delta \lambda = \tilde{c}_\lambda k^A \tag{47}$$

drives $\Delta \lambda$ to its fixed point value $\Delta \lambda_* = 0$ as $k$ is lowered. Thus $\lambda$ is an irrelevant coupling at the quantum gravity fixed point. For a complete theory that can be continued to arbitrary large $k$ according to the quantum gravity fixed point, one predicts $\lambda(\tilde{\rho})$ to be given by the scaling solution $\lambda_*(\tilde{\rho})$. For $\Delta \tilde{m}^2$ one obtains the solution:

$$\Delta \tilde{m}^2 = \tilde{c}_m k^{A-2} + \tilde{c}_\lambda \tilde{\rho} \, k^A \, . \tag{48}$$

With $A_\infty > 4$ also $\Delta \tilde{m}^2$ is irrelevant and $\tilde{m}^2(\tilde{\rho})$ is predicted to be the scaling solution $\tilde{m}_*^2(\tilde{\rho})$. These properties hold for the region of large $\tilde{\rho}$ for which $A(\tilde{\rho})$ exceeds 4.

The situation is more complicated for $\tilde{\rho} = 0$. If $\partial \lambda / \partial \tilde{\rho}$ remains finite or does not increase too rapidly for $\tilde{\rho} \to 0$, one finds again a fixed point $\lambda_* = 0$, $\tilde{m}_*^2 = 0$, and solutions in the vicinity of the fixed point:

$$\tilde{m}_0^2 = c_m k^{A-2} \, , \quad \lambda_0 = c_\lambda k^A \, . \tag{49}$$

Now $A$ is given by $A_0$ and therefore smaller than four. Since $A$ is positive, $\lambda_0$ is an irrelevant parameter and predicted to be at its fixed point value $\lambda_* = 0$. The mass term is irrelevant for $A > 2$, predicted to be $\tilde{m}_*^2 = 0$ in this case. For $A < 2$ it is a relevant parameter. Its value cannot be predicted since it involves the free constant $c_m$. We discuss in Appendix C.2 under which circumstances the scaling solution indeed leads to $\tilde{m}_*^2 = 0$, $\lambda_* = 0$ if gauge and Yukawa couplings are neglected and $w_*$ is a constant.

In the approximation of matter freedom the solution (49) only holds for the constant scaling solutions. In this case matter freedom is a self-consistent approximation for the scaling solution. For the crossover scaling solutions we find in Appendix B that the approximation of matter freedom leads to a divergence of $\lambda(\tilde{\rho})$ for $\tilde{\rho} \to 0$ such that the fixed point $\tilde{m}_*^2 = 0$, $\lambda_* = 0$ is not realized. This seems to contradict result (49). Taking into account in Appendix C, the deviation of the scalar contributions $\tilde{\pi}_s$ from the matter-freedom approximation yields for the flow of $\tilde{m}_0^2$ an additional contribution,

$$\partial_t \tilde{m}_0^2 = (A_0 - 2) \tilde{m}_0^2 - \frac{3 \lambda_0}{32 \pi^2 (1 + \tilde{m}_0^2)^2}. \tag{50}$$

This allows for a fixed point with $\tilde{m}_0^2 \neq 0$, as characteristic for the crossover scaling solutions, provided that $\lambda_0 \neq 0$. Now the first Equation (49) holds for $\tilde{m}_0^2(k) - \tilde{m}_0^2$. Similar properties hold for $\lambda_0$.

We observe a connection between the behavior of deviations from the scaling solution and the asymptotic behavior of the scaling solution itself for $\tilde{\rho} \to 0$ and $\tilde{\rho} \to \infty$. Both are given by the same anomalous dimension $A$. The root of this connection resides in the general form of the flow equation for $u$ at fixed $\tilde{\rho}$ that can be written in the form:

$$(\partial_t - 2 \tilde{\rho} \, \partial_{\tilde{\rho}}) u = -4(u - c_U) = \tilde{\beta}_u \, . \tag{51}$$

A similar form holds for $\tilde{\rho}$-derivatives, as $\tilde{m}^2(\tilde{\rho}) = \partial_{\tilde{\rho}} u(\tilde{\rho})$,

$$(\partial_t - 2\tilde{\rho}\,\partial_{\tilde{\rho}})\tilde{m}^2 = -2\tilde{m}^2 + 4\partial_{\tilde{\rho}} c_U = \tilde{\beta}_{\tilde{m}^2} \,. \tag{52}$$

For the scaling solution one has:

$$2\tilde{\rho}\,\partial_{\tilde{\rho}} u = -\tilde{\beta}_u \,. \tag{53}$$

On the other hand, if $\tilde{\rho}\,\partial_{\tilde{\rho}} u$ can be neglected for $\tilde{\rho} \to 0$ or $\tilde{\rho} \to \infty$, one finds for these limits:

$$\partial_t u = \tilde{\beta}_u \,. \tag{54}$$

Both expressions (53) and (54) involve the same $\beta$-function $\tilde{\beta}_u$, but with the opposite sign. Thus $k^2 \to 0$ corresponds to increasing $\tilde{\rho}$.

So far we have obtained a consistent picture for both limiting regions $\tilde{\rho} \to 0$ and $\tilde{\rho} \to \infty$. The difficult issue contains the matching of these regions in a transition region around $\tilde{\rho}_s = 1/(64\pi^2)$. We discuss this question in detail in Appendix C.5. So far the only established scaling solutions are the constant scaling solutions. It may not be possible to follow the crossover solutions through the transition region in a regular way. A definite answer to the question if there exist global crossover scaling solutions would need a more sophisticated numerical approach than the one employed in the present work.

## 5. Gauge Couplings

The presence of non-vanishing gauge couplings or Yukawa couplings leads to important qualitative changes for the scaling solution as compared to matter freedom or scalars coupled only to gravity. The reason is that for non-vanishing gauge couplings, constant scaling solutions with $u(\tilde{\rho})$ independent of $\tilde{\rho}$ are no longer possible. Gauge or Yukawa couplings necessarily induce non-zero scalar mass terms $\partial u/\partial \tilde{\rho}$ for all $\tilde{\rho} \neq 0$. We will consider here the case of constant couplings. This refers either to a fixed point with non-zero gauge or Yukawa couplings, or to an approximation for a situation with slow enough flow of these couplings. We concentrate first on vanishing Yukawa couplings. This is directly relevant for the issue of spontaneous symmetry breaking in GUT-models, where some of the relevant scalar fields are in representations that do not allow for Yukawa couplings to the fermions of the known three generations.

### 5.1. Flow Equations

In this section we investigate the impact of non-vanishing gauge couplings on the flow of the scalar effective potential. For nonzero values of scalar fields coupling to gauge bosons with a gauge coupling $g$, the gauge bosons acquire a mass through the Higgs mechanism. This mass suppresses the contribution of gauge bosons to the flow of the scalar potential. As a result, for non-vanishing gauge couplings the flow generator $\bar{\zeta}$ in Equation (10) receives an additional contribution $\Delta\tilde{\pi}_{\text{gauge}}$, given by:

$$\frac{\Delta\tilde{\pi}_{\text{gauge}}}{k^4} = \frac{3}{32\pi^2} \sum_{i=1}^{N_V} \left(\frac{1}{1+w_i} - 1\right) . \tag{55}$$

Here the sum is over all gauge bosons and $w_i = m_i^2/k^2$ are the dimensionless squared gauge boson masses for the corresponding values of the scalar field:

$$w_i = \frac{m_i^2}{k^2} = g^2 a_i(\phi_a)/k^2 \,. \tag{56}$$

Typically, $a_i(\phi_a)$ is a quadratic form in the scalar fields $\phi_a$. The factor $(1 + w_i)^{-1}$ is a threshold function that suppresses the contribution from massive gauge bosons as compared to the massless ones.

In order to keep the discussion of the structure of these modifications simple we consider a toy flow equation where the squared mass of $\bar{N}_V$ gauge bosons is $c_g g^2 \rho$, while the other gauge bosons remain massless for the particular configuration of scalar fields that is used to define $\rho$. From the difference of the fluctuation contribution from massless and massive fields one obtains the modification of $\tilde{\pi}_{\text{gauge}}$,

$$\frac{\Delta \tilde{\pi}_{\text{gauge}}}{k^4} = -\frac{3 c_g \bar{N}_V g^2 \tilde{\rho}}{32 \pi^2 \left(1 + c_g g^2 \tilde{\rho}\right)} \ . \tag{57}$$

It vanishes for $\tilde{\rho} = 0$. The contribution to the flow of the scalar mass term at the origin $\tilde{\rho} = 0$ reads:

$$\partial_t \tilde{m}^2 = -\frac{3 c_g \bar{N}_V g^2}{32 \pi^2} + \dots , \tag{58}$$

while the contribution to the quartic coupling at $\tilde{\rho} = 0$, $\lambda = \partial^2 \tilde{U} / \partial \tilde{\rho}^2|_{\tilde{\rho}=0}$, becomes:

$$\partial_t \lambda = \frac{3 c_g^2 \bar{N}_V g^4}{16 \pi^2} + \dots \tag{59}$$

Equation (59) corresponds to the standard perturbative term $\sim g^4$ in the flow equation for quartic scalar couplings.

### 5.2. Spontaneous Symmetry Breaking

Due to the negative sign in Equation (57) a non-vanishing gauge coupling lowers the scaling solution for the effective potential for large $\tilde{\rho}$ as compared to $\tilde{\rho} = 0$. Indeed, the differential equation (26) for the scaling solution receives an additional positive contribution:

$$2\tilde{\rho} \frac{\partial u}{\partial \tilde{\rho}} = 4(u - c_{\text{U}}) - \frac{\Delta \tilde{\pi}_{\text{gauge}}}{k^4} , \tag{60}$$

enhancing the increase of $u$ with $\tilde{\rho}$. Initial conditions near $\tilde{\rho} = 0$ that would lead to a decrease of $u$ with increasing $\tilde{\rho}$ may be turned to an increase with $\tilde{\rho}$ for larger $\tilde{\rho}$. As a result, the effective potential can develop a minimum for $\tilde{\rho} \neq 0$. This is clearly seen for a numerical solution of Equation (60) in Figure 6, shown in more detail in Figure 7. We employ $N = 10$, $\bar{N}_V = 3$, $c_g = 1$, and $\alpha = g^2/4\pi = 1/40$, and set the initial condition by $u(\ln(\tilde{\rho}) = 1) = 0.02$. The three values of $w_0 = 0.06, 0.05, 0.042$ used in Figures 6 and 7 correspond to $A_0 = A(\tilde{\rho} = 0) = 0.68$, 1.0, and 1.64, and therefore all have $A_0 < 2$.

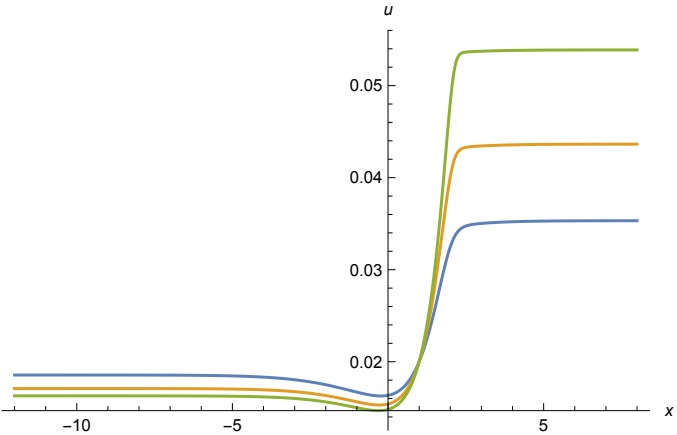

**Figure 6.** Effective potential $u(x)$ as function of $x = \ln(\tilde{\rho})$ for three values of $w_0$. The highest curve on the right (green) is for $w_0 = 0.06$, the middle curve (orange) for $w_0 = 0.05$, and the lowest curve (blue) for $w_0 = 0.042$. Initial values are set by $U(x = 1) = 0.02$. Parameters are $N = 10$, $\bar{N}_V = 3$, and $\alpha = g^2/4\pi = 1/40$.

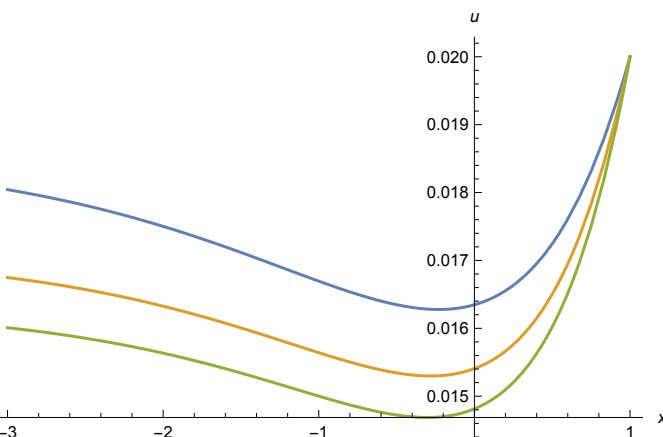

**Figure 7.** Minimum of effective potential $u(x)$. Parameters are the same as for Figure 6. The lowest green curve is for $w_0 = 0.06$, the middle orange curve for $w_0 = 0.05$, and the highest blue curve for $w_0 = 0.042$.

The minimum of $u(\tilde{\rho})$ at $\tilde{\rho}_0 \neq 0$ indicates spontaneous symmetry breaking already for the scaling solution. In this case the flow below the Planck mass away from the fixed point is not needed in order to induce spontaneous symmetry breaking. In most circumstances it will only have a sizable effect on small values of $\rho$, with $U(\rho)$ for large $\rho$ frozen at the value it has reached for $k \approx M$, or more precisely $k \approx k_t$. The minimum of the scaling potential will not be erased in this way. For our set of parameters it corresponds to $\rho \approx \tilde{\rho}M^2/(2w_0)$ according to Equation (8). With $\tilde{\rho}$ near one in Figure 5, one obtains for the parameters chosen a potential with a minimum at $\rho_0$ somewhat larger than $M^2$.

For realistic GUT models, the numbers $N_V$ and $\bar{N}_V$ are typically larger than the ones considered here. In addition, the influence of non-minimal couplings becomes important, see Section 7.6. We will not discuss in this paper the interesting question under which circumstances values of $\rho_0$ substantially smaller than $M^2$ are reached. We rather concentrate on general features of the scaling potential.

For a more detailed understanding we first consider the region of small $\tilde{\rho}$. The flow equation for the mass term at the origin, $\tilde{m}_0^2 = \tilde{m}^2(\tilde{\rho} = 0)$ reads, with $\lambda_0 = \lambda(\tilde{\rho} = 0)$,

$$\partial_t \tilde{m}_0^2 = (A_0 - 2)\tilde{m}_0^2 - \frac{3\lambda_0}{32\pi^2\left(1 + \tilde{m}_0^2\right)} - \frac{3c_g\bar{N}_V g^2}{32\pi^2}. \tag{61}$$

The fixed point occurs for ($A_0 \neq 2$):

$$\tilde{m}_{0,*}^2 = \frac{3}{32\pi^2(A_0 - 2)}\left(c_g\bar{N}_V g^2 + \frac{\lambda_0}{\left(1 + \tilde{m}_{0,*}^2\right)^2}\right). \tag{62}$$

For small $\lambda_{0,*}$ and $A_0 < 2$ the mass term is negative,

$$\tilde{m}_{0,*}^2 = -\frac{3c_g\bar{N}_V g^2}{32\pi^2(2 - A_0)}, \tag{63}$$

such that the origin at $\tilde{\rho} = 0$ is a local maximum of the effective potential. This is seen for the curves in Figures 6 and 7 which indeed have all $A_0 < 2$. We show in Figure 8 the mass term $\tilde{m}^2(\tilde{\rho})$ for the scaling solutions for the three sets of parameters used in Figures 6 and 7. For $\tilde{\rho} \to 0$ the result for $\tilde{m}^2(\tilde{\rho} \to 0)$ comes indeed very close to value (63).

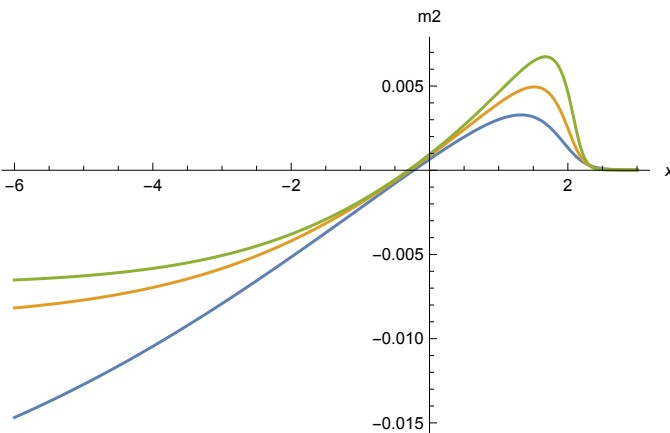

**Figure 8.** Mass term $\tilde{m}^2(x)$ as function of $x = \ln(\tilde{\rho})$ for $w_0 = 0.06$ (upper green curve), $w_0 = 0.05$ (middle orange curve), and $w_0 = 0.042$ (lower blue curve). Parameters are the same as for Figure 6.

For $A_0 > 2$, a negative value of $\tilde{m}_{0,*}^2$ remains possible if $\lambda_{0,*} < 0$. Indeed, the potential may have at the origin a local minimum or a maximum, depending on the relative size of the two terms on the r.h.s. of Equation (62). The fixed point value for $\lambda_{0,*}$ is negative, given for a single scalar by:

$$\lambda_{0,*} = -\frac{1}{A_0} \left( \frac{3 c_g^2 \bar{N}_V g^4}{16\pi^2} + \frac{9\lambda_{0,*}}{16\pi^2 (1 + \tilde{m}_{0,*}^2)^3} \right.$$
$$\left. - \frac{5 u_{0,*}^{(3)}}{32\pi^2 (1 + \tilde{m}_{0,*}^2)^2} + \frac{1}{w_0} \frac{\partial A}{\partial v} \tilde{m}_{0,*}^2 \right). \tag{64}$$

The size of a negative $\lambda_{0,*}$ and its influence is increased as $\tilde{m}_{0,*}^2$ comes close to $-1$. We show in Figure 9 the numerical scaling solutions for the effective potential with parameters $N = 20$, $\bar{N}_V = 3$, $c_g = 1$, $w_0 = 0.05$, and $\alpha = 1/40$. For these parameters one has $A_0 = 2.91$, and therefore $A_0 > 2$. The mass term is an irrelevant coupling in this case. Depending on the initial conditions we found solutions with a minimum at the origin or a minimum at $\tilde{\rho}_0 \neq 0$.

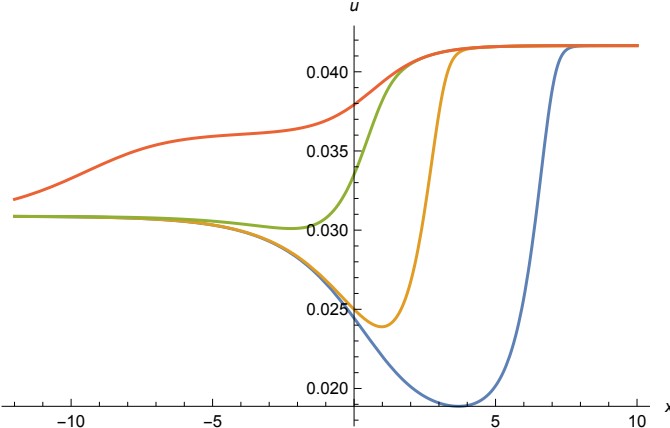

**Figure 9.** Effective potential $u(x)$ as function of $x = \ln(\tilde{\rho})$. Parameters are $N = 20$, $\bar{N}_V = 3$, $c_g = 1$, $w_0 = 0.05$, and $\alpha = g^2/4\pi = 1/40$. The initial conditions for the four curves from up to down are $u(x = 0) = 0.037923$, $u(x = 0) = 0.0335$, $u(x = 0) = 0.25$, and $u(x = 0) = 0.24447$. The initial values for the upper and lower curves limit the interval for which a scaling solution is found. For these solutions one has $A_0 = 2.91$.

*5.3. Asymptotic Behavior*

Consider next the limit $\tilde{\rho} \to \infty$. In this limit the correction (57) approaches a constant:

$$\lim_{\tilde{\rho} \to \infty} \left( \frac{\Delta \tilde{\pi}_{\text{gauge}}}{k^4} \right) = -\frac{3\bar{N}_V}{32\pi^2} . \tag{65}$$

This simply reflects that in the range of large $\tilde{\rho}$ only a reduced number of gauge bosons contributes to the number of "active degrees of freedom" $N$ in Equations (21)and (22). The gauge boson contribution to the running of $\tilde{m}^2(\tilde{\rho})$ and $\lambda(\tilde{\rho})$ is suppressed for large $\tilde{\rho}$ by threshold functions:

$$\partial_t \tilde{m}^2(\tilde{\rho}) = -\frac{3c_g \bar{N}_V g^2}{32\pi^2 \left(1 + c_g g^2 \tilde{\rho}\right)^2} + \dots ,$$

$$\partial_t \lambda(\tilde{\rho}) = \frac{3c_g^2 \bar{N}_V g^4}{16\pi^2 \left(1 + c_g g^2 \tilde{\rho}\right)^3} + \dots \tag{66}$$

that account for the decoupling of heavy degrees of freedom. As a consequence, the scaling solution for the effective potential reaches for $\tilde{\rho} \to \infty$, again a constant value, but with a different number of degrees of freedom $N_\infty$. Denoting by $N_0$ the number of light degrees of freedom for $\tilde{\rho} \to 0$, and $N_\infty$ the corresponding one for $\tilde{\rho} \to \infty$, one has:

$$N_\infty = N_0 - 3\bar{N}_V . \tag{67}$$

In contrast, the constant value $u_0 = u(\tilde{\rho} \to 0)$ is only indirectly influenced by $g^2 \neq 0$ due to nonzero $\tilde{m}_0^2$.

We can formulate this issue more generally. Applying the defining differential Equation (26) for the scaling form of the potential to a situation where $u(\tilde{\rho})$ is approximated by a constant both for $\tilde{\rho} \to 0$ and for $\tilde{\rho} \to \infty$ we obtain for $u_0$ and $u_\infty = u(\tilde{\rho} \to \infty)$:

$$u_0 = \frac{1}{96\pi^2} C_0 , \quad u_\infty = \frac{1}{96\pi^2} C_\infty , \tag{68}$$

with

$$C_0 = \frac{5}{1 - v_0} + \frac{1}{1 - v_0/4} + \frac{3(N_0 - 4)}{4} ,$$

$$C_\infty = \frac{5}{1 - v_\infty} + \frac{1}{1 - v_\infty/4} + \frac{3(N_\infty - 4)}{4} . \tag{69}$$

Here $v_0 = u_0/w_0$, $v_\infty = u_\infty/w_\infty$ with $w_0$ and $w_\infty$ the dimensionless coefficients of the curvature scalar for $\tilde{\rho} \to 0$ and $\tilde{\rho} \to \infty$, respectively. Similarly, $N_0$ and $N_\infty$ denote the number of effective matter degrees of freedom in the two limits. The potential difference:

$$\Delta u_\infty = u_\infty - u_0 = \frac{1}{96\pi^2} (C_\infty - C_0) \tag{70}$$

is positive if $C_\infty$ is larger than $C_0$. If the scalar field represented by $\tilde{\rho}$ does not couple to fermions one has $N_\infty < N_0$ if some of the bosons decouple effectively for $\tilde{\rho} \to \infty$, as in the case of gauge bosons discussed above. If also $v_\infty < v_0$ one concludes $u_\infty < u_0$. In this case one typically encounters a minimum of the effective potential for $\tilde{\rho} \to \infty$. In contrast, for $v_\infty > v_0$ one may find $u_0 < u_\infty$. This is the case for the examples shown in Figures 6–9. The minimum of $u(\tilde{\rho})$ occurs now for $\tilde{\rho} = 0$ or for finite $\tilde{\rho}$. For both cases the potential is flat for $\tilde{\rho} \to \infty$.

*5.4. Crossover Region*

For $g \neq 0$, constant scaling solutions are no longer possible. Every scaling solution has to make a crossover from $u_0$ for $\tilde{\rho} \to 0$ to $u_\infty$ for $\tilde{\rho} \to \infty$. The gauge coupling also changes the dynamics in the transition region as compared to scalars which have only gravitational interactions. In Figures 6–9, no apparent problem is visible in the transition region and it seems that a whole family of scaling solutions exists. We also observe that for all parameters and initial conditions investigated here the scalar mass term remains much smaller than one.

For a more quantitative understanding of the flow of the potential we may neglect the effect of the scalar mass term and quartic coupling in the contribution from the scalar fluctuations, e.g. setting $\Delta \tilde{\pi}_s = 0$. This approximation is always valid for large $\tilde{\rho}$. In the presence of gauge couplings, the validity may in certain cases extend to the whole range of $\tilde{\rho}$. In this approximation one finds for $\tilde{m}^2(\tilde{\rho}) = \partial u / \partial \tilde{\rho}$ in case of a single gauge field, $\bar{N}_V = 1$, $c_g = 1$,

$$\partial_t \tilde{m}^2(\tilde{\rho}) = (A - 2)\tilde{m}^2(\tilde{\rho}) + 2\tilde{\rho}\frac{\partial \tilde{m}^2}{\partial \tilde{\rho}} - \frac{3g^2}{32\pi^2}\left(1 + g^2\tilde{\rho}\right)^{-2}, \tag{71}$$

where

$$\frac{\partial \tilde{m}^2}{\partial \tilde{\rho}} = \lambda(\tilde{\rho}). \tag{72}$$

The scaling solution is defined by $\partial_t \tilde{m}^2(\tilde{\rho}) = 0$ and therefore obeys the differential equation:

$$2\tilde{\rho}\frac{\partial \tilde{m}^2}{\partial \tilde{\rho}} = (2 - A)\tilde{m}^2 + \frac{3g^2}{32\pi^2}\left(1 + g^2\tilde{\rho}\right)^{-2}. \tag{73}$$

The solution for small $\tilde{\rho} \to 0$ is given by Equation (62). For a given $\lambda_{0,*}$ this boundary condition fixes the integration constant of the general solution of Equation (73). On the other hand, fixing the integration constant by an initial value $\tilde{m}^2_{0,*}$ at $\tilde{\rho} = 0$, the fixed point value $\lambda_{0,*}$ follows from the solution of Equation (73).

For $\tilde{\rho} \to \infty$ the scaling solution for the potential becomes flat, namely:

$$\tilde{m}^2(\tilde{\rho} \to \infty) = 0, \tag{74}$$

implying also:

$$\lambda(\tilde{\rho} \to \infty) = \frac{\partial \tilde{m}^2}{\partial \tilde{\rho}}(\tilde{\rho} \to \infty) = 0. \tag{75}$$

For finite large $\tilde{\rho}$ we can approximate Equation (73) by:

$$\begin{aligned}\frac{\partial \tilde{m}^2}{\partial \rho} &= \frac{3g^2}{16(6 - A_\infty)\pi^2\tilde{\rho}(1 + g^2\tilde{\rho})^2} \\ &\approx \frac{3}{16(6 - A_\infty)\pi^2 g^2\tilde{\rho}^3}.\end{aligned} \tag{76}$$

Here, we employ $\tilde{m}^2 \sim \tilde{\rho}^{-2} \sim -(\tilde{\rho}\partial_{\tilde{\rho}}\tilde{m}^2)/2$. The asymptotic scaling solution for $\tilde{\rho} \to \infty$ is:

$$\tilde{m}^2 = \frac{3}{32(A_\infty - 6)\pi^2 g^2\tilde{\rho}^2}. \tag{77}$$

One typically has $A_\infty > 6$, $\tilde{m}^2 > 0$, such that $u_\infty$ is approached from below.

The differential equations for the scaling solution for the potential (60) or for a scalar mass term (73) do not show any problematic region. It seems likely that many initial conditions chosen at some intermediate $\tilde{\rho}$ can be continued both to $\tilde{\rho} \to 0$ and $\tilde{\rho} \to \infty$, establishing corresponding crossover scaling solutions.

### 5.5. Gravitational Coleman–Weinberg Mechanism

For the quartic coupling the flow equation reads:

$$\partial_t \lambda = A\lambda + B\tilde{m}^4 + 2\tilde{\rho}\frac{\partial \lambda}{\partial \tilde{\rho}} + \frac{3g^4}{16\pi^2}\left(1 + g^2\tilde{\rho}\right)^{-3}, \tag{78}$$

with

$$B = \frac{\partial A}{\partial u} = \frac{5}{12\pi^2 w^2}\left(\frac{1}{(1-v)^3} + \frac{1}{8}\frac{1}{(1-v/4)^3}\right). \tag{79}$$

The scaling solution for $\lambda(\tilde{\rho})$ therefore obeys:

$$2\tilde{\rho}\frac{\partial \lambda}{\partial \tilde{\rho}} = -A\lambda - B\tilde{m}^4 - \frac{3g^4}{16\pi^2}\left(1 + g^2\tilde{\rho}\right)^{-3}. \tag{80}$$

We can interpret the running of the quartic coupling with $\tilde{\rho}$ as a type of gravitational Coleman–Weinberg mechanism. Starting from $\tilde{\rho} \to \infty$ with $\lambda(\tilde{\rho} \to \infty) = 0$ and lowering $\tilde{\rho}$, the quartic coupling first is negative. Indeed, with $\lambda \sim \tilde{\rho}^{-3}$, $\tilde{m}^{-4} \sim \tilde{\rho}^{-4}$ we can neglect the term $B\tilde{m}^4$ for large $\tilde{\rho}$ and employ $\lambda = -(\tilde{\rho}\,\partial_{\tilde{\rho}}\,\lambda)/3$, such that:

$$\lambda = -\frac{3}{16(A_\infty - 6)\pi^2 g^2\tilde{\rho}^3}. \tag{81}$$

This coincides with the $\tilde{\rho}$-derivative of Equation (77), as it should be. As $\tilde{\rho}$ decreases, $\lambda(\tilde{\rho})$ first becomes increasingly negative, such that $\tilde{m}^2$ increases to larger positive values. As $A(\tilde{\rho})$ decreases for decreasing $\tilde{\rho}$, the influence of the first positive term $-A\lambda$ in Equation (80) becomes less important and $\lambda(\tilde{\rho})$ starts to increase due to the other negative terms. Once $\lambda$ becomes positive, $\tilde{m}^2(\tilde{\rho})$ starts to decrease until it reaches zero at some local minimum of the potential. For a rough qualitative estimate we replace $A_\infty$ by $A(\tilde{\rho})$ in Equations (77) and (81). The minimum occurs in a region where $A(\tilde{\rho}_{\min}) < 6$, with positive $\lambda(\tilde{\rho}_{\min})$. This qualitative behavior is well visible by taking the $\tilde{\rho}$-derivative of $\tilde{m}^2(\tilde{\rho})$ in Figure 8 or the second $\tilde{\rho}$-derivative of $u(\tilde{\rho})$ in Figures 6 and 7. The upper curve in Figure 9 shows that the appearance of a minimum of $u(\tilde{\rho})$ is not the only possibility.

The range of minimum values $\tilde{\rho}_{\min}$ is restricted, as seen from the lowest curve in Figure 9 which corresponds to a boundary curve for this type of scaling solutions. The question arises if $\tilde{\rho}_{\min}$ can be arbitrarily small. For $\tilde{\rho}_{\min} \to 0_+$ one needs $\tilde{m}^2_{0,*} = \tilde{m}^2(\tilde{\rho} = 0) = 0$. From Equation (62) we conclude that this requires negative $\lambda_0$,

$$\lambda_0 = -c_g \bar{N}_V g^2. \tag{82}$$

At least for small $g^2$ and not too large $u^{(3)}_{0,*}$, there seems to be a discrepancy with Equation (64). More generally, it is not clear if all solutions shown in Figures 6–9 can be extended to $\tilde{\rho} \to 0$ once the effects of nonzero $\tilde{m}^2$ and $\lambda$ in Equation (A13) are included.

### 5.6. Yukawa Couplings

For non-zero Yukawa couplings and vanishing gauge couplings, the structure of the flow equations is very similar to the case of non-zero gauge couplings and zero Yukawa couplings. The key difference is a change of the overall sign, due to the fermionic statistics. As a consequence, a non-zero Yukawa coupling favors a minimum at the origin. The fermion fluctuations yield a positive contribution to $\tilde{m}^2_0$ for the scaling solution. In Appendix D we describe the effects of non-zero Yukawa couplings in more detail.

## 6. Nonminimal Gravitational Coupling

The effective Planck mass may depend on the scalar field due to a nonminimal coupling $\xi$,

$$\mathcal{L} = -\frac{1}{2}\sqrt{g}\,\xi\rho\,R\,, \tag{83}$$

with $R$ the curvature scalar and $\rho$ a suitable scalar bilinear, as $\rho = h^\dagger h$ for the Higgs doublet. Any non-zero $\xi$ has a strong influence on the flow equations and the behavior of the scaling solutions for large values of $\tilde{\rho}$. In this limit the term (83) dominates the effective Planck mass and therefore the effective strength of the gravitational interaction. We typically find that non-zero $\xi$ induces spontaneous symmetry breaking for the scaling solution, with a minimum at $\tilde{\rho}_0$ somewhat below one.

### 6.1. Flow Equation with Non-Minimal Gravitational Coupling

As a consequence of the term (83), one has a field-dependent shift in the squared Planck mass $M^2 \to M^2 + \xi\rho$, or an additional field dependence in the dimensionless quantity $w(\tilde{\rho})$,

$$w(\tilde{\rho}) = w_0 + \frac{\xi}{2}\tilde{\rho}\,. \tag{84}$$

Here $w_0$ is the dimensionless squared Planck mass discussed in the previous sections. We assume in this section that both $w_0$ and $\xi$ are given by $k$-independent fixed point values and take them as undetermined parameters (these quantities may also depend on a further scalar singlet field $\chi$). In Section 7 we will extend this setting by treating both the effective potential $u(\tilde{\rho})$ and the coefficient function of the curvature scalar $w(\tilde{\rho})$ as $k$-dependent "flowing functions".

The non-minimal coupling $\xi$ does not affect the contribution from fermions and gauge bosons. Its main effect is a modification of the contribution from the metric fluctuations by replacing in Equation (20):

$$v(\tilde{\rho}) = \frac{u(\tilde{\rho})}{w(\tilde{\rho})} = \frac{u(\tilde{\rho})}{w_0 + \xi\tilde{\rho}/2}\,. \tag{85}$$

The coupling $\xi$ further influences the mixing between the physical scalar fluctuations in the metric and other scalars. We neglect this mixing in the present paper such that $\xi$ does not change the flow contribution from scalar fields. Then the replacement (85) is the only modification for nonzero $\xi$. Similar to our treatment of $w$ before, we do not compute here the flow equation for $\xi$.

The $\tilde{\rho}$-dependence of $v(\tilde{\rho})$ obeys [7]:

$$\frac{\partial v}{\partial \tilde{\rho}} = \frac{\tilde{m}^2}{w} - \frac{\xi v}{2w}\,. \tag{86}$$

As a result, the flow equation for $\tilde{m}^2 = \partial u/\partial\tilde{\rho}$ receives an additional contribution:

$$\partial_t \tilde{m}^2 = 2\tilde{\rho}\frac{\partial \tilde{m}^2}{\partial \tilde{\rho}} + (A-2)\tilde{m}^2 - \frac{\xi A v}{2} + \dots \tag{87}$$

where $A$ is given by Equation (37) with $v(\tilde{\rho})$ and $w(\tilde{\rho})$. The dots denote contributions from scalars, fermions, and gauge bosons that are not modified for $\xi \neq 0$. As a consequence of the contribution $\sim \xi$ constant scaling solutions with $\tilde{m}^2(\tilde{\rho}) = 0$ are no longer possible.

In particular, one finds for $\tilde{m}_0^2 = \tilde{m}^2(\tilde{\rho} = 0)$:

$$\partial_t \tilde{m}_0^2 = (A_0 - 2)\,\tilde{m}_0^2 - \frac{\xi A_0 v_0}{2} - \frac{(N_S' + 2)\lambda_0}{32\pi^2 (1 + \tilde{m}_0^2)^2}$$

$$- \frac{3\bar{N}_V g^2}{32\pi^2} + \frac{\bar{N}_F y^2}{16\pi^2}. \tag{88}$$

where we have taken $N_S'$ scalars with $\mathcal{O}(N_S')$-symmetric potential and assumed that $\lambda\tilde{\rho}$ and $\tilde{\rho}\,\partial\lambda/\partial\tilde{\rho}$ remain finite for $\tilde{\rho} \to 0$, as well as $c_g = 1$, $c_f = 1$. For $\xi \neq 0$, a flat potential at the origin ($\tilde{m}_0^2 = \lambda_0 = 0$) is no longer a scaling solution even for vanishing gauge and Yukawa couplings $g^2 = y^2 = 0$. The issue of spontaneous symmetry breaking of the scaling solution is directly linked to the sign of $\tilde{m}_0^2$ for the scaling solution. Equation (88) shows that this sign depends on the relative size of the various couplings. For $A_0 < 2$, positive $\xi v_0$, $g^2$, and $\lambda_0$ favor spontaneous symmetry breaking ($\tilde{m}_0^2 < 0$), while for $A_0 > 2$ the same couplings favor a minimum of $u$ at the origin.

### 6.2. Scaling Solution

In Figure 10 we show a numerical scaling solution of Equation (26), with $w(\tilde{\rho})$ given by Equation (84) and $g^2 = y^2 = 0$. We take $\xi = 0.1$ and plot four different values of $w_0 = 0.038, 0.042, 0.05$, and $0.06$. The corresponding values of $A_0$ are $A_0 = 2.74, 1.64, 1.0$, and $0.68$, such that the highest curve corresponds to $A_0 > 2$. For all curves, $u(\tilde{\rho})$ reaches a constant for $\tilde{\rho} \to 0$ and increases as $w(\tilde{\rho}) \sim \xi\,\tilde{\rho}/2$ for $\tilde{\rho} \to \infty$. A minimum at $\tilde{\rho}_0 \neq 0$ is clearly visible. The precise value of the minimum depends on the initial condition for the first order differential equation that we choose for Figure 10 as $u(\tilde{\rho} = 1)$. We find indeed a whole family of scaling solutions that may be parameterized by $u(\tilde{\rho} = 1)$. As discussed in Appendix C, it is so far not known if all scaling solutions can consistently be continued to $\tilde{\rho} = 0$ once the contribution from scalar fluctuations is included.

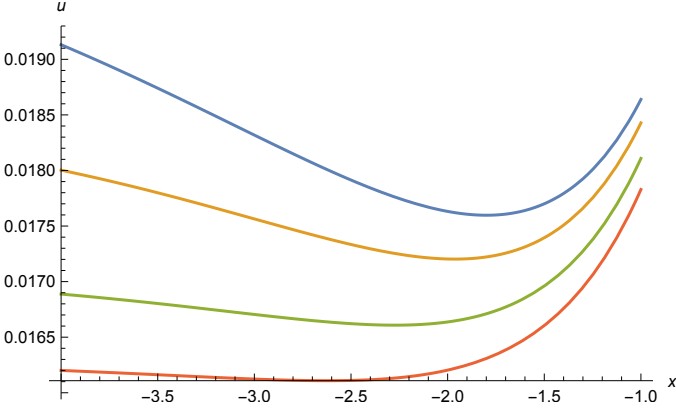

**Figure 10.** Effective potential $u$ in presence of a nonminimal coupling $\xi$ to gravity, in function of $x = \ln(\tilde{\rho})$. Parameters are $\xi = 0.1$, $N_{\text{eff}} = 10$. Different curves correspond to different values of $w_0 = 0.038, 0.042, 0.05$, and $0.06$, with corresponding $A_0$ given by $2.74, 1.64, 1.0$, and $0.68$, from top to bottom. We choose as initial condition $u(x = 0) = 0.03$. A local minimum occurs near $x = -2$.

We plot in Figure 11 the value $v(\tilde{\rho}) = u(\tilde{\rho})/w(\tilde{\rho})$ for the same set of parameters. All curves approach for $\tilde{\rho} \to \infty$ the asymptotic behavior $v(\tilde{\rho} \to \infty) \to 1$. This is consistent with the graviton barrier discussed in ref. [6,16]. Correspondingly, the increase of $u(\tilde{\rho})$ for $\tilde{\rho} \to \infty$ is bounded to be linear in $\tilde{\rho}$. This can be seen in Figure 12 for $\tilde{m}^2(\tilde{\rho}) = \partial_{\tilde{\rho}} u(\tilde{\rho})$. For large $\tilde{\rho}$, one finds the asymptotic value $\tilde{m}^2(\tilde{\rho} \to \infty) = \xi/2$. In the next two sections we will discuss an alternative asymptotic behavior for $\tilde{\rho} \to \infty$ with $w \to \xi\tilde{\rho}/2$ and $u \to u_\infty$.

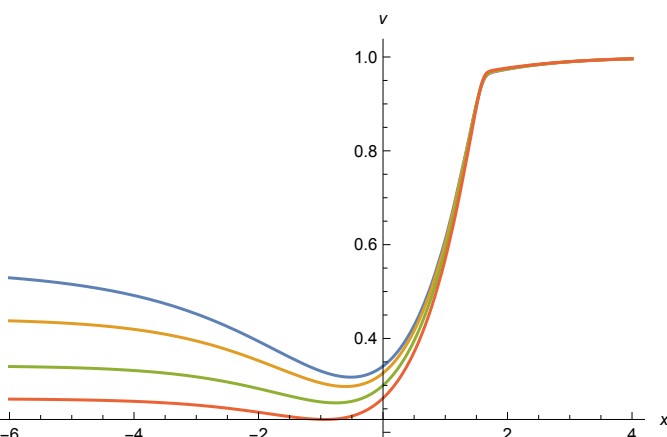

**Figure 11.** Ratio $v = u/w$ as function of $x = \ln(\tilde{\rho})$ for $\xi = 0.1$. Other parameters are as in Figure 10, with $w_0$ between 0.038 and 0.06 from top to bottom. One observes the common approach to the asymptotic value $v = 1$ for $x \to \infty$.

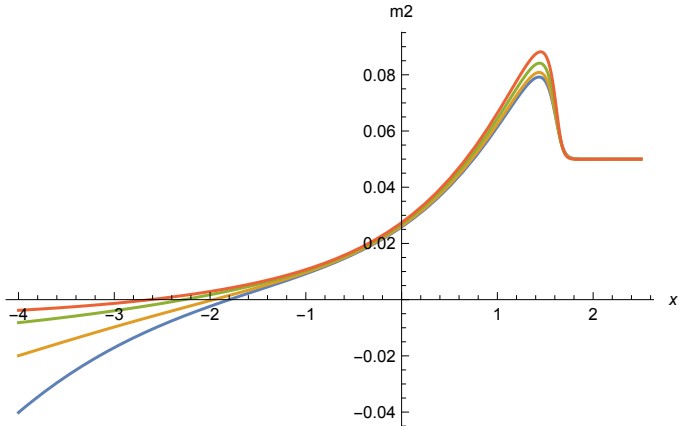

**Figure 12.** Mass term $\tilde{m}^2(x)$ with parameters as in Figure 10, with $w_0$ between 0.038 and 0.06 from bottom to top on the left side of the figure.

The mass term found in Figure 12 remains small for all $w_0$ and $\xi$ chosen, and for the whole range in $\tilde{\rho}$. This justifies the approximation of massless scalar field fluctuations. A whole family of scaling solutions seems to exist.

A non-minimal scalar-curvature coupling $\xi$ strongly influences the asymptotic behavior for $\tilde{\rho} \to \infty$. As a consequence, the discussion in Sections 3–6 can be relevant only for a restricted range of $\tilde{\rho}$, i.e.,

$$\tilde{\rho} < \frac{2w_0}{\xi}. \tag{89}$$

For small enough $\xi$ this range may be rather large. What we have called the asymptotic behavior in Sections 3–6 becomes in the presence of a small non-minimal coupling the range of large $\tilde{\rho}$ that still obeys Equation (89). In this range only, small corrections to the results of Sections 3–6 are expected from the non-minimal coupling $\xi$.

## 7. Scaling Solution with Field-Dependent Planck Mass

In this section we extend the truncation of the effective average action to two $k$-dependent functions $u(\tilde{\rho})$ and $w(\tilde{\rho})$. For vanishing gauge and Yukawa couplings, we recover the constant scaling solution which corresponds to the extension of the Reuter fixed point [2,62–64] of pure gravity to the presence of matter [4]. Our investigation is based on the gauge invariant flow equation [5] and the flow equations for $u(\tilde{\rho})$ and $w(\tilde{\rho})$ derived in [8]. In particular, we compute the flow equation for the non-minimal coupling $\xi$. We

discuss the vicinity of the constant scaling solutions as well as the behavior for large $\tilde{\rho}$ and possible crossover scaling solutions. We include the case of nonzero gauge and Yukawa couplings.

### 7.1. Flowing Planck Mass

So far we have made an ansatz for the function $F(\rho)$, or the associated dimensionless quantity $w(\tilde{\rho}) = F/(2k^2)$. In this section we investigate the combined system of flow equations for $u(\tilde{\rho})$ and $w(\tilde{\rho})$. For the truncation (9) with $Z_a = 1$, the flow equations have been computed from the gauge invariant flow equation in [8],

$$\partial_t u = 2\tilde{\rho}\,\partial_{\tilde{\rho}} u - 4u + \frac{5}{24\pi^2}\left(1 - \frac{u}{w}\right)^{-1} + \frac{\tilde{\mathcal{N}}_U}{32\pi^2}, \tag{90}$$

and

$$\partial_t w = 2\tilde{\rho}\,\partial_{\tilde{\rho}} w - 2w + \frac{25}{64\pi^2}\left(1 - \frac{u}{w}\right)^{-1} + \frac{\tilde{\mathcal{N}}_M}{96\pi^2}, \tag{91}$$

with

$$\tilde{\mathcal{N}}_U = N_S + 2N_V - 2N_F - \frac{8}{3},$$

$$\tilde{\mathcal{N}}_M = -N_S + 4N_V - N_F + \frac{43}{6} - \frac{3}{2}\frac{\tilde{\xi}}{N_\xi}N_\xi. \tag{92}$$

The new flow equation for $w$ involves the contributions of matter fluctuations $\sim N_S$, $N_V$, or $N_F$. There is also a contribution proportional to $\tilde{\xi}$, i.e. the non-minimal scalar coupling to the curvature scalar. The other contributions arise from the graviton fluctuations, with fluctuations of the physical scalar in the metric, gauge modes in the metric, and ghosts summarized in the term $43/6$ in $\tilde{\mathcal{N}}_M$. Equations (90)–(92) employ the Litim cutoff and simplify the subdominant sector of scalar metric fluctuations by neglecting the mixing with other scalar fields and omitting a factor $(1 - u/(4w))^{-1}$ in the scalar metric contribution.

For vanishing gauge and Yukawa couplings and massless fields, one has constant $N_S$, $N_V$, and $N_F$, which count the number of real scalars, gauge bosons, and Weyl fermions, respectively. For $g$ or $y$ different from zero, one has the effective $\tilde{\rho}$-dependent particle numbers that obtain by multiplication with "threshold functions" $(1 + \tilde{m}_i^2)^{-1}$. The field-dependent mass terms $\tilde{m}_i^2$ are of the type $\tilde{m}^2 = g^2\tilde{\rho}$ for gauge bosons, $\tilde{m}^2 = y^2\tilde{\rho}$ for fermions, and $\tilde{m}^2 = u' + 2\tilde{\rho}u''$ or $\tilde{m}^2 = u'$ for scalars in the radial or Goldstone directions. Different species may have different effective couplings. In practice, we will use the following:

$$2N_V = \left(\frac{3}{1 + g^2\tilde{\rho}} - 1\right)\bar{N}_V,$$

$$N_F = \frac{\bar{N}_F}{1 + y^2\tilde{\rho}}, \quad N_S = \frac{\bar{N}_S}{1 + u'}, \quad N_\xi = \bar{N}_S, \tag{93}$$

with constant particle numbers $\bar{N}_V$, $\bar{N}_F$, and $\bar{N}_S$. For the gauge bosons the contribution of the three physical transversal gauge bosons and the measure contribution (longitudinal gauge bosons and Faddeev–Popov determinant or ghosts) have the opposite sign.

The function $\tilde{\xi}$ is defined by:

$$\tilde{\xi} = \frac{\partial^2 F}{\partial\varphi^2}, \tag{94}$$

and may again be different for different scalar fields. It generalizes the nonminimal gravitational coupling $\xi$ of Section 6, with $N_{\xi}$ the number of states with this coupling. For $\tilde{\rho} = \varphi^2/(2k^2)$ it reads:

$$\tilde{\xi} = \xi + 4\,w_2\tilde{\rho}\,, \quad \xi = 2\frac{\partial w}{\partial \tilde{\rho}}\,, \quad w_2 = \frac{\partial^2 w}{\partial \tilde{\rho}^2}\,. \tag{95}$$

In case of $O(N)$-symmetry, $\tilde{\rho} = \varphi_a\varphi_a/(2k^2)$, there are in addition $N-1$-contributions from the Goldstone directions,

$$\tilde{\xi} = N\xi + 4w_2\tilde{\rho}, \tag{96}$$

such that for the Higgs-doublet with $N = 4$ one has:

$$N_{\xi}\tilde{\xi} = 4(\xi + \tilde{\rho}w_2). \tag{97}$$

Equation (95) defines the non-minimal coupling $\xi(\tilde{\rho})$ as a field-dependent function, given by the first $\tilde{\rho}$-derivative of $w(\tilde{\rho})$. As we have mentioned in the introduction, $\xi$ can take different values for large $\tilde{\rho}$ and for $\tilde{\rho} \to 0$. The flow equation for $\xi(\tilde{\rho})$ obtains by taking a $\tilde{\rho}$-derivative of the flow Equation (91) for $w(\tilde{\rho})$.

The system of differential Equations (90)–(92) is closed for fixed gauge couplings $g$ and Yukawa couplings $y$. With our approximation, it has a rather simple structure. It can be considered as the central equation for the investigations of the present paper.

### 7.2. Scaling Solutions

For a given model the system of Equations (90) and (91) is closed. We are interested in the scaling solution with $\partial_t u = \partial_t w = 0$. We discuss here a single representation of scalars with $\bar{N}_S$ components, coupling with a unique gauge coupling $g$ to $\bar{N}_V$ vector bosons and a unique Yukawa coupling $y$ to $\bar{N}_F$ fermions. We also assume $N_{\xi} = \bar{N}_S$ and neglect $u''$ and $w''$ on the r.h.s. of the flow equations. We omit the bars on $\bar{N}_S$, $\bar{N}_V$, and $\bar{N}_F$ in the following for the sake of simplicity of the formulae.

In this approximation, the two differential equations for the scaling solution can be inferred directly from Equations (90) and (91) by setting $\partial_t u = 0, \partial_t w = 0$. They read:

$$2\tilde{\rho}\,\partial_{\tilde{\rho}}u = 4\,(u - c_U)\,, \quad 2\tilde{\rho}\,\partial_{\tilde{\rho}}w = 2\,(w - c_M)\,, \tag{98}$$

with

$$c_U = \frac{5}{96\pi^2}\left(1 - \frac{u}{w}\right)^{-1}$$
$$+ \frac{1}{128\pi^2}\left[\frac{N_S}{1+u'} + \left(\frac{3}{1+g^2\tilde{\rho}} - 1\right)N_V - \frac{2N_F}{1+y^2\tilde{\rho}} - \frac{8}{3}\right], \tag{99}$$

and

$$c_M = \frac{25}{128\pi^2}\left(1 - \frac{u}{w}\right)^{-1}$$
$$+ \frac{1}{192\pi^2}\left[-N_S\left(\frac{1}{1+u'} + \frac{3w'}{(1+u')^2}\right)\right.$$
$$\left. + 2N_V\left(\frac{3}{1+g^2\tilde{\rho}} - 1\right) - \frac{N_F}{1+y^2\tilde{\rho}} + \frac{43}{6}\right]. \tag{100}$$

In the following, we discuss the solutions of these central equations.

We are interested in solutions for which $u'$ and $w'$ remain finite for $\tilde{\rho} \to 0$. This condition relates $u(0)$, $w(0)$ to $u'(0)$, $w'(0)$ according to:

$$u(0) = c_U(0) = \frac{5}{96\pi^2 (1 - v_0)}$$
$$+ \frac{1}{128\pi^2} \left[ \frac{N_S}{1 + u'(0)} + 2N_V - 2N_F - \frac{8}{3} \right], \tag{101}$$

and

$$w(0) = c_M(0) = \frac{25}{128\pi^2 (1 - v_0)}$$
$$+ \frac{1}{192\pi^2} \left[ -N_S \left( \frac{1}{1 + u'(0)} + \frac{3w'(0)}{(1 + u'(0))^2} \right) \right.$$
$$\left. + 4N_V - N_F + \frac{43}{6} \right], \tag{102}$$

with

$$v_0 = \frac{u(0)}{w(0)}. \tag{103}$$

Equations (98)–(100) are a system of non-linear first order differential equations for two functions $u(\tilde{\rho})$ and $w(\tilde{\rho})$. The general local solution has therefore two free integration constants that one could associate with $u'(0)$ and $w'(0)$. The question that arises is of which one of the local solutions can extend to global solutions valid for the whole range of $\tilde{\rho}$. In practice, we will often choose the free integration constants in a different way.

### 7.3. Constant Scaling Solution

For vanishing gauge and Yukawa couplings, $g = y = 0$, the system of differential Equations (99) and (100) admits a constant scaling solution, $u'(\tilde{\rho}) = 0$, $w'(\tilde{\rho}) = 0$, which has been discussed extensively in [8]. It is given by:

$$v_* = 1 - \frac{1}{4\tilde{\mathcal{N}}_M} \left( b + \sqrt{b^2 + 440\,\tilde{\mathcal{N}}_M} \right),$$
$$b = 2\tilde{\mathcal{N}}_M - 3\,\mathcal{N}_U - 75\,, \tag{104}$$
$$\tilde{\mathcal{N}}_{U,*} = N_S + 2N_V - 2N_F - \frac{8}{3}\,,$$
$$\tilde{\mathcal{N}}_{M,*} = -N_S + 4N_V - N_F + \frac{43}{6}\,. \tag{105}$$

The corresponding dimensionless potential and squared Planck mass are independent of $\tilde{\rho}$,

$$u_* = \frac{1}{128\pi^2} \left( \tilde{\mathcal{N}}_{U,*} + \frac{20}{3(1 - v_*)} \right),$$
$$w_* = \frac{1}{192\pi^2} \left( \tilde{\mathcal{N}}_{M,*} + \frac{75}{2(1 - v_*)} \right). \tag{106}$$

A second constant scaling solution exists only in a small regime of $\tilde{\mathcal{N}}_U$ and $\tilde{\mathcal{N}}_M$ and will not be discussed here explicitly.

For the major part of the model space $(\tilde{\mathcal{N}}_U, \tilde{\mathcal{N}}_M)$, we conclude that out of the two constant scaling solutions for $u$ for a fixed constant $w = w_0$ that we have discussed in Section 3.2, only one is compatible with a simultaneous scaling solution for $w$. It corresponds to the extended Reuter fixed point [4]. As a consequence, the crossover scaling

solution for $\tilde{u}_*(\tilde{\rho})$ discussed in Section 3.3 is not a valid scaling solution for the combined system of flow equations for $u(\tilde{\rho})$ and $w(\tilde{\rho})$. It could only be an approximation for a region of a scaling solution in which $w_*(\tilde{\rho})$ does not change much with $\tilde{\rho}$. Generic crossover scaling solutions can still exist for ranges in the field content and parameters for which two different constant scaling solutions exist. We learn even very rough features of scaling solutions as the number of possible solutions can depend on the truncation in an important way.

### 7.4. Scaling Solutions Close to a Constant Scaling Solution

We next address the question of if the constant scaling solution is isolated or part of a continuous family of scaling solutions. For this purpose we first discuss the possible scaling solutions in the vicinity of the constant scaling solution. These neighboring scaling solutions may only remain in the vicinity of the constant scaling solution in the range of small $\tilde{\rho}$. For any non-zero $\xi$, one expects a strong deviation for $\tilde{\rho} \to \infty$.

We perform in Appendix E, a detailed investigation of the system of linear differential equations for small deviations from the constant scaling solution. The overall picture emerging is that scaling solutions that are close to the constant scaling solution for $\tilde{\rho} \to 0$ diverge away from the constant scaling solution as $\tilde{\rho}$ increases. In the other direction, a large class of potential scaling solutions approaches the vicinity of the constant scaling solution for $\tilde{\rho} \to 0$. In the absence of gauge and Yukawa couplings, we find a problematic transition region where the linear approximation leads to strong variations, casting doubt on the existence of global scaling solutions with these properties.

One possibility to avoiding such strong variations is that valid scaling solutions reach the transition region in a range where they still differ sufficiently from the constant scaling solution such that a linear treatment is not valid. We will observe below that the problem of strong variations seems not to be present for non-zero gauge or Yukawa couplings.

### 7.5. Asymptotic Scaling for Large Fields and the Cosmon Potential

Interesting candidate scaling solutions reach for large $\tilde{\rho}$ a constant value for $u$, while $w$ is dominated by a linear increase with $\tilde{\rho}$,

$$u(\tilde{\rho} \to \infty) = u_\infty, \quad w(\tilde{\rho} \to \infty) = \frac{\xi_\infty}{2}\tilde{\rho}. \tag{107}$$

Such behavior leads to cosmologies with a light scalar field—the cosmon [65]—that could account for dynamical dark energy or quintessence [15]. The quantity relevant for cosmology is the dimensionless ratio:

$$\frac{u}{4w^2} = \frac{U(\rho)}{M^4(\rho)} = \frac{u_\infty}{\xi_\infty^2 \tilde{\rho}^2} = \frac{u_\infty k^4}{\xi_\infty^2 \rho^2}. \tag{108}$$

It is the same in all metric frames and related to the scalar potential in the Einstein frame $V_E$ by:

$$V_E = \frac{u M_E^4}{4w^2} = \frac{u_\infty k^4 M_E^4}{\xi_\infty^2 \rho^2}, \tag{109}$$

with $M_E$ the fixed Planck mass in the Einstein frame. The potential $V_E$ constitutes the dark energy density in the universe, supplemented by a smaller contribution from the kinetic energy of the scalar field. It decreases towards zero for $\rho \to \infty$. "Runaway cosmological solutions" lead indeed to an unbounded increase of $\rho$ for increasing time. A potential of the type (107) therefore solves the cosmological constant problem dynamically by dark energy decreasing to zero in the infinite future. Details of the translation to cosmology and corrections for realistic cosmologies can be found in [11]. We mention that the factor $k^4$ is absorbed by a proper normalization of the kinetic term for the scalar field, which turns $V_E$ into an exponentially decreasing potential.

Scaling solutions with asymptotic behavior (107) have already been investigated in the context of "dilaton quantum gravity" [17,18]. Interesting candidate scaling solutions have been found numerically. We employ here a simpler system of flow equations which may help towards partial analytical understanding of the main features of possible scaling solutions. We also extend the scope, including additional fields and possible non-zero gauge and Yukawa couplings.

For scaling solutions with an asymptotic behavior (107) we expand:

$$u(\tilde{\rho}) = u_\infty + \frac{u^{(1)}}{\tilde{\rho}} + \frac{u^{(2)}}{\tilde{\rho}^2} + \dots$$

$$w(\tilde{\rho}) = \frac{1}{2}\xi_\infty\tilde{\rho} + w^{(0)} + \frac{w^{(1)}}{\tilde{\rho}} + \frac{w^{(2)}}{\tilde{\rho}^2} + \dots, \tag{110}$$

where dots denote higher-order terms in an expansion in $\tilde{\rho}^{-1}$. This expansion has been investigated to much higher orders for dilaton quantum gravity [17,18]. We also expand:

$$c_U = c_{U,\infty} + \frac{c_U^{(1)}}{\tilde{\rho}} + \frac{c_U^{(2)}}{\tilde{\rho}^2} + \dots$$

$$c_M = c_{M,\infty} + \frac{c_M^{(1)}}{\tilde{\rho}} + \frac{c_M^{(2)}}{\tilde{\rho}^2} + \dots, \tag{111}$$

such that the differential Equation (98) for the scaling solution takes the form:

$$2\left(u_\infty - c_{U,\infty}\right) + \frac{3u^{(1)} - 2c_U^{(1)}}{\tilde{\rho}} + \frac{4u^{(2)} - 2c_U^{(2)}}{\tilde{\rho}^2} + \dots = 0, \tag{112}$$

and

$$w^{(0)} - c_{M,\infty} + \frac{2w^{(1)} - c_M^{(1)}}{\tilde{\rho}} + \frac{2w^{(2)} - c_M^{(2)}}{\tilde{\rho}^2} + \dots = 0. \tag{113}$$

The solution expresses the coefficients $u^{(i)}$, $w^{(i)}$ in terms of $c_U^{(i)}$, $c_M^{(i)}$, with $\xi_\infty$ a free integration constant. We therefore have a one-parameter family of asymptotic scaling solutions, parameterized by $\xi_\infty$.

For $g = y = 0$ one has:

$$c_U = \frac{5}{96\pi^2\left(1 - v\right)} + \frac{1}{128\pi^2}\left[\tilde{\mathcal{N}}_U + N_S\left(\frac{1}{1 + u'} - 1\right)\right],$$

$$c_M = \frac{25}{128\pi^2\left(1 - v\right)}$$

$$+ \frac{1}{192\pi^2}\left[\tilde{\mathcal{N}}_{M,*} - N_S\left(\frac{1}{1 + u'} - 1\right) - \frac{3N_S\,w'}{(1 + u')^2}\right]. \tag{114}$$

With,

$$(1 - v)^{-1} = 1 + \frac{2u_\infty}{\xi_\infty\tilde{\rho}} + \frac{2}{\xi_\infty\tilde{\rho}^2}\left(u^{(1)} + \frac{2u_\infty}{\xi_\infty}(u_\infty - w^{(0)})\right), \tag{115}$$

one obtains:

$$u_\infty = c_{U,\infty} = \frac{5}{96\pi^2} + \frac{\tilde{\mathcal{N}}_{U,*}}{128\pi^2},$$

$$w^{(0)} = c_{M,\infty} = \frac{25}{128\pi^2} + \frac{1}{192\pi^2}\left(\tilde{\mathcal{N}}_{M,*} - \frac{3N_S\,\tilde{\xi}_\infty}{2}\right). \tag{116}$$

In the next order one finds:

$$u^{(1)} = \frac{2}{3}c_U^{(1)} = \frac{5\,u_\infty}{72\pi^2\,\tilde{\xi}_\infty} \tag{117}$$

and

$$w^{(1)} = \frac{1}{2}c_M^{(1)} = \frac{25\,u_\infty}{128\pi^2\,\tilde{\xi}_\infty}. \tag{118}$$

Continuation to the terms $\sim \tilde{\rho}^{-2}$ yields:

$$u^{(2)} = \frac{1}{2}c_U^{(2)} = \frac{u^{(1)}}{768\pi^2}\left(3N_S + \frac{40}{\tilde{\xi}_\infty}\right)$$

$$+ \frac{5\,u_\infty}{48\pi^2\,\tilde{\xi}_\infty^2}(u_\infty - w^{(0)}), \tag{119}$$

and

$$w^{(2)} = \frac{1}{3}c_M^{(2)} = \frac{u^{(1)}}{192\pi^2}\left(\frac{25}{\tilde{\xi}_\infty} - \frac{N_S}{3} - N_S\,\tilde{\xi}_\infty\right)$$

$$+ \frac{25\,u_\infty}{96\pi^2\,\tilde{\xi}_\infty^2}(u_\infty - w^{(0)}) + \frac{N_S\,w^{(1)}}{192\pi^2}. \tag{120}$$

This can be continued to higher orders [17,18] and yields an accurate description of possible scaling solutions with behavior (107). For a given particle content the family of these candidate scaling solutions is parameterized by the free constant $\tilde{\xi}_\infty$.

The question arises as to which of these asymptotic solutions correspond to true scaling solutions for the whole range of $\tilde{\rho}$. For a numerical investigation we employ initial conditions for large $\tilde{\rho}$, $\tilde{\rho} = \tilde{\rho}_{as}$, say $\tilde{\rho}_{as} = 1000$, or even larger. The initial conditions for $u(\tilde{\rho}_{as})$, $w(\tilde{\rho}_{as})$ are taken from the asymptotic solution (110), with $\tilde{\xi}_\infty$ a free parameter. For these large values of $\tilde{\rho}_{as}$, the asymptotic solution (110) obeys the full differential Equations (98) and (99) with an accuracy of $10^{-13}$ or better. We then solve the differential Equations (98) and (99) numerically towards smaller $\tilde{\rho}$ and ask for which values of $\tilde{\xi}_\infty$ the solution extends to $\tilde{\rho} \to 0$. For $g^2 = y^2 = 0$ and $N_S = 1$, $N_V = 0$, and $N_F = 0$ we plot the solution for different values of $\tilde{\xi}_\infty$ in Figures 13 and 14. This could be a typical setting for a scalar singlet associated to the inflation of the cosmon, the scalar field mediating dynamical dark energy as $\chi$ moves to infinity.

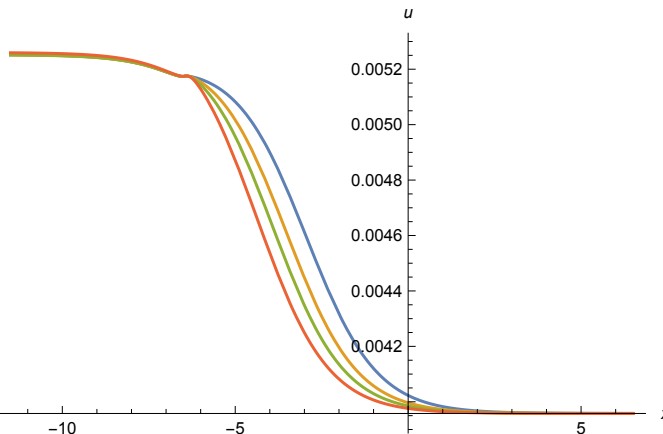

**Figure 13.** Effective potential $u$ as function of $x = \ln \tilde{\rho}$. We display curves for four different values $\xi_\infty = 0.405$ (blue), $\xi_\infty = 0.7$ (orange), $\xi_\infty = 1.0$ (green), and $\xi_\infty = 1.5$ (red), from top to bottom. Initial values as set at $\tilde{\rho}_{as} = 1000$ and the particle content is given by $N_S = 1$, $N_V = N_F = 0$.

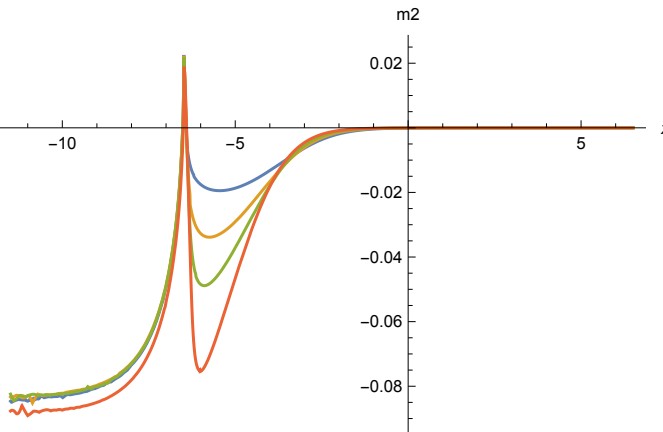

**Figure 14.** Mass term $u'$ as function of $x = \ln \tilde{\rho}$. Parameters and color coding is the same as for Figures 13 and 15. We observe strong variations in the transition region of $x$ between $-7$ and $-5$. The negative $u'(\rho \to 0)$ corresponds to a maximum of the potential at the origin. The small local minimum of $u$ at $x \approx -6.5$ may be an artifact of the truncation.

In Figure 13 we show $u(x)$ for $\xi_\infty = 0.405$, 0.7, 1.0, and 1.5. For smaller or larger $\xi_\infty$ outside the range of the plotted values, the solutions typically diverge within the interval of $x = \ln \tilde{\rho}$ shown. Only the solutions for $\xi_\infty$ within the restricted interval are candidates for valid scaling solutions. Compared to [17,18], this seems to restrict the parameter range for possible scaling solutions. In Figure 15 we display $w'(x)$ for the same values of $\xi_\infty$. One observes a switch from positive $w = \xi_\infty/2$ for large $\tilde{\rho}$ to negative $w'$ for $\tilde{\rho} \to 0$. The mass term $\tilde{m}^2 = u'$ is displayed in Figure 14. While it remains small everywhere, including the transition region, it shows substantial variation in the transition region. It remains open if this variation is damped by including the neglected terms $\sim u''$, $w''$ in the flow equation, or not.

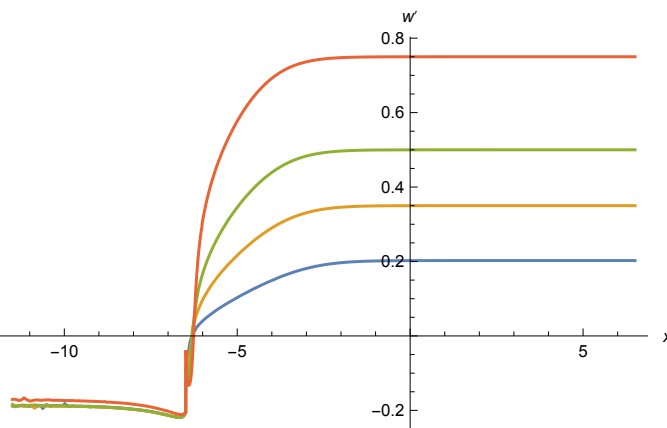

**Figure 15.** Nonminimal scalar coupling $w'$ as function of $x = \ln \tilde{\rho}$. Parameters and color coding are the same as for Figure 13. The value $\xi_\infty = 2\,w'(x \to \infty)$ can be read off directly.

In view of this open question it is not yet possible to decide if the asymptotic solutions can be extended to $\tilde{\rho} = 0$ or not. Better numerical precision, and possibly the inclusion of non-zero $u''$, $w''$ in the flow equations would be needed to decide if all parameters in the range of $\xi_\infty$ between 0.405 and 1.5 correspond to global scaling solutions, if only particularly tuned solutions can make a smooth transition, or if none of the candidate scaling solutions is globally viable. We will next see that the transition region is strongly influenced by the particle content and the possible presence of gauge or Yukawa couplings. Some of the examples below will show a much smoother behavior in the transition region.

### 7.6. Scaling Solutions with Gauge and Yukawa Couplings

The cosmon has to be a singlet with respect to the $SU(3) \times SU(2) \times U(1)$-gauge symmetry of the standard model. A very light non-singlet scalar would have been found by present experiments. It may, nevertheless, belong to a non-trivial representation of some grand unified gauge group, as the 24 of $SU(5)$ or the 45 or 54 of $SO(10)$. Its non-zero cosmological value $\chi$ would account for the spontaneous breaking of the GUT-gauge symmetry. This is phenomenologically without problems since scale symmetry implies that the Fermi scale and the confinement scale of QCD (quantum chromodynamics) are also proportional to $\chi$ [11,15]. The observable ratios of scales would remain invariant even for cosmologies where $\chi$ increases with $t$.

The important new ingredient for the scaling solutions in this type of scenario are the gauge couplings of $\chi$ to the heavy gauge bosons of the grand unified gauge symmetry. Since these gauge bosons have masses $\sim g\chi$, they are effectively massless at scales $k \gg g\chi$, corresponding to small values of $\tilde{\rho}$. The massive gauge bosons decouple for $k \ll g\chi$, or large values of $\tilde{\rho}$. Nevertheless, their presence will influence the non-leading terms in the expansions (110) and (111). Non-zero gauge couplings play an important role for the smoothness of the transition region.

With a standard normalization of the kinetic term for $\chi$, the scale of grand unification $M_{\mathrm{GUT}}$ is set by the mass of the heavy gauge bosons, while the effective Planck mass in the range of large $\xi$ is given by:

$$M_{\mathrm{GUT}}^2 = g^2 \rho, \quad M^2 = \xi_\infty \rho, \quad \frac{M_{\mathrm{GUT}}}{M} = \frac{g}{\sqrt{\xi_\infty}}. \tag{121}$$

Small values of $M_{\mathrm{GUT}}/M$ correspond to large values of $\xi_\infty$. We may also have a situation where $\chi$ is dominantly a singlet with respect to the GUT-symmetry. The scalar field responsible for GUT-symmetry breaking may take a smaller value $\chi_{\mathrm{GUT}}$. Scale symmetry requires $\chi_{\mathrm{GUT}}/\chi$ to be a constant $r_{\mathrm{GUT}}$. With:

$$M_{\mathrm{GUT}}^2 = g^2 \rho_{\mathrm{GUT}} = g^2 r_{\mathrm{GUT}}^2 \rho, \tag{122}$$

one finds an effective gauge coupling of the singlet $\chi$ given by:

$$g_{\text{eff}} = r_{\text{GUT}} g. \tag{123}$$

Employing $g_{\text{eff}}$, our discussion of non-zero gauge couplings also applies to the case of a scalar singlet with respect to the GUT-symmetry. The ratio $r_{\text{GUT}}$ can suppress the coefficient $c_g$ in Equation (57). Equation (121) remains valid if $g$ is replaced by $g_{\text{eff}}$. Even small values of $\xi_\infty$ are compatible with a small ratio $M_{\text{GUT}}/M$.

Nonvanishing gauge couplings change the character of the scaling solution. We discuss here the approximation that the gauge coupling $g$ is independent of $\tilde{\rho}$, and make the simplification that all $N_V$ vector bosons have the same mass term $g^2 \tilde{\rho}$. Constant scaling solutions no longer exist for the whole range of $\tilde{\rho}$ if $g^2 > 0$. This is connected to the simple property that the effective number of gauge bosons is not the same for $\tilde{\rho} \to 0$ and $\tilde{\rho} \to \infty$. For small $\tilde{\rho}$, $g^2 \tilde{\rho} \ll 1$, the effective number of gauge bosons is $N_V$. On the other hand, for large $\tilde{\rho}$, the mass term suppresses effectively the number of gauge bosons. Only the gauge modes are massless, replacing effectively $N_V$ by $N_{V,\infty} = -N_V/2$. As a consequence, the effective numbers $\tilde{\mathcal{N}}_U$ and $\tilde{\mathcal{N}}_M$ for $\tilde{\rho} \to 0$ are replaced for $\tilde{\rho} \to \infty$ by:

$$\tilde{\mathcal{N}}_{U,\infty} = N_S - N_V - 2N_F - \frac{8}{3},$$
$$\tilde{\mathcal{N}}_{M,\infty} = -N_S - 2N_V - N_F + \frac{43}{6} - \frac{3}{2}\frac{\tilde{\xi}}{2} N_{\tilde{\xi}}. \tag{124}$$

One may still have an almost flat potential for $\tilde{\rho} \to \infty$ as well as for $\tilde{\rho} \to 0$. The values of the flat potentials in the two limits will be different, however. Scaling solutions have then to describe a crossover between the two flat solutions, similar to Section 5, see Figures 6 and 9.

The situation is similar for nonzero Yukawa couplings. For $\tilde{\rho} \to \infty$ the effective number of fermions reduces to zero if we assume that all fermions have nonvanishing Yukawa couplings. We will consider a setting where all fermions have the same $\tilde{\rho}$-independent Yukawa coupling $y$. Taking further $N_{\tilde{\xi}} = N_S$, the effective particle numbers become for $\tilde{\rho} \to \infty$:

$$\tilde{\mathcal{N}}_{U,\infty} = N_S - N_V - \frac{8}{3},$$
$$\tilde{\mathcal{N}}_{M,\infty} = -N_S - 2N_V + \frac{43}{6} - \frac{3}{2}\frac{\xi_\infty}{2} N_S. \tag{125}$$

We may again explore the asymptotic scaling solutions of the type (110). The coefficients $u_\infty$, $w^{(0)}$ are now given by:

$$u_\infty = c_{U,\infty} = \frac{5}{96\pi^2} + \frac{\tilde{\mathcal{N}}_{U,\infty}}{128\pi^2},$$
$$w^{(0)} = c_{M,\infty} = \frac{25}{128\pi^2} + \frac{1}{192\pi^2}\tilde{\mathcal{N}}_{M,\infty}. \tag{126}$$

For the coefficients of the terms $\sim \tilde{\rho}^{-1}$ one obtains:

$$u^{(1)} = \frac{2}{3}c_U^{(1)} = \frac{5\,u_\infty}{72\pi^2\,\xi_\infty} + \frac{1}{64\pi^2}\left(\frac{N_V}{g^2} - \frac{2\,N_F}{3y^2}\right),$$
$$w^{(1)} = \frac{1}{2}c_M^{(1)} = \frac{25\,u_\infty}{128\pi^2\,\xi_\infty} + \frac{1}{384\pi^2}\left(\frac{6\,N_V}{g^2} - \frac{N_F}{y^2}\right). \tag{127}$$

Finally, the coefficients $u^{(2)}$, $w^{(2)}$ acquire additional contributions as well,

$$u^{(2)} = u_0^{(2)} + \Delta u^{(2)}, \quad w^{(2)} = w_0^{(2)} + \Delta w^{(2)}, \tag{128}$$

where $u^{(2)}$ and $w_0^{(2)}$ are given by Equations (119) and (120), respectively, and:

$$\Delta u^{(2)} = \frac{1}{2}\Delta c_U^{(2)} = -\frac{1}{256\pi^2}\left(\frac{3\,N_V}{g^4} - \frac{2\,N_F}{g^4}\right),$$

$$\Delta w^{(2)} = \frac{1}{3}\Delta c_M^{(2)} = -\frac{1}{192\pi^2}\left(\frac{2\,N_V}{g^4} - \frac{N_F}{3y^2}\right). \tag{129}$$

In Figures 16 and 17 we plot the scaling solutions that connect to the asymptotic scaling solutions for constant $g^2/4\pi = y^2/4\pi = 1/40$. We choose the particle content of the standard model, $N_S = 4$, $N_V = 12$, and $N_F = 45$ and set the asymptotic initial conditions at $\tilde{\rho}_{as} = 5000$. We choose two values $\xi_\infty = 0.1$ and $1.0$ and compare the solutions with the corresponding solutions for $g^2 = y^2 = 0$. For $\xi_\infty \gtrsim 1.5$ ($g^2 > 0$) or $\xi_\infty \gtrsim 2$ ($g^2 = 0$), the coefficient $w$ turns negative inside the interval of $x$ shown. These solutions are not acceptable scaling solutions, such that the allowed range of scaling solutions does not admit large scalar-curvature couplings $\xi_\infty$.

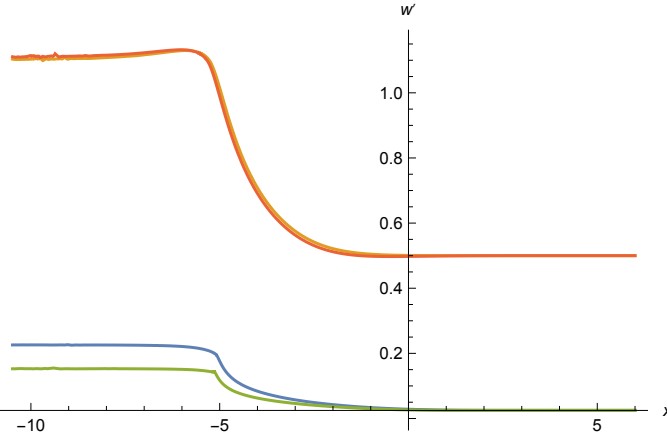

**Figure 16.** Nonminimal scalar coupling $w'(\tilde{\rho})$ as function of $x = \ln \tilde{\rho}$. Parameters and color coding are the same as for Figures 17 and 18. The value $\xi_\infty = 2w'(x \to \infty)$ can be read easily.

In Figure 17 we plot the dimensionless effective potential $u(x)$, $x = \ln \tilde{\rho}$. The upper two curves for large $x$ correspond to $g^2/4\pi = 1/40$, and the two lower ones to $g^2 = 0$. We observe indeed a crossover between two regions of almost constant potential for $\tilde{\rho} \to 0$ and $\tilde{\rho} \to \infty$. The potential is higher for $\tilde{\rho} \to \infty$. For $g^2 > 0$, this corresponds to the effective particle numbers $\tilde{N}_{U,\infty}$ and $\tilde{N}_{M,\infty}$. The dependence on $\xi_\infty$ is small except for the transition region. For the curves with $g^2 \approx 0.3$ the transition occurs for $\tilde{\rho} \approx 3$, according to $g^2\tilde{\rho} \approx 1$. The behavior of the two curves with $g^2 > 0$ is clearly dominated by the effect of the gauge and Yukawa couplings that result in the variation of the effective degrees of freedom. For the two curves with $g^2 = 0$ the crossover occurs for smaller $\tilde{\rho}$ as compared to the curves with $g^2 > 0$. For all curves, the potential $u(\tilde{\rho} \to 0)$ is close to the constant scaling solution $u_*$.

In Figure 18 we display the dimensionless mass term $\tilde{m}^2(\tilde{\rho}) = u'(\tilde{\rho})$. The parameters are the same as for Figure 17. For all curves, the mass term vanishes for $\tilde{\rho} \to \infty$, as expected for the asymptotic scaling solution. For $\tilde{\rho} \to 0$, the mass term switches to positive values, indicating that for all curves the minimum of the effective potential is situated at $\tilde{\rho} = 0$. The two upper curves on the left hand side correspond to the value $\xi_\infty = 1.0$, the lower ones to $\xi_\infty = 0.05$. For $\xi_\infty = 1$ the mass term at $\tilde{\rho} = 0$ is substantial. It depends only mildly on the value of $g^2$. For small $\xi_\infty = 0.05$ (two lower curves on the left), the mass term at $\tilde{\rho} = 0$ is much smaller. We observe that none of the curves show strong variations in the transition region that were found for $N_S = 1$, $N_V = N_F = 0$ near $\tilde{\rho} = 1/(64\pi^2)$. There seems to be no obstacle to continue the solutions to $\tilde{\rho} \to 0$. The small variations observed for large negative $x$ are presumably numerical errors, which blow up if the region is further

extended towards $x \to -\infty$. The observed mass terms are compatible with the difference $u(\tilde{\rho} = 0) - u_*$ according to Equation (101).

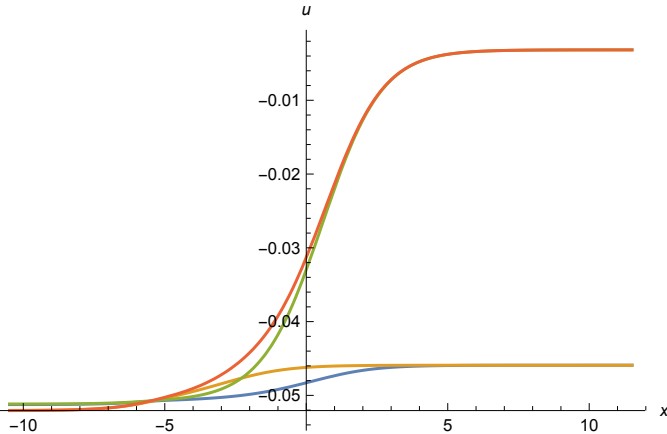

**Figure 17.** Effective potential $u$ as function of $x = \ln \tilde{\rho}$. The two upper curves are for $g^2/4\pi = y^2/4\pi = 1/40$. The red curve corresponds to $\xi_\infty = 1.0$, the green curve to $\xi_\infty = 0.05$. The two lower curves are for $g^2 = y^2 = 0$, with the orange curve for $\xi_\infty = 1.0$ and the blue curve for $\xi_\infty = 0.05$. Particle numbers are $N_S = 4$, $N_V = 12$, and $N_F = 45$ and initial conditions are set at $\tilde{\rho}_{as} = 5000$.

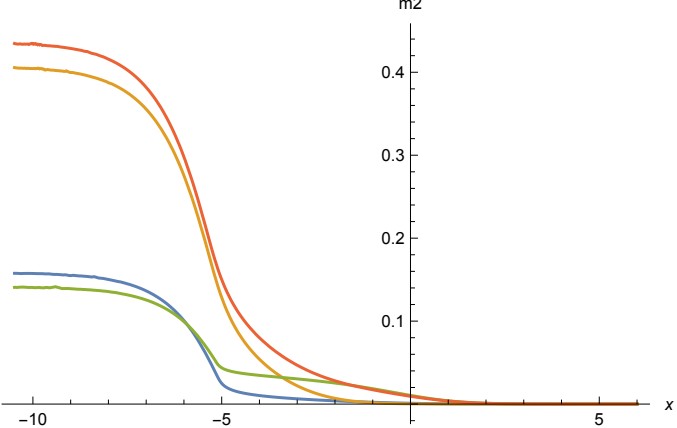

**Figure 18.** Mass term $u'(\tilde{\rho})$ as function of $x = \ln \tilde{\rho}$. Values of parameters and color coding are the same as in Figure 17. The two upper curves on the left (red and orange) correspond to $\xi_\infty = 1.0$, the two lower ones (blue and green) to $\xi_\infty = 0.05$.

Figure 16 displays $w'(\rho)$. For large $\tilde{\rho}$, it approaches the asymptotic value $\xi_\infty/2$, and it makes a crossover to larger values at $\tilde{\rho} = 0$. For large $\xi_\infty$ (two upper curves), the dependence on the value of $g^2$ is very small. The dependence on $g^2$ is more pronounced for the two lower curves with $\xi_\infty = 0.05$. The green curve for $g^2 > 0$ seems to show a somewhat sharper transition than the blue curve for $g^2 = 0$. Both for $u'(\tilde{\rho})$ and $w'(\rho)$, the transition between the constant values for $\tilde{\rho} \to \infty$ and $\tilde{\rho} \to 0$ occurs for $x \approx -5$, corresponding to $\tilde{\rho} \approx 1/(64\pi^2)$. It is related to the scalar sector, since the gauge boson masses $\sim g^2 \tilde{\rho}$ are already very small in the transition region. The caveats about the neglection of $u''$, $w''$ in this region remain valid.

We conclude that interesting crossover scaling solutions for the potential between two constant values for $\tilde{\rho} \to 0$ and $\tilde{\rho} \to \infty$ seem possible for an effective Planck mass that increases $\sim \tilde{\rho}$ due to a non-minimal coupling $\xi_\infty$. The region where the existence of a global solution is decided is the transition region around $\tilde{\rho} = 1/(64\pi^2)$. The smoothness in this transition region depends substantially on the particle content. It seems not unlikely that the number of global scaling solutions also depends on the particle content in a critical way.

## 8. Scaling Solutions for the Standard Model

The standard model of particle physics coupled to quantum gravity may be a consistent quantum field theory. This requires the presence of an UV fixed point, rendering the model asymptotically safe. The fixed point does not only concern a small number of couplings. It requires the existence of a scaling solution for functions as $u(\tilde{\rho})$ and $w(\tilde{\rho})$. Within our approximation, such scaling solutions indeed exist. For the standard model coupled to gravity, the gauge and Yukawa couplings can be asymptotically free. This is necessary for the non-abelian couplings, while for the abelian coupling $g_1$ another fixed point with $g_{1*} \neq 0$ may exist [66–71]. For the scaling solution we take here $g^2 = y^2 = 0$.

### 8.1. Constant Scaling Solution

The constant scaling solution with $\tilde{\rho}$-independent $u$ and $w$ is a viable scaling solution. This scaling solution predicts a vanishing quartic scalar coupling at the fixed point. Close to the fixed point, the quartic scalar coupling is an irrelevant parameter with critical exponent given by $-A$. In a complete theory it can therefore be predicted to take its fixed point value. The gauge and Yukawa couplings are relevant parameters at the fixed point. They will increase as the flow moves away from the fixed point towards the infrared. The flowing gauge and Yukawa couplings generate, in turn, a nonzero value for the quartic scalar coupling. As long as the graviton fluctuations remain important this value remains very small. A more substantial increase happens only for scales below the Planck mass. This simple picture has successfully predicted [3] the mass of the Higgs boson in the range that has later been observed [72–74].

For the constant scaling solution, the effective potential is flat and corresponds to a cosmological constant:

$$U(\rho) = U_0 = u_* k^4 \,. \tag{130}$$

The cosmological constant is negative, $u_* < 0$, and vanishes for $k \to 0$. The cosmological constant is a relevant parameter. Its flow away from the fixed point leads to a value of $U(\rho = 0)$ different from the one for the scaling solution (130). It is a free parameter and can be chosen arbitrarily, for example to coincide with the present observed dark energy density. The mass term for the Higgs potential is also a relevant parameter at the fixed point. Its value at $k = 0$ can be chosen such that the expectation value of the Higgs scalar coincides with the observed Fermi scale.

The constant scaling solution cannot account for Higgs inflation, however. The relevant coupling corresponding to the cosmological constant has to be chosen such that for $k \to 0$, the cosmological constant is very small. This implies that for $k$ larger than the Fermi scale, $U(k)$ is given by the scaling solution. Even if we could somehow identify $U(k)$ with the cosmological constant for a scale $k$ corresponding to the Hubble parameter $H$, (such an identification is far from obvious,) the value of $U(k)$ is negative and cannot describe cosmology close to de Sitter space. We extend the discussion of this issue to other candidate scaling solutions in Section 8.3, with a similar negative outcome. A possible alternative for inflation for the pure standard model coupled to gravity could be a large coefficient of the term $\sim R^2$ in the effective action, leading to Starobinsky inflation [75].

### 8.2. Crossover Potential

A possible realization of Higgs inflation, ref. [9] for the standard model coupled to quantum gravity needs a scaling solution different from the constant scaling solution. The non-minimal coupling $\xi$ should not vanish. We therefore explore the possible existence of other scaling solutions beyond the constant scaling solution. For large $\tilde{\rho}$, scaling solutions different from the constant scaling solution can take the form of the asymptotic scaling solution (110). A numerical investigation shows that such scaling solutions indeed seem to exist. As before, we fix initial conditions at some large $\tilde{\rho}_{as}$ as a function of the free parameter $\xi_\infty$. We find two ranges of solutions, one for small $\xi_\infty$ in the range $0 \leq \xi_\infty \lesssim 1.5$, the other

for large $\tilde\zeta_\infty \gtrsim 1000$. In the range $1.5 \lesssim \tilde\zeta_\infty \lesssim 1000$, the coupling $w$ turns negative, not consistent with stable gravity.

For this type of solution, shown in Figure 19, the potential is a crossover potential, as shown in Figure 20 for low values of $\tilde\zeta_\infty = 0.1$, 1, $10^3$, and $10^4$. All curves show a crossover from larger values of $u$ for $\tilde\rho \to \infty$ ($x \to \infty$) than for $\tilde\rho = 0$ ($x \to -\infty$). The minimum of the effective potential is at the origin, $\tilde\rho = 0$. For $\tilde\rho \to \infty$, all crossover scaling solutions approach a common constant, given by $u_\infty < 0$ according to Equation (116), with $\tilde{\mathcal{N}}_{U,*} = N_S + 2N_V - 2N_F - 8/3$. This crossover behavior continues for smaller values $\tilde\zeta_\infty < 0.1$, with a location of the crossover shifted further to the right. The parameter determining the location of the crossover is given by $\tilde\rho/\tilde\zeta_\infty$. We show $u(x)$ in Figure 21 in a smaller range around $x = \ln(1/(16\pi^2))$ for $\tilde\zeta_\infty = 2 \cdot 10^{-5}$, $10^{-4}$, $10^{-3}$, 0.03. As $\tilde\zeta_\infty$ approaches zero, the curves approach the constant scaling solution which is also shown as the horizontal straight line. Simultaneously, the location of the crossover to larger values moves to $\tilde\rho_{\mathrm{cross}} \to \infty$ as $\tilde\zeta_\infty \to 0$, realizing a smooth limit for the approach to the constant scaling solution. We show the corresponding scaling solutions $w(x)$ for the same small values of $\tilde\zeta_\infty$ in Figure 19, with constant scaling solution $w_*$ given by the horizontal line. We observe that the scaling solutions for small $\tilde\zeta_\infty$ all meet in a common point $\ln\tilde\rho \approx -5.05$, both for $u$ and for $w$. Around this point, the linearized differential Equation (A100) is valid.

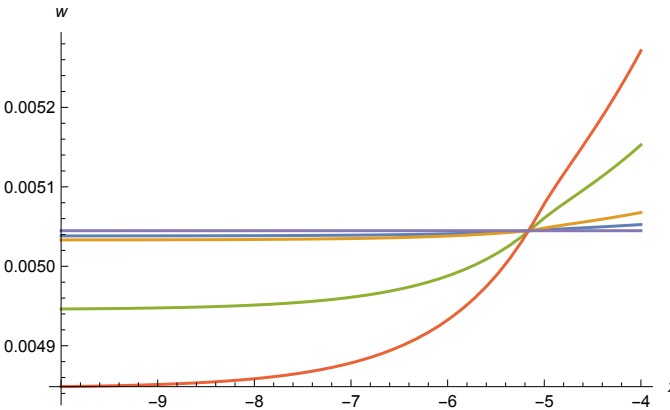

**Figure 19.** Dimensionless squared Planck mass $w$ as function of $x = \ln\tilde\rho$ for $\tilde\zeta_\infty = 2 \cdot 10^{-5}$ (blue), $10^{-4}$ (orange), $10^{-3}$ (green), and 0.003 (red), from top to bottom on the left. The horizontal line denotes the scaling solution which is approached for $\tilde\zeta_\infty \to 0$. All curves meet in a common point at $x \approx -5.05$.

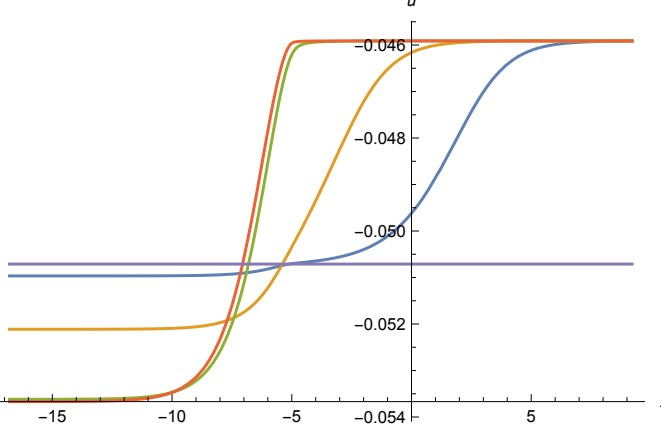

**Figure 20.** Effective potential $u$ as function of $x = \ln\tilde\rho$ for $\tilde\zeta_\infty = 0.1$ (blue), 1.0 (orange), $10^3$ (green), and $10^4$ (red), from right to left in the right part and from top to bottom in the left part. The horizontal line indicates the scaling solution. The particle content is the one of the standard model, $N_S = 4$, $N_V = 12$, and $N_F = 45$.

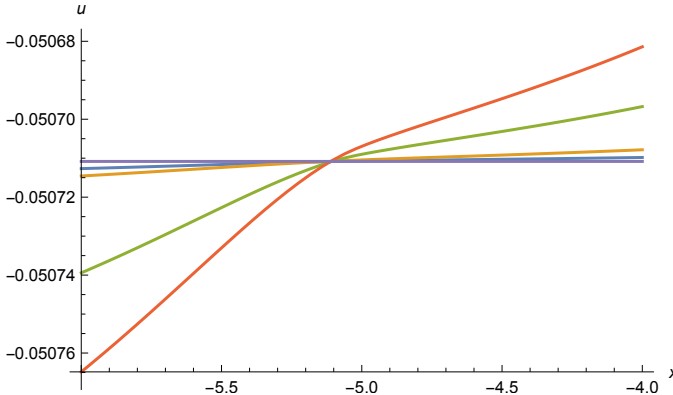

**Figure 21.** Effective potential $u$ as function of $x = \ln \tilde\rho$ for $\tilde\zeta_\infty = 2 \cdot 10^{-5}$ (blue), $10^{-4}$ (orange), $10^{-3}$ (green), and 0.003 (red), from bottom to top on the right side. The constant scaling solution, given by the horizontal line, is approached for $\tilde\zeta_\infty \to 0$. All curves meet in a common point at $x \approx -5.05$.

For all solutions the mass term $u'(\tilde\rho)$ increases as $\tilde\rho$ decreases, as shown in Figure 22 for $\tilde\zeta_\infty = 0.01$, 1.0, $10^3$, and $10^4$ from bottom to top. We display $\tilde m_0^2 = u'(\tilde\rho = 0)$, corresponding to the asymptotic limit $x \to -\infty$, in Table 1 (for very small $\tilde\rho$ our numerical solution starts to be unstable, and we take in practice $u'(\tilde\rho = 10^{-11})$). In addition, $\tilde\zeta(\tilde\rho) = 2\,w'(\tilde\rho)$ increases as $\tilde\rho$ decreases. The values $\tilde\zeta_0$ for $\tilde\rho \to 0$ are shown as well in Table 1. The numerical solutions for $\tilde\zeta_\infty$ in the range between $10^{-3}$ and 1 show a very narrow spike for a value of $x$ smaller than the point where all curves for $u$ and $w$ meet. So far we have not attempted for a better resolution of the spike. It is doubtful that the solutions in this range are acceptable scaling solutions. This would leave for $\tilde\zeta_\infty$ only two windows, either $\tilde\zeta_\infty < 10^{-3}$ or $\tilde\zeta_\infty > 10^3$.

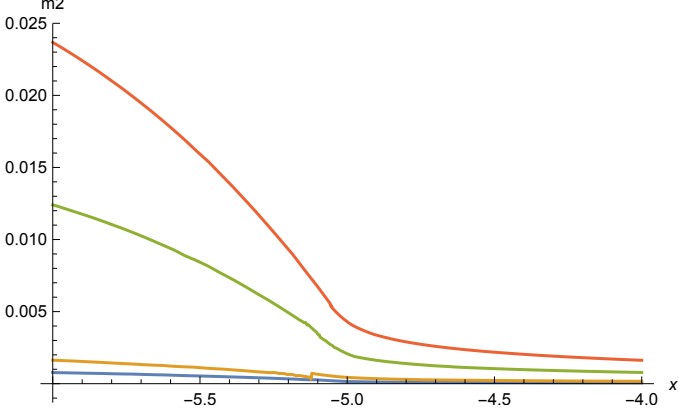

**Figure 22.** Dimensionless mass term $\tilde m^2 = u'$ as function of $x = \ln \tilde\rho$, for $\tilde\zeta_\infty = 0.1$ (blue), 1 (orange), $10^3$ (green), and $10^4$ (red), from bottom to top.

**Table 1.** Values of parameters at $\tilde{\rho} = 0$ for asymptotic scaling solutions with different parameters $\zeta_\infty$. We show the dimensionless mass term $\tilde{m}_0^2$, the nonminimal scalar-gravity coupling $\zeta_0$, and the quartic scalar coupling $\lambda_{H0}$. For $\zeta_\infty \geq 0.01$, the quartic scalar coupling gets very large and is not displayed here.

| $\zeta_\infty$ | $\tilde{m}_0^2$ | $\zeta_0$ | $\lambda_{H0}$ |
|---|---|---|---|
| $2 \cdot 10^{-5}$ | $8.6 \cdot 10^{-4}$ | $1.6 \cdot 10^{-3}$ | $1.3 \cdot 10^{-3}$ |
| $10^{-4}$ | $1.6 \cdot 10^{-3}$ | $2.5 \cdot 10^{-3}$ | $3.2 \cdot 10^{-3}$ |
| $10^{-3}$ | 0.019 | 0.035 | 0.63 |
| 0.003 | 0.034 | 0.065 | 2.28 |
| 0.01 | 0.073 | 0.156 | |
| 1.0 | 0.43 | 2.4 | |
| $10^3$ | 3.29 | 28.8 | |
| $10^4$ | 4.4 | 40.5 | |

### 8.3. Higgs Inflation

Higgs inflation [9,10] has been proposed as a possibility to accommodate the inflationary universe within the standard model. The original proposal has employed rather large values of the nonminimal scalar-gravity coupling $\zeta$. Smaller values seem also possible, while generally values $\zeta \gtrsim 10$ are assumed. In the presence of quantum gravity effects, even small $\zeta \ll 1$ could be compatible with realistic inflation [11]. The reason is the generic flattening of the scalar potential for large field values due to the fluctuations of the metric field.

Discussing these proposals in the light of the scaling solutions for quantum gravity, one encounters a major problem: The scaling potential remains negative for the whole range of $\tilde{\rho}$, while a positive potential would be required for inflation. The relevant quantity for inflation is actually the potential in the Einstein frame (with $\bar{M}$ the observed fixed Planck mass).

$$V_E = \frac{\bar{M}^4 \, U}{F^2} = \frac{\bar{M}^4 \, u}{4 \, w^2} \, . \tag{131}$$

We display $V_E$ in Figure 23 for $\zeta_\infty = 2 \cdot 10^{-5}$, $10^{-4}$, $10^{-3}$ and $0.003$ from right to left. It has a flat tail for $\rho \to \infty$, as suitable for inflation,

$$V_E(\rho \to \infty) = \frac{\bar{M}^4 \, u_\infty}{\zeta_\infty^2 \, \tilde{\rho}^2} = \frac{u_\infty \, \bar{M}^4 \, k^4}{\zeta_\infty^2 \, \rho^2} \, . \tag{132}$$

Successful inflation would need, however, a shift to positive values. A positive scaling potential could be achieved by adding additional bosonic particles, as in GUT models, but it is not possible for the particle content of the standard model alone.

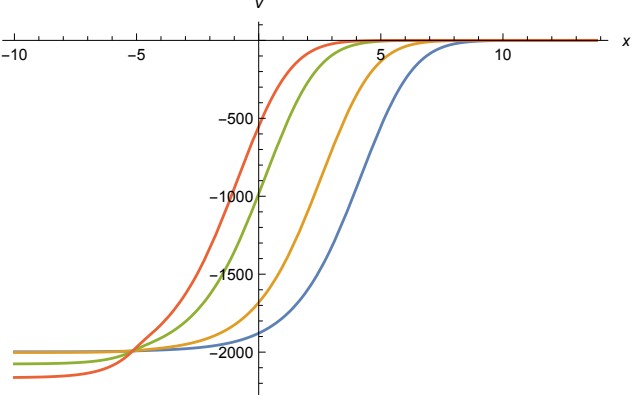

**Figure 23.** Potential in the Einstein frame $V_E$ as function of $x = \ln \tilde{\rho}$, for $\zeta_\infty = 2 \cdot 10^{-5}$ (blue), $10^{-4}$ (orange), $10^{-3}$ (green), and $0.003$ (red), from right to left.

### 8.4. Standard Model in the Einstein Frame

For a connection to observable quantities it is useful to transform all quantities to the Einstein frame with a constant Planck mass $\bar{M}$. The ratio:

$$\hat{V} = \frac{U}{F^2} = \frac{V_E}{\bar{M}^4} = \frac{u}{4\,w^2} \tag{133}$$

is a frame-invariant quantity. It does not change under a Weyl transformation of the metric, i.e.,

$$(g_E)_{\mu\nu} = \frac{F}{\bar{M}^2} g_{\mu\nu}\,. \tag{134}$$

With $K$ the prefactor of the kinetic term, ($K = 1$ in our truncation,) the frame-invariant expression for the kinetic term is [17,76]:

$$\hat{K} = \frac{K\,k^2}{F} + \frac{6\rho\,k^2}{F^2}\left(\frac{\partial F}{\partial \rho}\right)^2 = \frac{1}{2w} + \frac{6}{\tilde{\rho}}\left(\frac{\partial \ln w}{\partial \ln \tilde{\rho}}\right)^2\,. \tag{135}$$

In the Einstein frame, the kinetic term for the Higgs doublet $h$ reads ($\rho = h^\dagger h$):

$$\mathcal{L}_{kin} = \frac{\bar{M}^2\,\hat{K}}{k^2}\partial^\mu h^\dagger\,\partial_\mu h = \partial^\mu h_E^\dagger\,\partial_\mu h_E\,, \tag{136}$$

with $h_E$ the canonically normalized field in the Einstein frame, related to $h$ by:

$$\frac{\partial h_E}{\partial h} = \frac{\bar{M}}{k}\sqrt{\hat{K}}\,. \tag{137}$$

For the Einstein frame we define the dimensionless invariant:

$$\tilde{\rho}_E = \frac{\rho_E}{\bar{M}^2} = \frac{h_E^\dagger h_E}{\bar{M}^2}\,. \tag{138}$$

For a formulation with canonical kinetic terms we need to express $V_E/\bar{M}^4$ in terms of $\tilde{\rho}_E$.

For the relation between $\tilde{\rho}$ and $\tilde{\rho}_E$, we integrate Equation (137),

$$\tilde{\rho}_E = \frac{1}{4}\left(\int_0^{\tilde{\rho}} d\tilde{\rho}'\,\sqrt{\frac{\hat{K}}{\tilde{\rho}'}}\right)^2\,. \tag{139}$$

The dimensionless invariant $\tilde{\rho}_E$ only depends on the dimensionless variable $\tilde{\rho}$, without any explicit dependence on $k$. For the scaling solution, $\hat{V}$ and $\hat{K}$ are functions of $\tilde{\rho}$ without explicit dependence on $k$. Thus $k$ completely disappears in the Einstein frame. If a model is defined precisely on the fixed point, exact quantum scale symmetry is realized [11]. In this case the relevant cosmological field equations in the Einstein frame can be directly extracted from the field equation derived from the action:

$$S = \int_x \sqrt{g_E}\left\{-\frac{\bar{M}^2}{2}R_E + \frac{1}{2}\partial^\mu h_E^\dagger\,\partial_\mu h_E + \bar{M}^4\,\hat{V}(h_E)\right\}\,. \tag{140}$$

They do not involve the scale $k$, which therefore does not need to be specified.

The possible absorption of the re-normalization scale $k$ into a suitable normalization of fields is an important general property of scaling solutions. For scaling solutions $k$ constitutes the only field-independent mass scale. It can be interpreted as the scale at which an observer looks, such that fluctuations with wavelength larger than $k^{-1}$ do not influence the observation effectively. Since $k$ is the only scale, its value is arbitrary. It is therefore not surprising that it can be absorbed into a suitable field definition. Nevertheless, since some

scale must be present in order to provide for mass units of the fields, a scale will also be present if $k$ is absorbed in the field normalization. For the Einstein frame, this is the fixed Planck mass $\bar{M}$ in Equation (140). As an important consequence of this setting there is actually no need for a flow away from the scaling solution in order to make contact with observation.

For our scaling solutions, Equation (139) can be solved easily for limiting cases. For $\tilde{\rho} \to 0$ one has $w = w_0 + \xi_0 \tilde{\rho}/2$ and therefore:

$$\hat{K} = \frac{1}{2\,w_0} + (6\,\xi_0 - 1)\,\frac{\xi_0}{4w_0^2}\,\tilde{\rho}\,. \tag{141}$$

In leading order this yields:

$$\tilde{\rho} = 2\,w_0\,\tilde{\rho}_E\,. \tag{142}$$

In the limit of large $\tilde{\rho}$ we use $w = \xi_\infty\,\tilde{\rho}/2$ and

$$\hat{K} = \left(6 + \frac{1}{\xi_\infty}\right)\tilde{\rho}^{-1}\,. \tag{143}$$

This implies for $\tilde{\rho} \to \infty$:

$$\tilde{\rho}_E = \frac{1}{4}\left(6 + \frac{1}{\xi_\infty}\right)\ln^2\frac{\tilde{\rho}}{\tilde{\rho}_0} = \frac{1}{4}\left(6 + \frac{1}{\xi_\infty}\right)(x - x_0)^2\,. \tag{144}$$

Up to a constant factor the variable $x$ used in our figures can be associated directly with $|h_E|$ in the region of large $x$.

In the large-field region $\rho_E \gg \bar{M}^2$ the potential in the Einstein frame approaches exponentially zero,

$$V_E = \frac{u_\infty\,\bar{M}^4}{\xi_\infty^2}\,\exp(-2x_0)\,\exp\left\{-4\sqrt{\frac{\xi_\infty}{6\xi_\infty + 1}}\,\frac{\sqrt{h_E^\dagger h_E}}{\bar{M}}\right\}\,. \tag{145}$$

If one could shift $V_E$ by a positive constant this would be a flat region suitable for inflation. No such shift is possible, however, since $\hat{V}$ has to go to zero for $\tilde{\rho} \to \infty$. Since $u_\infty < 0$, the potential in the Einstein frame approaches zero from below. The scaling potential for the standard model is not compatible with Higgs inflation.

This issue extends to the scaling solution for many other models. Whenever $\xi_\infty > 0$, the potential in the Einstein frame has to approach zero for large field values. This results from the boundedness of $U$ by the graviton barrier [6]—namely $u$ cannot increase faster than $\tilde{\rho}$—combined with the increase $w \sim \tilde{\rho}$. The large field region is interesting for late cosmology since it leads to an asymptotic approach of $V_{\rm E}$, and therefore the dark energy density, to zero. This property excludes, on the other hand, the use of the large field region for inflation. What could still be possible is a role of intermediate regions where $\tilde{\rho}$ has not yet reached the asymptotic regime. This is particularly relevant for small values of $\xi_\infty$, since even for rather large $\tilde{\rho}$ the function $w(\tilde{\rho})$ may not be dominated yet by the increase in $\sim \tilde{\rho}$.

*8.5. Flow Away from the Scaling Solution*

The use of the scaling solution for all aspects of observation is possible, but not compulsory. The quantum field theory for the standard model and gravity may be defined only by an asymptotic approach to the fixed point in the ultraviolet. Then the values of relevant parameters play a role. The Planck mass and cosmological constant are relevant parameters. They can deviate from the scaling solution for small $k$. The leading relevant parameter is the Planck mass, typically in the form:

$$F = \bar{M}^2 + 2\,w_*\left(\frac{\rho}{k^2}\right)k^2\,, \tag{146}$$

with $w_*(\tilde{\rho})$ given by the scaling solution. For $k \to 0$, it assumes the form:

$$F = \bar{M}^2 + \xi_\infty \rho, \tag{147}$$

and we associate the integration constant $\bar{M}$ with the observed Planck mass (note that the constant $\bar{M}$ has the same value, but a different role as compared to the scaling solution in Equation (140)). The transition scale $k_t$ for the crossover from the scaling solution to the solution (147) for $k \to 0$ depends on $\rho$ according to:

$$\frac{\bar{M}^2}{k_t^2} = 2\,w^{(0)} + \xi_\infty \frac{\rho}{k_t^2}, \tag{148}$$

which yields:

$$k_t^2(\rho) = \frac{\bar{M} - \xi_\infty \rho}{2\,w_0}\,\theta\big(\bar{M}^2 - \xi_\infty \rho\big). \tag{149}$$

For large $\rho \gg \bar{M}^2/\xi_\infty$, the integration constant $\bar{M}^2$ plays no role and we can employ the scaling solution, e.g. $k_t(\rho) = 0$. In particular, there cannot be any constant shift in the behavior of $u(\tilde{\rho} \to \infty)$. As a consequence, the asymptotic behavior for $\tilde{\rho} \to \infty$ of $u(\tilde{\rho})$, $w(\tilde{\rho})$, and therefore also $V_E$ in Equation (145) is not affected. Modifications arise only for small enough $\tilde{\rho}$.

For $w$ the flow away from the scaling solution implies a strong increase of $w$ for $k \to 0$:

$$w = w_*(\tilde{\rho}) + \frac{\bar{M}^2}{2k^2} \approx w_{0*} + \frac{\xi_\infty \rho + \bar{M}^2}{2k^2}, \tag{150}$$

where we have parameterized an approximate form of $w_*(\tilde{\rho}) \approx w_{0*} + \xi_\infty \tilde{\rho}/2$. This increase is responsible for the decoupling of gravity for low $k$.

For the scalar potential in the Einstein frame $V_E$ away from the scaling solution we may use the ansatz:

$$U = u_*(\tilde{\rho})k^4 + V(\rho), \tag{151}$$

with $V = 0$ for the scaling solution. This results in:

$$V_E = \frac{V + u_* k^4}{\big(1 + \xi_\infty \rho/\bar{M}^2 + 2w_0^* k^2/\bar{M}^2\big)^2}. \tag{152}$$

If we assume that for the relevant epochs in cosmology we can take $k \to 0$, we remain with a potential that vanishes for $\rho \to \infty$, provided $V(\rho)$ does not increase too rapidly with $\rho$,

$$V_E = \frac{V(\rho)}{\big(1 + \xi_\infty \rho/\bar{M}^2\big)^2}. \tag{153}$$

What is needed is an understanding of $V(\rho)$. This function corresponds to a solution of the flow equation for $U$ as $k \to 0$. At fixed $\rho$ one has:

$$\partial_t U = 4c_U(\tilde{\rho})k^4. \tag{154}$$

With the ansatz (151) one finds:

$$\partial_t U = 4u_* k^4 - 2\tilde{\rho}\partial_{\tilde{\rho}} u_* k^4 + \partial_t V = 4c_{U*}(\tilde{\rho})k^4 + \partial_t V, \tag{155}$$

where we have inserted the equation for the scaling solution $u_*(\tilde{\rho})$. A comparison with Equation (154) yields:

$$\partial_t V = 4(c_U(\tilde{\rho}) - c_{U*}(\tilde{\rho}))k^4. \tag{156}$$

We recover the scaling solution for $V = 0$ and $c_U(\tilde{\rho}) = c_{U*}(\tilde{\rho})$. For small deviations from the scaling solution one can linearize the flow equation. In this regime, $\partial_t V$ is

characterized by a critical exponent. Typically $V$ corresponds to a relevant coupling which implies at least one free parameter for the general solution for $V$.

For $k^2 \ll \bar{M}^2$, the deviations from the scaling solution are not small. Understanding the $\rho$-dependencies of $V$ will require a numerical solution of the flow equation, which also takes into account the flow of other couplings, as gauge and Yukawa couplings, away from their fixed points. This is outside the scope of this paper. For $\rho \ll \bar{M}^2$, one expects that $V$ approaches the perturbative, almost quartic potential of the standard model. It is, however, the behavior of the potential at a much higher $\rho$, typically in the order $k_t^2$, that is relevant for Higgs inflation. For this region not much can be said at the present level of our investigation. The only direct consequence of the scaling solution remains the value of $\xi_\infty$. Given the restrictions on the asymptotic behavior for $\tilde{\rho} \to \infty$, however, compatibility with Higgs inflation seems unlikely. In particular, large values of $\xi_\infty$ are disfavored since $\rho$ enters the regime of the scaling solution already for values much smaller than $\bar{M}^2$.

*8.6. Non-Minimal Higgs-Curvature Coupling and Prediction for the Mass of the Higgs Boson*

The existence of scaling solutions places important constraints on the value of the non-minimal coupling. For the allowed branch of small values one typically needs $\xi_\infty \lesssim 10^{-3}$. There seems to exist another branch of high $\xi_\infty \gtrsim 10^3$, but it is not clear if scaling solutions with these rather extreme values survive a more extended truncation. From the point of view of observation only the branch with low $\xi_\infty$ is allowed. This is connected with the influence of the non-minimal coupling $\xi$ on the predicted mass of the Higgs boson. As a result of its relevance to particle physics, we discuss here this connection in detail.

The quartic coupling $\lambda$ of the Higgs self-interaction corresponds to an irrelevant parameter at the fixed point. It can therefore be predicted to take at short distances its fixed point value. The corresponding prediction for the mass of the Higgs boson to be 126 GeV with a few GeV uncertainty [3] agrees well with the experimental value of 125 GeV found later. The central value of the prediction depends on the pole mass of the top quark $m_t$. For $m_t = 171$ GeV, the prediction for the central value is lowered to 125 GeV.

The fixed point value of $\lambda$ is influenced by the non-minimal Higgs-curvature coupling $\xi_0 = \xi(\tilde{\rho} \to 0)$ [11]. This was assumed to be negligible for the prediction in [3] Since the scaling solutions restrict the possible values of $\xi_0$, we investigate here the influence of $\xi_0$ on the prediction of the mass of the Higgs boson. More generally, we investigate the influence of metric fluctuations on the position of the fixed point value $\lambda_*$.

The couplings $\lambda$ and $\xi_0$ are defined by:

$$\lambda = \left.\frac{\partial^2 u}{\partial \tilde{\rho}^2}\right|_{\tilde{\rho}=0}, \qquad\qquad \xi_0 = 2\left.\frac{\partial w}{\partial \tilde{\rho}}\right|_{\tilde{\rho}=0}. \tag{157}$$

The flow equations for $\lambda$ and $\xi_0$ can be obtained by taking suitable $\tilde{\rho}$-derivatives of Equations (90) and (91), evaluated at $\tilde{\rho} = 0$. For the flow of the quartic coupling one finds [7,11]:

$$\partial_t \lambda = A\lambda + \beta_\lambda^{(p)} - C_g, \tag{158}$$

with $\beta_\lambda^{(p)}$ the part induced by fluctuations of gauge bosons, fermions, and scalars, and $C_g$ a gravitational contribution. For $\beta_\lambda^{(p)}$ we may employ here the approximate one-loop expression:

$$\beta_\lambda^{(p)} = -\frac{3y_t^4}{4\pi^2} + \frac{171\alpha^2}{50}, \tag{159}$$

that is the same as in standard perturbation theory. Here $y_t$ is the Yukawa coupling of the top quark and $\alpha = g_2^2/(4\pi)$, with $g_2$ the SU(2)-gauge coupling of the standard model (we have taken for the hypercharge coupling $g_1$ the approximation $g_1 = g_2$, neglected all Yukawa couplings to fermions except for the top quark, as well as small contributions $\sim \lambda\alpha$, $\lambda y_t^2$, $\lambda^2$).

For $C_g$ one finds [7,11]:

$$C_g = \frac{A\xi_0}{w}\left(\tilde{m}^2 - \frac{\xi_0 v}{2}\right) - \frac{2A}{(1-v)w}\left(\tilde{m}^2 - \frac{\xi_0 v}{2}\right)^2 + Avw_2. \tag{160}$$

Here all quantities have to be evaluated at $\tilde{\rho} = 0$ and

$$w_2 = \frac{\partial^2 w}{\partial\tilde{\rho}^2}\Big|_{\tilde{\rho}=0}. \tag{161}$$

With,

$$v_1 = \frac{\partial v}{\partial\tilde{\rho}}\Big|_{\tilde{\rho}=0} = \left(\tilde{m}^2 - \frac{\xi_0 v}{2}\right)/w, \tag{162}$$

we observe that $C_g$ vanishes if $v$ is independent of $\tilde{\rho}$ and $w_2 = 0$,

$$C_g = A\left(\xi_0 - \frac{2w}{1-v}v_1\right)v_1 + Avw_2. \tag{163}$$

We may neglect $w_2$ and investigate the flow equations for $\tilde{m}^2$, $\xi_0$, and $v_1$. For the crossover scaling solutions one observes a very rapid increase of $v_1$ from $v_1(\xi_\infty = 10^{-4}) \approx 3$ to $v_1(\xi_\infty = 10^{-3}) \approx 40$. We doubt that a strong increase of $\partial v/\partial\rho \gtrsim 3$ is compatible with a consistent scaling solution. The variation of $v(\tilde{\rho})$ becomes even more dramatic for larger values of $\xi_\infty$. We take as a rather conservative bound $\xi_\infty < 10^{-4}$, restricting further the allowed range of small $\xi_\infty$ for which global scaling solutions could become possible.

It turns out that the value of $\tilde{m}_0^2$ for the scaling solution of the standard model is actually rather small. If we neglect it, one has:

$$v_1 = -\frac{\xi_0 u}{2w^2}. \tag{164}$$

This approximation yields the simplified flow Equation (1), which permits a qualitative view on the influence of the non-minimal coupling. For quantitative computations we include the effect of $\tilde{m}^2$.

In the gravity-dominated regime for $k^2 \gg \bar{M}^2$ and for small $\beta_\lambda^{(p)}$, the flow of $\lambda(k)$ is characterized by an approximate partial fixed point:

$$\lambda_* = \frac{3y_t^4}{4\pi^2 A} - \frac{171\alpha^2}{50A} + \frac{C_g}{A}. \tag{165}$$

This partial fixed point is valid for $k > k_t$ and constitutes the "initial value" for the flow in the low-energy regime $k < k_t$, for which gravitational effects vanish rapidly due to decreasing $A$, and only $\beta_\lambda^{(p)}$ survives effectively. For $v$ and $w$ we can take the values for the scaling solution at the UV-fixed point, while $y_t$ and $\alpha$ can be found from extrapolating the observed low energy couplings to $k_t$ by use of the perturbative re-normalization group.

*8.7. Prediction for the Mass of the Top Quark*

It is our aim to investigate the effect of the non-vanishing Higgs-curvature coupling $\xi_0$ on the prediction of the mass of the Higgs boson. Since the prediction of $\lambda(k_t)$ actually results in a prediction of the ratio of the mass of the Higgs boson compared to the mass of the top quark [11], and the mass of the Higgs boson is accurately measured, we may turn this to an investigation of the effect of:

$$\Delta\lambda_g = \frac{C_g}{A} \tag{166}$$

on the prediction for the mass of the top quark $m_t$. For a quantitative analysis we employ the estimate [11] that a value $\Delta\lambda = 0.014$ decreases $m_t$ by 1 GeV,

$$\frac{m_t - m_{t,0}}{\text{GeV}} = -71.4\Delta\lambda, \tag{167}$$

with $m_{t,0}$ the prediction for $C_g = 0$.

The values of $v$ and $w$ at the fixed point are given for the standard model as:

$$
\begin{aligned}
u_* &= -0.0507 \, (-0.0508), \\
v_* &= -10.05 \, (-10.27), \\
w_* &= 0.00505 \, (0.00495). 
\end{aligned}
\tag{168}
$$

Here the first value corresponds to the constant scaling solution, as computed in [8], while the value in brackets corresponds to a typical crossover scaling solution as described in this section. Since the two values are rather similar, only a modest uncertainty is related to the difference of these values. The gravity induced anomalous dimension reads:

$$A = \frac{5}{24\pi^2 w(1-v)^2} + \frac{1}{96\pi^2 w(1+v/4)^2} \approx 0.051, \tag{169}$$

where the second term arises from the physical scalar fluctuation in the metric. Due to the large negative value of $v$ for the standard model, one finds that $A$ is substantially smaller than one, and the second term contributes of similar size as the first tern, in contrast to $v > 0$.

Let us first discuss the value of $\Delta\lambda$ for the scaling solution with vanishing gauge and Yukawa couplings. For the constant scaling solution one has $\Delta\lambda = 0$. This solution has a vanishing Higgs-curvature coupling, $\xi_0 = \xi_\infty = 0$. For the possible crossover scaling solutions, $\Delta\lambda$ corresponds to $\lambda_{H0}$ in Table 1. There is a one-parameter family of crossover solutions parameterized by $\xi_\infty$. Only a range of very small $\xi_\infty$ of the order of a few times $10^{-4}$ or less is consistent with the observed mass ratio between Higgs boson and top quark mass, even if we admit an uncertainty in the present experimental determination of the pole mass for the top quark of one or two GeV and theoretical uncertainties of a similar order. In particular, large values of $\xi_\infty \gg 1$, as often used for Higgs inflation, are not compatible with asymptotic safety for quantum gravity coupled to the standard model. This points towards the constant scaling solution, which is the only one that is firmly established within our truncation.

The constant scaling solution for the standard model predicts $\lambda_* = 0$. This is compatible with the observed value of the mass of the Higgs boson and top quark. Since the scaling solution has a vanishing gauge and Yukawa couplings, and these couplings are non-zero at the transition scale $k_t$ where the metric fluctuations decouple, the gauge and Yukawa couplings have to flow away from the fixed point before $k_t$ is reached. We next estimate the quantitative effect of this "flow away" for the prediction of the top quark mass.

*8.8. Influence of Gauge and Yukawa Couplings*

For the constant scaling solution (or crossover scaling solutions with small $\xi_\infty$), the flow away from the fixed point could induce a more sizable $\Delta\lambda$ due to the effects of gauge and Yukawa couplings. We therefore include next the effect of gauge and Yukawa couplings

to the flow of $\xi_0$. They may lead to a value of $\xi_0(k_t)$ that differs from the fixed point value given in Table 1. The $\tilde{\rho}$- derivative of Equation (91) at $\tilde{\rho} = 0$, yields:

$$
\begin{aligned}
\partial_t \xi =& \frac{25\left(\tilde{m}^2 - \frac{\xi v}{2}\right)}{128\pi^2 w(1-v)^2} \\
&- \frac{\tilde{m}^2 - \frac{\xi v}{2}}{1152\pi^2 w} \frac{\partial}{\partial v}\left(\frac{8 + v - \frac{9}{16}v^2}{\left(1 - \frac{v}{4}\right)^2}\right) - \frac{3w_2}{32\pi^2} \\
&+ \frac{1}{192\pi^2}\left\{4\partial_{\tilde{\rho}} N_V - \partial_{\tilde{\rho}} N_S - \partial_{\tilde{\rho}} N_F\right\}.
\end{aligned}
\tag{170}
$$

Here the first term arises from graviton (transverse traceless tensor) fluctuations, the second term from the physical scalar in the metric in the approximation of neglected mixing with other scalars, and the third accounts for the non-minimal coupling to gravity for the scalar fluctuations. The remaining parts reflect the contribution from gauge couplings, Yukawa couplings and scalar mass terms and self interactions.

The particle contributions result from the reduction of effective particle numbers due to mass terms. For gauge bosons with squared masses $m_i^2 = g_i^2 \rho$ one has:

$$
\partial_{\tilde{\rho}} N_V = \frac{3}{2}\sum_i \partial_{\tilde{\rho}} \frac{1}{1 + g_i^2 \tilde{\rho}} = -\frac{3}{2}\sum_i \frac{g_i^2}{(1 + g_i^2 \tilde{\rho})^2}.
\tag{171}
$$

For the standard model this results in a contribution:

$$
\partial_t \xi^{(g)} = -\frac{1}{32\pi^2}(2g_w^2 + g_z^2) = -\frac{3}{64\pi^2}\left(g_2^2 + \frac{g_1^2}{5}\right),
\tag{172}
$$

where we use $g_w^2 = g_2^2/2 = 2\pi\alpha$ and $g_z^2 = g_2^2/2 + 3g_1^2/10$. For the top quark with $m_t^2 = h_t^2 \rho$ one has:

$$
\partial_{\tilde{\rho}} N_F = 6\partial_{\tilde{\rho}} \frac{1}{1 + y_t^2 \tilde{\rho}} = -\frac{6y_t^2}{(1 + y_t^2 \tilde{\rho})^2},
\tag{173}
$$

resulting in:

$$
\partial_t \xi^{(t)} = \frac{y_t^2}{32\pi^2}.
\tag{174}
$$

Finally, for the scalar fluctuations one has:

$$
\begin{aligned}
\partial_{\tilde{\rho}} N_S &= \partial_{\tilde{\rho}}\left(\frac{3}{1 + \tilde{m}^2} + \frac{1}{1 + \tilde{m}^2 + 2\tilde{\rho}\lambda}\right) \\
&= -\frac{3\lambda}{(1 + \tilde{m})^2} - \frac{3\lambda}{(1 + \tilde{m}^2 + 2\tilde{\rho}\lambda)^2},
\end{aligned}
\tag{175}
$$

and therefore:

$$
\partial_t \xi^{(s)} = \frac{\lambda}{32\pi^2(1 + \tilde{m})^2}.
\tag{176}
$$

The results (172), (174), and (176) for $\tilde{m}^2 = 0$ and $\tilde{\rho} = 0$ can also be obtained from one-loop perturbation theory.

We will neglect the scalar contribution (176) as compared to the much larger top-quark contribution (174). Furthermore, the physical scalar metric fluctuations contribute to a large negative $v$ similar to the graviton fluctuations with $25/128$ replaced by $-7/144$. Neglecting $w_2$ we obtain the approximate flow equation for $\xi_0$,

$$\partial_t \xi_0 = \frac{19 v_1}{128 \pi^2 (1-v)^2} + \frac{y_t^2}{32 \pi^2} - \frac{9 \alpha}{40 \pi}. \tag{177}$$

With $y_t (10^{18} \text{GeV}) \approx 0.38$, $y_t^2 / (32 \pi^2) \approx 4.6 \cdot 10^{-4}$, $9 \alpha / (40 \pi) \approx 1.8 \cdot 10^{-3}$, the gauge boson fluctuations tend to induce for $v_1 = 0$ a small positive $\xi_0$ as $k$ flows towards the IR. We further need the flow equation for $v_1$:

$$\partial_t v_1 = \frac{1}{w} \left\{ \partial_t \tilde{m}^2 - \frac{\xi_0}{2w} \partial_t u - \frac{v}{2} \partial_t \xi_0 + \left( \frac{\xi_0 v}{2w} - v_1 \right) \partial_t w \right\}. \tag{178}$$

The flow of $u$ and $w$ at $\tilde{\rho} = 0$ does not depend directly on the gauge and Yukawa couplings. It vanishes for the fixed point until the transition region near $k_t$ is reached. We may neglect $\partial_t u$ and $\partial_t w$ in Equation (178), such that the influence of the gauge and Yukawa couplings arises from the flow of $\tilde{m}^2$ and $\xi_0$. We can employ:

$$\begin{aligned} \partial_t \tilde{m}^2 &= (A-2) \tilde{m}^2 - \frac{1}{2} A \xi_0 v \\ &\quad + \frac{1}{32 \pi^2} (\partial_{\tilde{\rho}} N_S + 2 \partial_{\tilde{\rho}} N_V - 2 \partial_{\tilde{\rho}} N_F) \\ &\approx -2 \tilde{m}^2 + A w v_1 + \frac{3 y_t^2}{8 \pi^2} - \frac{9}{64 \pi^2} \left( g_2^2 + \frac{g_1^2}{5} \right). \end{aligned} \tag{179}$$

The constant scaling solution has $\tilde{m}^2 = \xi_0 = v_1 = 0$. If we assume a bound for the effective contribution of gauge and Yukawa couplings as:

$$|v_1| < \frac{3 c_1 y_t^2}{8 \pi^2 w}, \qquad\qquad |\xi_0| < \frac{y_t^2 c_2}{32 \pi^2}, \tag{180}$$

we conclude that the contribution of flowing gauge and Yukawa couplings to $\Delta \lambda$ is bounded by:

$$|\Delta \lambda|_{g,h} < \frac{3 c_1 c_2 y_t^4}{256 \pi^4 w}, \tag{181}$$

which is of the order of a few times $c_1 c_2 10^{-6}$. Given that $c_1$ and $c_2$ are typically smaller than one due to cancellations between Yukawa and gauge couplings, and the flow of $g_2$ and $y_t$ deviating substantially from zero only in vicinity of $k_t$, we conclude that the effect of the flowing gauge and Yukawa couplings is too small for influencing the prediction of the top quark mass.

For the flow away from the constant scaling solution of the standard model coupled to gravity, the dominant contribution to $\Delta \lambda$ seems to arise from the particle fluctuations,

$$\Delta \lambda = \frac{C_p}{A} = \frac{1}{A} \left( \frac{3 y_t^4}{4 \pi^2} - \frac{171 \alpha^2}{50} \right) = -\frac{\beta_\lambda}{A}. \tag{182}$$

Typically, $\beta_\lambda (k_t)$ is slightly positive, with details depending on $m_t$ [77,78]. With the unusually small value of $A$ for the standard model, $1/A \approx 20$, value $\beta_\lambda \approx 10^{-3}$ enhances the central value of the prediction for the top quark mass by around $1.5 \, \text{GeV}$. This comparatively large effect is due to the exceptionally small value of $A$ for the standard model. It remains, nevertheless, within the uncertainty quoted in [3]. If the theoretical uncertainties can be reduced below this level, a precision measurement of the pole mass for the top quark could distinguish between the asymptotically safe standard model coupled to gravity with its small value of $A$, and other models as grand unification which typically have $A$ of the order one or even larger. A dedicated solution of the combined set of flow

equations in the threshold region around $k_t$, together with a matching to three-loop running for $k \ll k_t$, should improve our rough estimates in this case.

For the standard model, coupled to quantum gravity, a rather consistent picture emerges. The scaling solution is the constant scaling solution. The ratio between Higgs boson mass and top quark mass is predicted in the range where it is observed. This prediction is rather robust even for a rather extended truncation in the gravitational sector. Higgs inflation is unlikely to be realized. Inflation will then require a scalar degree of freedom beyond the Higgs doublet. This may either be an explicit additional singlet field as the cosmon, or an effective scalar field arising from terms $\sim R^2$ in the effective action which could realize Starobinsky inflation.

### 8.9. Simultaneous Prediction of Top Quark Mass and Higgs Boson Mass

Let us look at Equation (158) from a different perspective. For a scaling solution, the r. h. s. of Equation (158) has to vanish,

$$\beta_\lambda^{(p)} = C_g - A\lambda. \tag{183}$$

We can write this in the form:

$$y_t^4 - \frac{114\pi^2}{25}\alpha^2 = \frac{4\pi^2 A}{3}X, \tag{184}$$

with

$$X = \lambda - \left(\xi_0 - \frac{2w}{1-v}v_1\right)v_1 - vw_2. \tag{185}$$

If a scaling solution fixes $X$, or implies a bound limiting $X$ to sufficiently small values, the Yukawa coupling can be related to the gauge coupling. This fixes the ratio between the top quark mass and the mass of the W-boson. The precise relation, taking into account the difference between $g_1$ and $g_2$ and the running of all couplings, can easily be worked out. The outcome is that for vanishing $X$, the prediction of $m_t$ agrees well with observation. If the properties of the scaling solution imply a small enough $X$ independently of the precise value of $\lambda$, both $m_t/m_W$ and $m_H/m_t$ are predicted simultaneously. The question we raise here concerns possible restrictions on $X$ that do no depend (or only very mildly depend) on $\lambda$.

A first simple case concerns constant scaling solutions which predict $\lambda = 0$, $\xi_0 = 0$, $v_1 = 0$, $w_2 = 0$, and $X = 0$. It is an interesting question if a constant scaling solution is also possible for non-zero fixed point values of gauge and Yukawa couplings, $g_*^2 > 0$, $y_*^2 > 0$. This concerns GUT models [19] as well as the standard model if the hypercharge coupling $g_1$ takes a value $g_{1*} \neq 0$ [66–71]. If such a fixed point exists, the contributions of particle fluctuations to the scaling form of $\partial_{\tilde{\rho}} u_*(\tilde{\rho})$ and $\partial_{\tilde{\rho}} w_*(\tilde{\rho})$ have to vanish. For the Higgs potential this implies that $\beta_\lambda^{(p)}$ in Equations (158) and (159) has to vanish, since a constant scaling solution has $\lambda_* = 0$, $C_g = 0$. As a direct consequence, the Yukawa coupling of the top quark and therefore $m_t$ can be predicted as a function of the gauge coupling. In this case not only the ratio $m_h/m_t$, but $m_t$ and $m_H$ separately are indeed predicted. The extrapolation of the running Yukawa and gauge couplings to the vicinity of the Planck mass yields indeed a very small value of $\beta_\lambda$. For $m_t$ near 171 GeV, both $\lambda$ and $\beta_\lambda$ vanish in this region. Keeping in mind small corrections from the flow away from the fixed point the prediction agrees with the observation. For this type of prediction, it is actually sufficient that the scaling solution is constant in the region near $\tilde{\rho} \approx 1$.

Such a scenario seems not to be compatible with a minimal standard model since the weak and strong gauge couplings have to flow away from their vanishing fixed point values substantially before $k_t$ is reached (this concerns either the flow with $k$ or the flow with $\tilde{\rho}$). It could be realized in GUT models however, where all gauge couplings of the standard model take a common fixed point value $\alpha \approx 1/40$. A realization of this scenario

needs a truncation beyond the present one. Within our truncation one has for a constant scaling solution:

$$\frac{\partial u_*}{\partial\tilde{\rho}} = 0 \Leftrightarrow \partial_{\tilde{\rho}}N_V = \partial_{\tilde{\rho}}N_F \tag{186}$$

and

$$\frac{\partial w_*}{\partial\tilde{\rho}} = 0 \Leftrightarrow 4\partial_{\tilde{\rho}}N_V = \partial_{\tilde{\rho}}N_F. \tag{187}$$

This implies that $\partial_{\tilde{\rho}}N_V$ and $\partial_{\tilde{\rho}}N_F$ both have to vanish, and therefore zero gauge and Yukawa couplings. In an extended truncation, other couplings may contribute to $\partial_{\tilde{\rho}}w_*$ and a constant scaling solution could become possible with $\partial_{\tilde{\rho}}N_V \neq 0$, $\partial_{\tilde{\rho}}N_F \neq 0$.

A possible criterion selecting for $\tilde{\rho} \approx 1$ local scaling solutions $u(\tilde{\rho})$, $w(\tilde{\rho})$, which are very close to the constant solution (i.e., both $\lambda$ and $X$ very small), could be that a stronger dependence of the local scaling functions on $\tilde{\rho}$ could lead to problems in the transition region $\tilde{\rho}_t \approx 1/(64\pi^2)$. In this case, smooth global scaling solutions would only be possible if $g$ and $y$ are related, thus leading to the prediction of $m_t/m_W$.

## 9. Conclusions

Gravitational effects have a profound impact on the effective potential for scalar fields. They largely determine the shape of the potential at re-normalization scales close to the Planck mass. This constitutes the initial conditions for the flow towards smaller scales where comparison with observation becomes possible.

### 9.1. Summary

The consistency conditions for asymptotically safe quantum field theories require the existence of scaling solutions for functions of fields. They extend the re-normalizability conditions for a finite number of couplings in asymptotically free theories. We investigated the shape of the effective potential for scalar fields at and near the ultraviolet fixed point of asymptotically safe quantum gravity. We found constant scaling solutions with a completely flat potential and vanishing gauge and Yukawa couplings. More general scaling potentials do not have a polynomial form. Often they are characterized by a crossover between two constants for small and large fields. Nonvanishing gauge couplings can induce spontaneous symmetry breaking due to a potential minimum at nonzero field values. In contrast, Yukawa couplings to fermions tend to stabilize a minimum at zero field values. A nonminimal coupling between the scalar field and gravity is associated to a field-dependent effective Planck mass. It can induce spontaneous symmetry breaking. In general, nonzero gauge, Yukawa, or nonminimal couplings prevent the scaling potential to be completely flat. We discussed scaling solutions with a constant effective potential for large fields and a non-zero nonminimal scalar coupling to gravity. They solve the cosmological constant problem asymptotically for the large field values that are reached for large cosmic times. For the standard model coupled to asymptotically safe quantum gravity, the non-minimal Higgs-curvature coupling is bound to be small, $\xi_\infty \lesssim 10^{-4}$, in contrast to large values $\xi_\infty > 1$ often assumed for Higgs inflation. For the pure standard model coupled to quantum gravity, it seems unlikely that Higgs inflation is compatible with asymptotic safety. We have discussed small modifications of asymptotic safety prediction for the ratio of top quark and Higgs boson mass due to nonzero gauge, Yukawa, and non-minimal couplings.

### 9.2. Ultraviolet Fixed Point

An ultraviolet fixed point defines a consistent quantum field theory for gravity coupled to particle physics. In this asymptotic safety scenario, not only a few couplings as the dimensionless Planck mass or cosmological constant take fixed values at the fixed point. Whole functions, as the dimensionless effective potential $u = U/k^4$, take a fixed form as

functions of dimensionless invariants formed from scalar fields. Typical invariants are $\tilde{\rho} = \chi^2/(2k^2)$ for a singlet field $\chi$ or $\tilde{\rho} = h^\dagger h/k^2$ for the Higgs doublet $h$. This fixed form is the scaling potential $u_*(\tilde{\rho})$ which does not depend on $k$. Another important scaling function is $w_*(\tilde{\rho}) = F/(2k^2)$, with $F$ the field-dependent coefficient of the curvature scalar in the effective action, corresponding to a field- and scale-dependent squared Planck mass. A fixed point is characterized by infinitely many such scaling functions that constitute the scaling functional, corresponding to a $k$-independent effective average action expressed in terms of dimensionless variables.

The fixed point is characterized by a powerful symmetry, namely quantum scale symmetry. It is realized if the flow generators or $\beta$-functions for whole functions vanish. The differential equations corresponding to this condition may be called the "scaling equations". Solutions of these scaling equations are the "scaling solutions". The requirement of existence of global scaling solutions for the whole range of $\tilde{\rho}$ from zero to infinity imposes new, very strong constraints on the short distance behavior of a given model. These conditions are conceptually similar to the condition of re-normalizability in quantum field theories in flat space. Concerning whole functions, which correspond to infinitely many couplings, the scaling conditions are stronger than the usual conditions for re-normalizability. They constrain the allowed microscopic values of many re-normalizable parameters and lead to an enhanced predictivity of a theory. In short, they are the new consistency criteria for models that remain valid to infinitely short distances and need no further ultraviolet completion.

In this paper we investigated the form of the scaling potential $u_*(\tilde{\rho})$, together with $w_*(\tilde{\rho})$. We also discuss the flow of $u(\tilde{\rho})$ and $w(\tilde{\rho})$ away from the ultraviolet fixed point. In the vicinity of the fixed point, one can linearize the flow of small deviations of couplings from their fixed point values, encoded in $\delta u(\tilde{\rho}) = u(\tilde{\rho}) - u_*(\tilde{\rho})$ and $\delta w(\tilde{\rho}) = w(\tilde{\rho}) - w_*(\tilde{\rho})$. A few relevant parameters, that describe deviations that increase for the flow away from the fixed point towards the infrared, determine all observable quantities of a given model. If there are less relevant parameters as compared to the number of re-normalizable couplings in the standard model, relations between the standard model couplings become predictable.

Due to the presence of relevant parameters, the observable effective potential $U(\rho) = k^4 u(\rho/k^2)$ for $k \to 0$, evaluated in units of the Planck mass, $U/F^2 = u/(4w^2)$, differs from the scaling potential $u_*/(4w_*^2)$. Nevertheless the scaling form determines many properties of the observable effective potential in the Einstein frame, $V_E = \bar{M}^4 U/F^2$, for $k \to 0$. The scaling form is the boundary value or "initial value" of the flow for $k \to \infty$. For example, it determines the UV-value of all quartic scalar couplings in GUT models which therefore become predictable for a given model [79].

We find that there is actually no need for deviations from the scaling potential. Certain scaling solutions may be compatible with observation. If all relevant parameters for deviations from the fixed point vanish, the model exhibits "fundamental scale invariance" (see below). The predictivity of a theory with fundamental scale invariance is enhanced even further, and no free relevant parameters are available anymore.

*9.3. Scaling Solutions*

A particular scaling solution is the constant scaling solution for which $u_*(\tilde{\rho})$ and $w_*(\tilde{\rho})$ are independent of $\tilde{\rho}$, while gauge and Yukawa couplings vanish. This constant scaling solution is the simplest extension of the Gaussian fixed point in particle physics, for which all particles are massless free particles, to quantum gravity where the gravitational couplings do not vanish. If there is a unique constant scaling solution it corresponds to the extended Reuter fixed point [4]. For a truncation with two scale dependent functions $u(\tilde{\rho}; k)$, $w(\tilde{\rho}; k)$, this fixed point has been studied in detail for many particle physics models in [8].

In the present paper, we ask if there could be other fixed points distinct from the extended Reuter fixed point with constant scaling solutions. Non-trivial scaling functions $u_*(\tilde{\rho}), w_*(\tilde{\rho})$ may be induced by non-vanishing gauge or Yukawa couplings. In our trun-

cation with two functions $u(\tilde{\rho})$ and $w(\tilde{\rho})$, plus constant gauge couplings $g$ and Yukawa couplings $y$, any nonzero $g$ or $y$ necessarily induces a non-trivial $\tilde{\rho}$-dependence of $u_*(\tilde{\rho})$ and $w_*(\tilde{\rho})$. Thus any fixed point that is not asymptotically free ($g_* = y_* = 0$) in the particle sector leads to nonzero $\partial_{\tilde{\rho}} u_*(\tilde{\rho})$ and $\partial_{\tilde{\rho}} w_*(\tilde{\rho})$ in this truncation.

One possibility that we have not yet investigated in the present paper is a field-dependence of gauge and Yukawa couplings, e. g. replacing constant $g$ and $y$ by functions $g(\tilde{\rho})$ and $y(\tilde{\rho})$. A dependence on the dimensionless combination $\tilde{\rho}$ is consistent with a scaling solution and does not introduce any intrinsic mass or length scale. For an interesting scenario, a scaling solution may approach for $\tilde{\rho} \to 0$ the Reuter fixed point of a constant scaling solution with $g(\tilde{\rho} = 0) = 0$, $y(\tilde{\rho} = 0) = 0$. In contrast, for large $\tilde{\rho}$ a different scaling behavior may be approached, with $g(\tilde{\rho} \to \infty) = g_* \neq 0$, $y(\tilde{\rho} \to \infty) = y_* \neq 0$. For the flow with $k$, it was observed that certain GUT models admitted two fixed points with $g = 0$ and $g = g_*$, flowing from the first to the second as $k$ decreased [19,79]. This is expected to translate to the $\tilde{\rho}$-flow for scaling solutions, with decreasing $k$ corresponding to increasing $\tilde{\rho}$. In models where $w(\tilde{\rho} \to \infty) = \xi_\infty \tilde{\rho}/2$, the values of gauge and Yukawa couplings near the Planck scale correspond to the fixed point values $g_*$ and $y_*$ for $\tilde{\rho} \to \infty$. They will be predicted for a given particle content of a model with nonzero $g_*$ and $y_*$. We expect that many features of the scaling solutions with constant $g$ and $y$ carry over to the scenario with field dependent $g(\tilde{\rho})$ and $y(\tilde{\rho})$. Even if the fixed point is the extended Reuter fixed point, the flow away from the fixed point will involve nonzero $g$ and $y$. If the flow of $g$ and $y$ is slow, the flowing functions $u(\tilde{\rho})$ and $w(\tilde{\rho})$ will still be characterized by partial fixed points that are close to the scaling functions for nonzero constant $g$ and $y$. Understanding the $\tilde{\rho}$-dependence of possible scaling functions or approximate scaling functions for non-vanishing particle couplings is crucial for a connection to the observable quantities for $k \to 0$.

### 9.4. Properties of the Scaling Potential

A complete answer to the question of the possible shapes of $u(\tilde{\rho})$ and $w(\tilde{\rho})$ in the presence of gauge and Yukawa couplings is rather complex. At the present stage of the investigation, we found several important general features of candidate scaling functions $u_*(\tilde{\rho})$, $w_*(\tilde{\rho})$ which have a non-trivial $\tilde{\rho}$-dependence:

1. The scaling functions $u_*(\tilde{\rho})$ and $w_*(\tilde{\rho})$ do not have a polynomial form;
2. The scaling potential $u_*(\tilde{\rho})$ is often characterized by a crossover between a constant $u_0$ for $\tilde{\rho} \to 0$ to a different constant $u_\infty$, for $\tilde{\rho} \to \infty$;
3. A negative quartic coupling $\lambda = \partial^2 u / \partial \tilde{\rho}^2$ at $\tilde{\rho} = 0$ is not a sign of instability. It only characterizes the Taylor expansion around $\tilde{\rho} = 0$, which does not describe the overall potential if the scaling potential does not have a polynomial form. An example is the crossover of a potential with a minimum at $\tilde{\rho} = 0$ and $\tilde{m}_0^2 = \partial u / \partial \tilde{\rho}$ at $\tilde{\rho} = 0$ being positive, to a constant value $u_\infty > u_0$ for $\tilde{\rho} \to \infty$. A negative value of $\lambda$ only implies that $\tilde{m}^2(\tilde{\rho})$ decreases for increasing $\tilde{\rho}$, which is typical for a crossover;
4. It is possible that families of scaling solutions exist, characterized by one or even several continuous parameters. This would be in contrast to a single UV-fixed point or a discrete set of such fixed points. We found candidates for one-parameter families of crossover scaling solutions within our truncation. The present numerical approximation to the flow equations for $u$ and $w$ is, however, not sufficient for the clarification if all candidates are really overall solutions of the corresponding system of differential equations for the scaling functions. Furthermore, it is not guaranteed that all members of such families "survive" extended truncations. This is demonstrated by a family of crossover solutions for $u_*(\tilde{\rho})$ obtained at constant $w$ (truncation 1), for which only particular members solved the combined system of differential equations for $u_*(\tilde{\rho})$ and $w_*(\tilde{\rho})$ (truncation 2);
5. Non-vanishing gauge couplings $g^2 > 0$ have the tendency to create a minimum of $u_*(\tilde{\rho})$ at $\tilde{\rho}_0 \neq 0$. This implies spontaneous symmetry breaking for the scaling solution. This tendency dominates if the non-minimal scalar-curvature coupling $\xi$ is small

enough. This observation is relevant for GUT models where certain scalar fields couple to the gauge bosons while no Yukawa couplings to the fermions are allowed. In particular, for a fixed point with $g_*^2 > 0$ [19,79] this could account for the partial symmetry breaking of the GUT-symmetry close to the Planck mass;

6. Non-zero Yukawa couplings $y^2 > 0$ have the opposite tendency of generating a minimum of $u_*(\tilde{\rho})$ at $\tilde{\rho} = 0$. The competition between gauge and Yukawa couplings is relevant both for the standard model and GUT models;

7. A non-minimal scalar-curvature coupling $\xi > 0$ favors a minimum of $u_*(\tilde{\rho})$ at $\tilde{\rho} = 0$ or at $\tilde{\rho} \to \infty$, depending on the particle content, see Figures 13, 17 and 20;

8. All scaling solutions or candidate scaling solutions that we have found so far for non-vanishing matter couplings lead in the Einstein frame to an effective scalar potential that vanishes precisely for large field values. This amounts to a dynamical solution of the cosmological constant problem via "runaway solutions" of the cosmological field equations for which scalar fields increase without bounds towards the infinite future. In particular, we found candidate scaling potentials with constant asymptotic behavior, $u_*(\tilde{\rho} \to \infty) = u_\infty$. Together with $w_*(\tilde{\rho} \to \infty) = \xi_\infty \tilde{\rho}/2$, this implies for the potential in the Einstein frame:

$$\frac{V_E(\tilde{\rho} \to \infty)}{\bar{M}^4} = \frac{u_\infty}{\xi_\infty^2 \tilde{\rho}^2} = \frac{u_\infty k^4}{\xi_\infty^2 \rho^2}. \tag{188}$$

Standard normalization of the scalar kinetic term in the Einstein frame replaces $k^4/\rho^2 \to \exp(-\alpha \varphi/\bar{M})$ [15];

9. A dynamical solution of the cosmological constant problem by the decrease of $V_E$ to zero for $\rho \to \infty$ within a "runaway cosmology" leads to dynamical dark energy or quintessence [15].

The runaway solution for the cosmological constant problem is a rather generic feature of scaling solutions, involving only that for $\tilde{\rho} \to \infty$, the coefficient $w_*^2(\tilde{\rho})$ increases faster than $u_*(\tilde{\rho})$. We have not investigated here another possible asymptotic behavior allowed by the graviton barrier [6], namely $u(\tilde{\rho} \to \infty) \sim \tilde{\rho}$, $v(\tilde{\rho} \to \infty) = u(\tilde{\rho} \to \infty)/w(\tilde{\rho} \to \infty) \leq 1$. We refer to [6,16] for a detailed discussion. Again, $V_E$ vanishes for $\rho \to \infty$.

### 9.5. Consequences for Particle Physics

Already at the present stage of the investigation it is apparent that the understanding of the scaling form of the effective potential has important consequences for particle physics and cosmology. The effective potential is the central ingredient for the phenomenon of spontaneous symmetry breaking as well as for inflation and dynamical dark energy. We highlight here three important consequences for the standard model coupled to quantum gravity:

1. For a minimal model of the standard model coupled to quantum gravity, the non-minimal Higgs-curvature coupling has to be small, $\xi_\infty < 10^{-4}$. Higgs inflation with a large coupling $\xi_\infty > 1$, as usually assumed, is not possible. The scaling potential $u_*(\tilde{\rho})$ and associated Einstein frame potential $V_E(h^\dagger h)$ does not allow for Higgs inflation. A positive $V_E$ for large $h^\dagger h$ could only be generated by the flow away from the scaling solution. Higgs inflation with very small $\xi_\infty$ [11] may not seem likely, but a detailed study of the flow away from the fixed point is necessary in order to settle this question. The minimal model may still permit Starobinsky inflation [75] if the coefficient of the squared curvature scalar $R^2$ if the effective action is large enough;

2. The ratio between the Higgs scalar mass $m_H$ and the top quark mass $m_t$ can be predicted [3]. The investigations of the present paper provide further evidence for the robustness and precision of this successful prediction. For the extended Reuter fixed point with constant scaling solution, the flow away from the fixed point affects the prediction of $m_H/m_t$ only very mildly. The non-minimal Higgs-curvature coupling $\xi$

generated by the flow due to non-zero gauge and Yukawa couplings is too small to affect the predicted value. Possible crossover scaling solutions , if established, could lead to somewhat larger $\zeta$. In the parameter region, where such crossover solutions are reasonable candidates for scaling solutions, the small value of $\zeta$ seems to have only a small effect on the ratio $m_H/m_t$. These findings reduce possible errors for the prediction arising from uncertainties in the physics near the Planck scale. With future possible precision measurements of the pole mass for the top quark it will become important to settle even the size of small effects for the predicted value $m_H/m_t$;

3.  Certain models could lead to a simultaneous prediction of the two ratios $m_H/m_t$ and $m_t/m_W$. This could be realized for scaling solutions for which both gauge and Yukawa couplings take non-zero values. The conditions for the existence of global scaling solutions may be strong enough to enforce for $\tilde{\rho} \approx 1$ both $u(\tilde{\rho})$ and $w(\tilde{\rho})$ to be very close to the constant scaling solution. In this case the ratio of the top quark mass to the W-boson mass would be an independent prediction.

*9.6. Implications for Cosmology*

Concerning cosmology, our approach offers new perspectives for the understanding of the very early and present epochs, and their possible connection. Typical cosmological solutions of the field equations derived from the effective action show a crossover between the region near the ultraviolet fixed point in the past, and the infrared region in the future [14]. This is realized by the cosmic evolution of scalar fields, which is associated to an increase of the dimensionless ration $\tilde{\rho} = \chi^2/(2k^2)$ from zero in the past to infinity in the future. The effective scalar potential $u(\tilde{\rho})$, or the associated potential in the Einstein frame $V_E(\chi)$, is the key quantity for the understanding of the cosmic evolution. The present work computes the scaling form $u_*(\tilde{\rho})$ of this effective potential for arbitrary $\tilde{\rho}$. This connects the region of small $\tilde{\rho}$ for very early cosmology to the region of large $\tilde{\rho}$ for the present cosmology.

The inflationary epoch in very early cosmology is related to the limit $\tilde{\rho} \to 0$. For fixed $\chi$, this corresponds to the ultraviolet limit $k \to \infty$. We find that for $\tilde{\rho} \to 0$ both $u(\tilde{\rho})$ and $w(\tilde{\rho})$ typically approach constant values $u_0$ and $w_0$. The indirect dependence on the scale $k$ through the dependence on $\tilde{\rho}$ disappears, expressing the ultraviolet fixed point behavior. Inflation corresponds to the vicinity of this fixed point, which is the origin of the almost scale invariant primordial fluctuation spectrum [11]. The potential in the Einstein frame approaches a constant for $\tilde{\rho} \to 0$, $V_E \to \bar{M}^4 u_0/(4w_0^2)$. In the vicinity of the fixed point for small $\tilde{\rho}$, the potential $V_E$ is almost flat, leading to slow roll inflation. In contrast to the usual approach of simply assuming a form of the potential, the present work computes the form of the effective potential as a result of the fluctuation effects in quantum gravity.

The present cosmological epoch is dominated by dark energy. Its dynamics is determined by the effective scalar potential for large $\tilde{\rho}$. For fixed $\chi$ this corresponds to the infrared limit $k \to 0$. All scaling solutions found so far lead for $\chi \to \infty$ to a potential in the Einstein frame which decreases to zero. No tuning of parameters is necessary for this property. This behavior of $V_E(\chi)$ solves the cosmological constant problem asymptotically, since $V_E$ vanishes in the infinite future for $\chi(t \to \infty) \to \infty$. At present, the universe is old, but not infinitely old. The potential $V_E$ has not yet reached zero and constitutes the dark energy density of the universe. The detailed dynamics of dark energy will become accessible once in addition to $u(\tilde{\rho})$ and $w(\tilde{\rho})$, as well as the $\tilde{\rho}$-dependent coefficient of the scalar kinetic term is computed.

By computing the effective scalar potential for all values of $\tilde{\rho}$ we connect inflation and quintessence. Both may be due to the same scalar field, the cosmon. The different characteristics of very early and very late cosmology simply correspond to different regions in the potential $u(\tilde{\rho})$

*9.7. Fundamental Scale Invariance*

On the conceptual side, an important finding of the present work is the observation that there is no need for a flow away from the scaling solution in order to obtain agreement with

observations. For functions depending only on dimensionless ratios as $\tilde{\rho}$, the limit $k \to 0$ can equivalently be obtained by $\chi \to \infty$. For all $\chi \neq 0$, scale symmetry is spontaneously broken, resulting in massive particles. For scaling solutions describing a crossover between an ultraviolet and an infrared fixpoint, a given model has only to specify which value of $\tilde{\rho}$ corresponds to the present cosmological time.

One may entertain the hypothesis that our world is precisely described by a scaling solution. No parameter the dimension of length or mass that enters the effective action in this case – this is fundamental scale invariance. As a consequence, the relevant parameters for the flow away from the scaling solution play no role. This scenario is very predictive, since relevant parameters are no longer available as undetermined quantities. They are all set to zero. If the scaling solution is unique, this scenario contains no free dimensionless parameters. Free parameters are only possible if families of scaling solutions exist.

The present investigation of a fixed point with non-constant scaling functions and/or non-zero gauge and Yukawa couplings leaves many questions open. Already now it shows that simple extrapolations of perturbative features, as approximately polynomial potentials, to the quantum gravity regime are not correct. It will be necessary to gain further understanding and intuition for central quantities as the effective potential in the quantum gravity regime. Features that seem "unnatural" from a perturbative point of view, as the gauge hierarchy or the tiny value of dark energy, may find explanations in the genuinely non-perturbative setting of quantum gravity.

**Funding:** This research received no external funding.

**Institutional Review Board Statement:** Not applicable.

**Informed Consent Statement:** Not applicable.

**Data Availability Statement:** Not applicable.

**Acknowledgments:** Not applicable.

**Conflicts of Interest:** There is no conflict of interest.

**Appendix A. Calculation Flow**

Since the content of this work is rather extended, it may be convenient for the reader to find a short summary of the flow of the calculations. The basic ansatz (9) for the effective action contains up to two derivatives. Correspondingly, we evaluate the flow equations in a derivative expansion with up to two derivatives.

The central flow equation for the potential $U$ is Equation (10), which is supplemented by the flow equation for the coefficient of the curvature scalar $F$ in Section 7. We do not include a computation of the flow of the wave function re-normalization factors $Z_a$ in the present paper. We concentrate on the flow equations for the dimensionless ratios (13) $u = U/k^4$ and $w = F/2k^2$. A summary of approximated flow equations for $u$ and $w$ is given by Equations (90)–(92). The effective particle numbers in Equation (92) contain treshold functions which account for the effective decoupling of massive particles, as given by Equation (93). The effective non-minimal scalar coupling $\tilde{\xi}$ is defined in Equations (95) and (96). For given fixed gauge couplings $g$ and Yukawa couplings $y$, the flow equations for $u$ and $w$ are self-contained. The corresponding differential equations defining the scaling solutions are Equations (98)–(100).

Several approximations are made in order to arrive at the rather simple system of Equations (90)–(92) and (98)–(100). They are discussed in more detail in the text, but their precise understanding is not crucial for the main outcomes of the present paper.

For a better understanding of the main mechanisms, the present paper proceeds by approximations that are extended step by step. In Sections 2–5 we consider constant $w$, while in Section 6 we investigate a two-parameter approximation $w = w_0 + \xi \tilde{\rho}/2$. The full coupled flow equations for $u$ and $w$ are discussed in Sections 7 and 8.

**Appendix B. General Scaling Solutions for Matter Freedom**

In this appendix we investigate general properties of scaling solutions for matter freedom. We start by solving Equation (35) in the vicinity of the constant scaling solution. We consider here for the constant solution the vicinity of $\tilde{\rho} = 0$, such that $A$ corresponds to $A_0$, as given by Equation (37) for the values $v_0$, $w_0$ of the constant scaling solution. The general solution of Equation (35) reads:

$$\Delta u(\tilde{\rho}) = c_0 \tilde{\rho}^{\frac{4-A_0}{2}}, \tag{A1}$$

involving $c_0$ as a free integration constant. For all $A_0 < 4$ the general scaling solution indeed obeys $\Delta u(\tilde{\rho} \to 0) \to 0$, such that $u(\tilde{\rho} \to 0) \to u_0$. The quartic coupling, defined by:

$$\lambda(\tilde{\rho}) = \frac{\partial^2 u}{\partial \tilde{\rho}^2}, \tag{A2}$$

diverges, however, for $c_0 \neq 0$ according to:

$$\lambda = c_0 \left(2 - \frac{A_0}{2}\right)\left(1 - \frac{A_0}{2}\right)\tilde{\rho}^{-A_0}. \tag{A3}$$

This holds except for the special case $A_0 = 2$. For a diverging $\lambda$ our approximation of neglecting the quartic scalar coupling and mass term, which leads to constant $b_U$ in Equation (26), no longer holds. The dimensionless scalar mass term:

$$\tilde{m}^2 = \frac{\partial u}{\partial \tilde{\rho}} \tag{A4}$$

behaves for $\tilde{\rho} \to 0$ as:

$$\tilde{m}^2 = c_0 \left(2 - \frac{A_0}{2}\right)\tilde{\rho}^{1 - \frac{A_0}{2}}. \tag{A5}$$

For $c_0 \neq 0$ it diverges for $A_0 > 2$, and vanishes for $A_0 < 2$. In Appendix C.2 we include effects of nonzero $\tilde{m}^2$ and $\lambda$ in the flow equations. This cures the divergence of $\tilde{\lambda}$ (and possibly $\tilde{m}^2$) for $\tilde{\rho} \to 0$.

There is a particular situation for which the whole crossover can be described within the validity of the matter freedom approximation. In our truncation this occurs for the choice $w_0 = w_0^{(2)}$ for which $A(w_0^{(2)}) = 2$. Then:

$$\Delta u = c_0 \tilde{\rho} \tag{A6}$$

leads for $\tilde{\rho} \to 0$ to $\tilde{m}_0^2 = c_0$ and $\lambda_0 = 0$. For small $c_0$ matter freedom remains valid for arbitrarily small $\tilde{\rho}$.

There is a critical value $w_c$ for which the two constant solutions with $v_+$ and $v_-$ merge. For $N > -4$, it corresponds to the minimum of the curves in Figure 1. For approximation (30), this happens if the square root vanishes:

$$(1 - (N_0 - 4)z_c)^2 = 32 z_c. \tag{A7}$$

For an appropriate range of $w_0$ the generic flow equation (24) has for $\tilde{\rho}$-independent $u$ two fixed points where $\beta_u = 0$. They merge at the critical $w_c$ for which $\partial \beta_u / \partial u$ and $\beta_u$ vanish simultaneously. At this merging point one has $\partial c_U / \partial u = 4$ and therefore:

$$A(w_c) = 4. \tag{A8}$$

This can be seen in Figure 4. At $v_c = v(w_c)$, one finds indeed $A(v_c) = 4$.

For $w < w_c$ no fixed point remains. For $w > w_c$, the solution $v_+$ has $A > 4$, while for the solution $v_-$ one finds $A < 4$. This follows directly from the monotonic behavior of $A(v)$. From Equation (A1) we conclude that the fixed point $v_-$ is attractive for $\tilde{\rho} \to 0$, while $v_+$ is repulsive.

Consider next the behavior of the scaling solution for $\tilde{\rho} \to \infty$. For a valid scaling solution one needs $v(\tilde{\rho}) < 1$ for all $\tilde{\rho}$, since the pole of $\beta_u$ for $v = 1$ should not be crossed. In addition, negative $v$ cannot diverge for $\tilde{\rho} \to \infty$ – such a behavior would lead to an unbounded effective potential which is not compatible with a stable theory. For all scaling solutions one requires a finite constant $v_\infty$:

$$\lim_{\tilde{\rho} \to \infty} v(\tilde{\rho}) = v_\infty \,. \tag{A9}$$

In turn, this enforces:

$$\lim_{\tilde{\rho} \to \infty} \left( \tilde{\rho} \frac{\partial v}{\partial \tilde{\rho}} \right) = 0 \,, \tag{A10}$$

since otherwise $v(\tilde{\rho})$ would diverge for $\tilde{\rho} \to \infty$. With Equation (A10) the computation of $v_\infty$ is the same as for $v_0$, with possible solutions $v_+$ and $v_-$ given by Equation (30). In addition, the discussion of solutions that approach $u_\infty$ for $\tilde{\rho} \to \infty$ remains unchanged, with $A_0$ and $c_0$ in Equation (A1) replaced by $A_\infty$ and $c_\infty$. The constant $u_\infty$ is approached by neighboring solutions for $\tilde{\rho} \to \infty$ if $A_\infty > 4$. While $v_-$ is attractive for $\tilde{\rho} \to 0$, $v_+$ is attractive for $\tilde{\rho} \to \infty$. Taking things together, the general scaling solution for matter freedom is a crossover from the fixed point $v_-$ for $\tilde{\rho} \to 0$ to the fixed point $v_+$ for $\tilde{\rho} \to \infty$.

**Appendix C. Scalar Mass Term and Quartic Coupling**

The crossover solution for matter freedom discussed in Section 3 is not self-consistent except for $A_0 = 2$. The approximation to the scalar fluctuation contribution $\tilde{\pi}_s$, which neglects in Equation (15) the scalar mass term $\tilde{m}^2$, breaks down for $\tilde{\rho} \to 0$. For $A_0 > 2$, the mass term diverges for $\tilde{\rho} \to 0$, see Equation (A5). On the other hand, for $A_0 < 2$ the potential according to matter domination decreases faster than $\tilde{\rho}$ for $\tilde{\rho} \to 0$. In this case, the role of a neglected non-zero mass term would dominate. Furthermore, according to Equation (A3) both $\tilde{\rho} \partial \lambda / \partial \tilde{\rho}$ and $\lambda$ would diverge for $\tilde{\rho} \to 0$. We conclude that in the region $\tilde{\rho} \to 0$, matter freedom or the approximation of the flow contribution from scalars by a constant is no longer valid. We can no longer neglect $\tilde{m}_A^2$ in Equation (15). Taking the effects of $\tilde{m}^2 \neq 0$ into account will lead to corrections for a small range of $\tilde{\rho}$ near zero. In this appendix we will keep the approximation of constant $w(\tilde{\rho}) = w_0$ and vanishing gauge and Yukawa couplings. We include now on the r. h. s. of the flow equation the effects of $\tilde{\rho}$-derivatives of $u(\tilde{\rho})$.

*Appendix C.1. Flow Equation*

Let us consider a single real scalar field with discrete symmetry $\phi \to -\phi$ and $\rho = \phi^2/2$. Neglecting in Equations (15) and (18) the anomalous dimension, $\eta_A = 0$, it contributes to the flow of $u$ by a term:

$$(\partial_t u)_s = \frac{1}{32\pi^2} \left( 1 + \tilde{m}^2 + 2\tilde{\rho}\lambda \right)^{-1} \,. \tag{A11}$$

Here we employ for the squared scalar mass term:

$$\tilde{m}_A(\tilde{\rho}) = \frac{\partial u}{\partial \tilde{\rho}} + 2\tilde{\rho} \frac{\partial^2 u}{\partial \tilde{\rho}^2} = \tilde{m}^2(\rho) + 2\tilde{\rho}\lambda(\tilde{\rho}) \,. \tag{A12}$$

The difference between the expression (A11) and the value for $\tilde{m}^2 = 0$, $\lambda = 0$, which is already incorporated in $b_U$, yields to the flow equation for $u$ an additional contribution:

$$\Delta_s \partial_t u = \frac{\Delta \pi_s}{k^4} = 4c_{U,s} = \frac{1}{32\pi^2}\left[\frac{1}{1 + \tilde{m}^2 + 2\tilde{\rho}\lambda} - 1\right]$$
$$= -\frac{1}{32\pi^2}\frac{\tilde{m}^2 + 2\tilde{\rho}\lambda}{1 + \tilde{m}^2 + 2\tilde{\rho}\lambda}. \tag{A13}$$

The particular fixed point solution $u(\tilde{\rho}) = u_0 = v_- w$ corresponds to $\tilde{m}^2 = 0$, $\lambda = 0$. It is not changed by the additional term (A13). While the particular constant scaling solutions remain valid in the presence of $\Delta \pi_s$, this does not hold for the generic crossover scaling solutions for matter freedom. For the latter, $\Delta \pi_s$ will induce substantial corrections for $\tilde{\rho}$-derivatives of $u$ in the region of small $\tilde{\rho}$.

The correction to the flow of the mass term is given by the $\tilde{\rho}$-derivative of Equation (A13). Combining with Equation (40) one has:

$$\partial_t \tilde{m}^2 = (A - 2)\tilde{m}^2 + 2\tilde{\rho}\lambda - \frac{3\lambda + 2\tilde{\rho}\,\partial\lambda/\partial\tilde{\rho}}{32\pi^2(1 + \tilde{m}^2 + 2\tilde{\rho}\lambda)^2}. \tag{A14}$$

Possible scaling solutions for $\tilde{m}^2$ obtain if the r.h.s. of Equation (A14) vanishes. With $\lambda = \partial\tilde{m}^2/\partial\tilde{\rho}$ this condition amounts to a differential equation involving up to two $\tilde{\rho}$-derivatives of $\tilde{m}^2$.

In the presence of $\Delta \pi_s$, the structure of the differential equation for the scaling solutions for $u$ gets more complicated since it involves up to two $\tilde{\rho}$-derivatives of $u$. Denoting by $u' = \partial_{\tilde{\rho}} u$, $u'' = \partial_{\tilde{\rho}}^2 u$ the scaling solutions have to obey the differential equation:

$$2\tilde{\rho}u' = 4u - \frac{1}{24\pi^2}\left(\frac{5}{1 - u/w} + \frac{1}{1 - u/4w}\right)$$
$$- \frac{N - 5}{32\pi^2} - \frac{1}{32\pi^2(1 + u' + 2\tilde{\rho}u'')}. \tag{A15}$$

For constant $w = w_0$, this is a nonlinear second order differential equation. The general solution involves two free integration constants, say $u(\tilde{\rho}_0)$ and $u'(\tilde{\rho}_0)$ at some suitably chosen $\tilde{\rho}_0$. The question is which one of the local solutions remains valid for the whole range of $\tilde{\rho}$ without encountering a singularity. Finding the valid scaling solutions is not a simple task since the generic local solutions become singular at some $\tilde{\rho}$. The two constant scaling solutions are regular solutions of Equation (A15) for the whole range of $\tilde{\rho}$. One would like to know if a one-parameter family of regular solutions can be found that replaces the one-parameter family of crossover solutions for the approximation of matter freedom.

A similar question arises for extensions to nonvanishing gauge couplings, Yukawa couplings or nonminimal scalar-gravity couplings that we will discuss in the following sections. Omitting $\Delta \pi_s$ we will find one-parameter families of scaling solutions, similar to the crossover scaling solutions for matter freedom. Again the question will arise if a one-parameter family of regular solutions persists in the presence of $\Delta \pi_s$. It is therefore worthwhile to discuss this question in some detail.

At this point we should emphasize that even for the existence of a one-parameter family of regular scaling solutions for $u(\tilde{\rho})$ for a given $w(\tilde{\rho})$, there is no guarantee that full quantum gravity admits a one-parameter family of scaling solutions. It is possible that the flow equations of $w(\tilde{\rho})$ (and other invariants not considered here) are compatible only with a subclass of the regular scaling solutions for $u(\tilde{\rho})$. It is well conceivable that the fixed point for quantum gravity only admits a single solution. The present investigation should therefore be seen as an exploration of possibilities rather than a definite determination of the scaling solution. On the other hand, any overall scaling solution of quantum gravity

has to result in a regular scaling solution for $u(\tilde{\rho})$ for all other couplings taking their values for the scaling solution. The general properties of $u(\tilde{\rho})$ found in this paper therefore apply to any overall scaling solution for quantum gravity.

*Appendix C.2. Scaling Solution Near the Origin*

If $\tilde{\rho}\,\partial\lambda/\partial\tilde{\rho}$ vanishes for $\tilde{\rho} \to 0$, e.g., for finite $\lambda(\tilde{\rho} = 0)$, the fixed point value of the mass term at $\tilde{\rho} = 0$, $\tilde{m}_0^2 = \tilde{m}^2(\tilde{\rho} = 0)$, obeys for $A_0 \neq 2$:

$$\tilde{m}_0^2 = \frac{3\lambda_0}{32\pi^2(A_0 - 2)(1 + \tilde{m}_0^2)}. \tag{A16}$$

It vanishes only for $\lambda_0 = \lambda(\tilde{\rho} = 0) = 0$. We may use $\tilde{m}_0^2$ as a parameter to characterize a possible family of scaling solutions. For a given $\tilde{m}_0^2$, the quartic coupling $\lambda_0$ is fixed and computable. This continues to higher order couplings. The scaling solution for $\lambda_0$ fixes $\partial\lambda/\partial\rho(\tilde{\rho} = 0)$ and so on. The particular solution $\tilde{m}_0^2 = 0$ is the flat scaling solution.

For a non-zero scalar mass term the approach of a general scaling solution to $u(\tilde{\rho} = 0) = u_0$ gets an additional contribution, modifying Equation (35) to:

$$2\tilde{\rho}\,\frac{\partial\Delta u}{\partial\tilde{\rho}} = (4 - A)\Delta u - 4\,c_{\text{U},s}. \tag{A17}$$

Let us consider the vicinity of the constant scaling solution. Employing:

$$\tilde{m}^2 = \frac{\partial\Delta u}{\partial\tilde{\rho}}, \tag{A18}$$

we may linearize in $\tilde{m}^2$ and $\lambda$:

$$\left(2\tilde{\rho} - \frac{1}{32\pi^2}\right)\frac{\partial\Delta u}{\partial\tilde{\rho}} = (4 - A)\Delta u + \frac{\lambda\tilde{\rho}}{16\pi^2}. \tag{A19}$$

We first consider the limit $\tilde{\rho} \to 0$ where we neglect the last term $\sim \lambda\tilde{\rho}$ in Equation (A19). The solution,

$$\Delta u = \tilde{c}_0\left(\frac{1}{64\pi^2} - \tilde{\rho}\right)^{2 - \frac{A_0}{2}}, \tag{A20}$$

deviates substantially from Equation (A1) in the range of small $\tilde{\rho}$. Derivatives no longer diverge. For $\tilde{\rho} \to 0$ one finds finite $\tilde{m}^2$,

$$\tilde{m}^2 = -\tilde{c}_0\left(2 - \frac{A_0}{2}\right)\left(\frac{1}{64\pi^2} - \tilde{\rho}\right)^{1 - \frac{A_0}{2}}. \tag{A21}$$

The constant $\tilde{c}_0$ is related to $\tilde{m}_0^2 = \tilde{m}^2(\tilde{\rho} = 0)$ in Equation (A16) in the limit of small $\tilde{m}_0^2$.

For higher derivatives we have to take the term $\sim \lambda\tilde{\rho}$ in Equation (A19) into account. We make the ansatz:

$$\frac{\lambda}{32\pi^2} = f\tilde{m}^2. \tag{A22}$$

The resulting differential equation:

$$\left(\tilde{\rho}(1 - f) - \frac{1}{64\pi^2}\right)\frac{\partial\Delta u}{\partial\tilde{\rho}} = \left(2 - \frac{A_0}{2}\right)\Delta u \tag{A23}$$

has the solution:

$$\Delta u = c_0\left(\frac{1}{64\pi^2} - (1 - f)\tilde{\rho}\right)^{\frac{4 - A_0}{2(1 - f)}}. \tag{A24}$$

Identification with Equation (A20) for $\tilde{\rho} = 0$ relates $c_0$ and $\tilde{c}_0$:

$$c_0 = \tilde{c}_0 \left( \frac{1}{64\pi^2} \right)^{-\frac{f(4-A_0)}{2(1-f)}} . \tag{A25}$$

Taking a derivative of Equation (A24),

$$\tilde{m}^2 = \frac{\partial \Delta u}{\partial \tilde{\rho}} = -c_0 \frac{4 - A_0}{2} \left( \frac{1}{64\pi^2} - (1-f)\tilde{\rho} \right)^{\frac{4-A_0}{2(1-f)} - 1} , \tag{A26}$$

one observes that $\tilde{m}^2(\tilde{\rho} = 0)$ indeed coincides with the value (A21). Taking a further $\tilde{\rho}$-derivative of Equation (A26) and evaluating it at $\tilde{\rho} = 0$, one finds for $\tilde{\rho} \to 0$ the relation:

$$\frac{\lambda}{\tilde{m}^2} = 32\pi^2 (A_0 - 2 - 2f) . \tag{A27}$$

Comparison with Equation (A22) leads to a self-consistent determination of $f$,

$$f = \frac{A_0 - 2}{3} . \tag{A28}$$

This corresponds to Equation (A16) for small $\tilde{m}^2$. We infer for the limiting behavior of $u$ for $\tilde{\rho} \to 0$, $u_0 = v_- w$:

$$u(\tilde{\rho} \to 0) = u_0 + c_0 \left( \frac{1}{64\pi^2} - \frac{5 - A_0}{3}\tilde{\rho} \right)^{\alpha} \tag{A29}$$

with

$$\alpha = \frac{3(4 - A_0)}{2(5 - A_0)} . \tag{A30}$$

We conclude that for $\tilde{\rho} \to 0$ the inclusion of the correction term $\Delta\pi_s$ cures the divergence of derivatives of the crossover scaling solution. We will next establish that the fixed point solution is now compatible with the flow of couplings at fixed $\tilde{\rho}$.

*Appendix C.3. Relevant and Irrelevant Couplings*

For the flow of $\tilde{m}^2$ and $\lambda$ away from the fixed point we employ the $\tilde{\rho}$-derivative of Equation (A13):

$$\partial_t \tilde{m}^2 = (A - 2)\,\tilde{m}^2 + 2\tilde{\rho}\lambda - \frac{3\lambda + 2\tilde{\rho}\,u^{(3)}}{32\pi^2\,(1 + \tilde{m}^2 + 2\tilde{\rho}\lambda)^2} , \tag{A31}$$

with

$$u^{(3)}(\tilde{\rho}) = \frac{\partial\lambda(\tilde{\rho})}{\partial\tilde{\rho}} . \tag{A32}$$

For $\tilde{\rho} \to 0$, we take advantage of the finiteness of $\tilde{m}^2$, $\lambda$ and $u^{(3)}$ and evaluate the flow of $\tilde{m}_0^2 = \tilde{m}^2(\tilde{\rho} = 0)$, $\lambda_0 = \lambda(\tilde{\rho} = 0)$,

$$\partial_t \tilde{m}_0^2 = (A_0 - 2)\tilde{m}_0^2 - \frac{3\lambda_0}{32\pi^2(1 + m_0^2)^2} . \tag{A33}$$

For the fixed point we find the relation:

$$\lambda_{0,*} = \frac{32\pi^2(A_0 - 2)}{3}\,\tilde{m}_{0,*}^2 \left(1 + \tilde{m}_{0,*}^2\right)^2 . \tag{A34}$$

Linearizing for small $\lambda$ and $\tilde{m}^2$ this fixed point for the ratio $\lambda/\tilde{m}^2$ is indeed consistent with Equations (A22) and (A26). Taking a $\tilde{\rho}$-derivative of Equation (A31) and evaluating it at $\tilde{\rho} = 0$ yields $u_{0,*}^{(3)}$ in terms of $\tilde{m}_{0,*}^2$ and $\lambda_{0,*}$.

Deviations from $\tilde{m}_0^2$ from the fixed point value $\tilde{m}_{0,*}^2$ are denoted by

$$\gamma = \tilde{m}_0^2 - \tilde{m}_{0,*}^2 \,. \tag{A35}$$

For small $\gamma$ the flow equations read,

$$\partial_t \gamma = (A_0 - 2)\gamma + \frac{3\lambda_{0,*}\gamma}{32\pi^2 \left(1 + \tilde{m}_{0,*}^2\right)^2}$$

$$- \frac{3\,\delta\lambda}{32\pi^2 \left(1 + \tilde{m}_{0,*}^2\right)^2} + \frac{\tilde{m}_{0,*}^2}{w_0}\frac{\partial A}{\partial v}\,\delta u \,, \tag{A36}$$

with

$$\delta\lambda = \lambda_0 - \lambda_{0,*}\,, \quad \delta u = u_0 - u_{0,*}\,. \tag{A37}$$

Neglecting first $\delta\lambda$ and $\delta u$, the insertion of Equation (A34) yields:

$$\partial_t \gamma = (A_0 - 2)\left(1 + \frac{\tilde{m}_{0,*}^2}{\left(1 + \tilde{m}_{0,*}^2\right)^2}\right)\gamma = -\theta_\gamma \gamma\,. \tag{A38}$$

For a second order phase transition the flow of $\gamma$ must vanish for $\gamma = 0$ [80–86]. This ensures naturalness of very small $\gamma$ due to the enhanced quantum scale symmetry for $\gamma = 0$. This generalizes to more complex settings. The vacuum electroweak phase transition is almost of second order.

For $A_0 < 2$ the deviation from the scaling solution $\gamma$ is a relevant parameter. The critical exponent $\theta_\gamma$ is positive. For constant $\theta_\gamma$ the flow of $\gamma$ obeys:

$$\gamma = \gamma_\Lambda \left(\frac{k}{\Lambda}\right)^{-\theta_\gamma}\,. \tag{A39}$$

The initial value $\gamma_\Lambda = \gamma(k = \Lambda)$ specified at some arbitrary chosen scale $\Lambda$ determines the particular flow trajectory. Restoring dimensions, the renormalized scalar mass term at $\rho = 0$ behaves as:

$$m^2(k) = \tilde{m}_{0,*}^2 k^2 + \gamma_\Lambda \Lambda^{\theta_\gamma} k^{2-\theta_\gamma}\,. \tag{A40}$$

For $\gamma_\Lambda > 0$, the mass term is positive for all $k$. The origin $\rho = 0$ is a local minimum of the effective potential. The model is typically in the "symmetric phase" with unbroken $\mathbb{Z}_2$ symmetry $\phi \to -\phi$. In contrast, for $\gamma_\Lambda < 0$ the mass term turns negative for $k$ smaller than some critical $k_c$:

$$k_c = \Lambda \left(-\frac{\gamma_\Lambda}{\tilde{m}_{0,*}^2}\right)^{\frac{1}{\theta_\gamma}}\,. \tag{A41}$$

The origin at $\rho = 0$ becomes a local maximum for small enough $k$. This indicates a minimum of $U$ for $\rho = \rho_0 > 0$, and therefore a spontaneous breaking of the $\mathbb{Z}_2$-symmetry. The phase transition at the boundary between the symmetric phase and the phase with spontaneous symmetry breaking is given by $\gamma_\Lambda = 0$. This is the scaling solution. The existence of a scaling solution is a general feature of a second order phase transition.

We observe that the critical trajectory $\gamma_\Lambda = 0$ is never crossed by any flow trajectory. This generalizes to the critical surface of a second order phase transition. It follows generally from the existence of a scaling solution and continuity. For the scaling solution,

corresponding to the critical surface, the flow vanishes. For neighboring solutions the flow is very small by continuity, as exemplified by the flow of $\tilde{m}_0^2$ in the region where $\gamma$ remains small. Since the flow becomes arbitrarily slow if the scaling solution is approached arbitrarily close by, no flow trajectory can cross the critical surface.

For $\theta_\gamma < 0$, typically requiring $A_0 > 2$, the deviation $\gamma$ from the critical surface becomes an irrelevant parameter. One observes self-tuned criticality. For the electroweak phase transition this can explain the gauge hierarchy [87,88].

These general statements apply to the flow according to Equation (A36) for nonvanishing $\delta u$ and $\delta \lambda$ as well. The critical surface is now a two-dimensional hypersurface in the three-dimensional space of couplings $u_0$, $\tilde{m}_0^2$, and $\lambda_0$. On the critical surface, the couplings follow scaling solutions $u_{0,*}$, $\tilde{m}_{0,*}^2$, and $\lambda_{0,*}$ . Deviations from the scaling solution may be denoted by $\alpha_i = (\delta u, \gamma, \delta \lambda)$. For the scaling solution the flow of $\delta u$, $\gamma$, and $\delta \lambda$ vanishes. Critical exponents are the eigenvalues of the stability matrix $T$ which characterizes the linearized flow in the vicinity of the scaling solution:

$$\partial_t \alpha_i = -T_{ij} \alpha_j .\tag{A42}$$

For a computation of the stability matrix we need the flow of $\delta u$ and $\delta \lambda$. For $\delta u$ one finds from Equations (24) and (A11):

$$\partial_t \delta u = (A - 4)\delta u + 2\tilde{\rho}\gamma - \frac{1}{32\pi^2}\left(1 + \tilde{m}^2 + 2\tilde{\rho}\lambda\right)^{-2}(\gamma + 2\tilde{\rho}\delta\lambda) .\tag{A43}$$

For the flow of the quartic coupling we first take a $\tilde{\rho}$-derivative of Equation (A31):

$$\partial_t \lambda = A\lambda + 2\tilde{\rho}u^{(3)} + \frac{1}{w}\frac{\partial A}{\partial v}\tilde{m}^4 + \frac{1}{16\pi^2}\frac{\left(3\lambda + 2\tilde{\rho}u^{(3)}\right)^2}{(1 + \tilde{m}^2 + 2\tilde{\rho}\lambda)^3}$$
$$- \frac{1}{32\pi^2}\frac{5u^{(3)} + 2\tilde{\rho}u^{(4)}}{(1 + \tilde{m}^2 + 2\tilde{\rho}\lambda)^2} .\tag{A44}$$

Neglecting $u^{(3)}$ and $u^{(4)}$ the linearized flow equation for $\delta\lambda$ becomes:

$$\partial_t \delta\lambda = A\delta\lambda + \frac{1}{w}\frac{\partial A}{\partial v}\left(\lambda_* \delta u + 2\tilde{m}_*^2 \gamma\right)$$
$$+ \frac{1}{w^2}\frac{\partial^2 A}{\partial v^2}\tilde{m}_*^4 \delta u + \frac{9\lambda_* \delta\lambda}{8\pi^2(1 + \tilde{m}_*^2 + 2\tilde{\rho}\lambda_*)^3}$$
$$- \frac{27\lambda_*(\gamma + 2\tilde{\rho}\delta\lambda)}{16\pi^2(1 + \tilde{m}_*^2 + 2\tilde{\rho}\lambda_*)^4} .\tag{A45}$$

The stability matrix for $\tilde{\rho} = 0$ reads:

$$-T = \begin{pmatrix} A - 4 & -d & 0 \\ B\tilde{m}_*^2 & A - 2 + 3d\lambda_* & -3d \\ B\lambda_* + C\tilde{m}_*^4 & 2B\tilde{m}_*^2 - \frac{54d\lambda_*^2}{(1+\tilde{m}_*^2)^2} & A + \frac{36d\lambda_*}{1+\tilde{m}_*^2} \end{pmatrix} ,\tag{A46}$$

with

$$B = \frac{1}{w}\frac{\partial A}{\partial v} , \quad C = \frac{1}{w^2}\frac{\partial^2 A}{\partial v^2} , \quad d = \frac{1}{32\pi^2(1 + \tilde{m}_*^2)^2} .\tag{A47}$$

For $A$ of the order one and $3d$ smaller than 0.01, the off-diagonal elements in the upper right corner are small. For $d = 0$, the eigenvalues of $T$ are given by the diagonal elements. In addition, for small $\tilde{m}_*^2$, $\lambda_*$ the off-diagonal elements in the lower left corner are all small. Neglecting them, the eigenvalues are given by the diagonal elements even for $d \neq 0$. We conclude that to a very good approximation the critical exponents are given by the diagonal

elements of $T$. Since $A_0 < 4$, there is always one relevant coupling, which is dominantly given by $\delta u$. A second coupling, dominantly $\gamma$, occurs for $A_0 < 2 - 3d\,\lambda_*$. This coupling becomes irrelevant for $A_0 > 2 - 3d\,\lambda_*$. The third coupling, dominantly $\delta\lambda$, is irrelevant.

If we keep also $\Delta u^{(3)}$ etc., the flow for a finite number of couplings will not be closed. In principle, the number of couplings is infinite, such that the stability matrix $T$ is infinite dimensional. The almost diagonal structure of $T$ continues if we extend it to higher order couplings. The critical exponent for $\delta u^{(3)}$ is approximately $A + 2$, while for $\Delta u^{(4)}$ it amounts to $A + 4$. Higher order couplings are all irrelevant parameters. In the absence of off-diagonal elements they would be predicted to take exactly the values for the scaling solution. In the presence of the off-diagonal elements the values of $\delta\lambda$, $\delta u^{(3)}$ etc. remain predictable as a function of the relevant couplings that we may parameterize by $\delta u$ (and $\gamma$ if this is also relevant).

The flow away from the critical surface is determined by the relevant couplings. These are linear combinations of $\alpha_i$ that are eigenvectors to positive eigenvalues of $T$. The irrelevant couplings correspond to negative eigenvalues of $T$. They can be set to zero. In consequence, $\delta\lambda$ can always be expressed as a linear combination of $\gamma$ and $\delta u$. As an example, we take the flat scaling solution for which $\tilde{m}_{0,*}^2 = 0$, and therefore also $\lambda_{0,*} = 0$, as well as $u_{0,*}^{(3)} = 0$ and similar for higher order couplings.

From,

$$\partial_t \delta\lambda = A_0 \delta\lambda \tag{A48}$$

we conclude that $\delta\lambda$ is an irrelevant coupling. This coincides with the findings of some of the early investigations [22,27,28,32,89,90]. Setting $\delta\lambda = 0$ in the flow equation for $\gamma$ and $\delta u$ results in:

$$\partial_t \gamma = (A_0 - 2)\gamma, \quad \partial_t \delta u_0 = (A_0 - 4)\delta u_0 - \frac{1}{32\pi^2}\gamma. \tag{A49}$$

The eigenvectors of $T$ are $\gamma$ and $\delta u_0 + \gamma/(64\pi^2)$, with eigenvalues $2 - A_0$ and $4 - A_0$.

*Appendix C.4. Asymptotic Behavior for Large Scalar Fields*

For large values $\tilde{\rho} \to \infty$ the $\tilde{\rho}$-dependence of the scaling effective potential is given by Equations (43)–(45). With $A_\infty > 4$ both $\tilde{m}^2(\tilde{\rho})$ and $\lambda(\tilde{\rho})$ vanish for $\tilde{\rho} \to \infty$. We conclude that the contribution (A13) becomes negligible for large $\tilde{\rho}$ for the scaling solution and the vicinity of it. The scalar field is free and massless in the range of very large $\tilde{\rho}$. For deviations from the scaling solution, the stability matrix has vanishing off-diagonal elements in the lower left corner. The eigenvalues are the diagonal elements, with $A_\infty - 4$, $A_\infty - 2$, $A_\infty$, … all positive for $A_\infty > 4$. There are no relevant couplings in the scalar sector. As long as the dimensionless Planck mass $w$ can be approximated by a constant, the scalar potential for large $\tilde{\rho}$ is predicted to be given precisely by the scaling solution.

The family of possible scaling solutions can be parameterized by the coefficient $c_\infty$ in Equations (43)–(45). It specifies at which $\tilde{\rho}_t$ the crossover from the vicinity of the constant fixed point potential for $\tilde{\rho} \to \infty$ to the fixed point potential for $\tilde{\rho} \to 0$ takes place, see Figures 2 and 3. In the large $\tilde{\rho}$-region where $\tilde{m}^2$ and $\lambda$ can be neglected the differential Equation (26) for the scaling solution for $u$ does not involve $x = \ln(\tilde{\rho})$ explicitly:

$$\frac{\partial u}{\partial x} = 2(u - c_U(u)). \tag{A50}$$

In this range the family of scaling solutions can be obtained by constant shifts in $x$ or multiplicative rescalings of $\tilde{\rho}$. The family of possible scaling solutions is characterized by a single parameter $c_\infty$. This extends to the full characterization of scaling solutions in the whole range of $\tilde{\rho}$, provided that for each $c_\infty$ the solution can be continued to $\tilde{\rho} \to 0$. If a certain range of $c_\infty$ cannot be continued, the allowed family of scaling solutions will be restricted to a range in $c_\infty$.

*Appendix C.5. Transition Region*

So far we have found a family of local scaling solutions in the region $\tilde{\rho} \to 0$, parametrized by $\tilde{m}_0^2$, as well as a family of local scaling solutions in the range $\tilde{\rho} \to \infty$, parametrized by $c_\infty$. The question arises which ones of this solutions can be combined into a global solution for the whole range of $\tilde{\rho}$. This will decide if the differential Equation (A15) has a whole family of global crossover solutions, only a discrete number of crossover solutions, or no crossover solution. In the last case only the constant solutions remain as global scaling solutions. The matching between the two limiting regions occurs in a transition region near $\tilde{\rho}_s = 1/(64\pi^2)$.

One could attempt a numerical solution of the second order differential Equation ( A15). This is intricate due to the appearance of the highest derivative in the denominator of a rather small correction term. The situation becomes numerically rather unstable, as can be understood from the analytical approaches that we follow here.

The approximations employed for the derivation of the scaling solution near $\tilde{\rho} = 0$ break down if $\tilde{\rho}$ comes close to the value $\tilde{\rho}_s = 1/(64\pi^2)$. The region of $\tilde{\rho}$ around $\tilde{\rho}_s$ describes the transition from the UV-region near $\tilde{\rho} = 0$, where the details of the scalar fluctuations may matter, to the region of larger $\tilde{\rho}$ where $\tilde{m}^2$ and $\lambda$ can be neglected. For small $\Delta u$, $\tilde{m}^2 = \partial(\Delta u)/\partial\tilde{\rho}$ and $\lambda = \partial^2(\Delta u)/\partial\tilde{\rho}^2$, we can use Equation (A19) as a second order differential equation:

$$\left(\tilde{\rho} - \frac{1}{64\pi^2}\right)\frac{\partial\Delta u}{\partial\tilde{\rho}} = \frac{4 - A}{2}\Delta u + \frac{\tilde{\rho}}{32\pi^2}\frac{\partial^2\Delta u}{\partial\tilde{\rho}^2} \,. \tag{A51}$$

In terms of the variable:

$$s = 64\pi^2\tilde{\rho} \tag{A52}$$

this reads:

$$(s - 1)\frac{\partial\Delta u}{\partial s} + \frac{A - 4}{2}\Delta u - 2s\frac{\partial^2\Delta u}{\partial s^2} = 0 \,. \tag{A53}$$

Except for the constant scaling solution $\Delta u = 0$, the function $\Delta u$ grows outside the range of this linear approximation as $s$ increases. We also investigated numerically the nonlinear second order differential equation:

$$\tilde{\rho}\,\frac{\partial u}{\partial\tilde{\rho}} = 2(u - c_{\mathrm{U}} - c_{\mathrm{U},s}) \,, \tag{A54}$$

with $\tilde{m}^2 = \partial u/\partial\tilde{\rho}$, $\lambda = \partial^2 u/\partial\tilde{\rho}^2$ in Equation (A13) which yields Equation (A15). The only scaling solutions extending for the whole range $0 \le \tilde{\rho} < \infty$ that we have found so far are the constant scaling solutions. The failure to find other solutions may, however, be due to numerical instabilities which could prevent the detection of global crossover solutions.

The issues of finding scaling solutions for scalars coupled to gravity can be understood by neglecting the term $2\lambda\rho$ in Equations (26) and (A13). In this approximation the scaling solution has to obey the differential equation:

$$2\tilde{\rho}\,\frac{\partial u}{\partial\tilde{\rho}} = 4(u - c_{\mathrm{U}}) + \frac{N_{\mathrm{S}}}{32\pi^2}\left(1 - \frac{1}{1 + \frac{\partial u}{\partial\tilde{\rho}}}\right) \,. \tag{A55}$$

Here we consider $N_{\mathrm{S}}$ scalars with mass term $m^2 = \partial u/\partial\tilde{\rho}$, while $N - N_{\mathrm{S}}$ accounts for additional massless particles as fermions, gauge bosons or further scalars. Equation (A55) can be written as a quadratic equation in $\tilde{\rho}\,\partial u/\partial\tilde{\rho}$ and therefore can be transformed to:

$$\tilde{\rho}\,\frac{\partial u}{\partial\tilde{\rho}} = \frac{1}{2}\left\{2u - 2c_{\mathrm{U}} + \frac{N_{\mathrm{S}}}{64\pi^2} - \tilde{\rho} \pm \sqrt{W}\right\} \,, \tag{A56}$$

with

$$W = \left(2u - 2c_U + \frac{N_S}{64\pi^2} + \tilde{\rho}\right)^2 - \frac{N_S\tilde{\rho}}{16\pi^2}.$$ 
(A57)

Let us first consider the limit $\tilde{\rho} \to \infty$ where:

$$\sqrt{W} = \tilde{\rho} + 2u - 2c_U - \frac{N_S}{64\pi^2} + \frac{N_S(u - c_U)}{4\pi^2\tilde{\rho}} + \dots$$ 
(A58)

Employing the solution with the plus sign in Equation (A56) one finds:

$$\tilde{\rho}\frac{\partial u}{\partial \tilde{\rho}} = (2u - 2c_U)\left(1 + \frac{N_S}{64\pi^2\tilde{\rho}}\right).$$ 
(A59)

One sees again that the scaling solution approaches for $\tilde{\rho} \to \infty$ the constant $u = c_U$ discussed in Section 3.

For the opposite limit for $\tilde{\rho} \to 0$ one has:

$$\sqrt{W} = 2u - 2c_U + \frac{N_S}{64\pi^2} + \frac{u - c_U - \frac{N_S}{128\pi^2}}{u - c_U + \frac{N_S}{128\pi^2}}\tilde{\rho}.$$ 
(A60)

If $\partial u/\partial \tilde{\rho}$ remains finite for $\tilde{\rho} \to 0$ the relative minus sign in Equation (A56) is appropriate, yielding:

$$\frac{\partial u}{\partial \tilde{\rho}} = -\frac{u - c_U}{u - c_U + \frac{N_S}{128\pi^2}}.$$ 
(A61)

The particular asymptotic behavior $u = c_U$ for $\tilde{\rho} = 0$ implies a vanishing mass term $\tilde{m}^2 = \partial u/\partial \tilde{\rho} \to 0$. With $u_0$ given by the solution $u_0 = c_U(u_0)$ as in Section 3, and $u = u_0 + \Delta u$, the linear expansion of Equation (A56) in $\Delta u$ becomes:

$$\frac{\partial \Delta u}{\partial \tilde{\rho}} = -\frac{(4 - A)\Delta u}{(4 - A)\Delta u + \frac{N_S}{32\pi^2}} \approx -\frac{32\pi^2(4 - A)}{N_S}\Delta u,$$ 
(A62)

in accordance with Equation (A19) for $\tilde{\rho} \to 0$ for $N_S = 1$. The sign of the mass term at the origin is opposite to the sign of $\Delta u$.

For the differential Equation (A56), the relative plus sign for the square root applies for $\tilde{\rho} \to \infty$, while for $\tilde{\rho} \to 0$ one needs the relative minus sign. There has to be a matching, which must occur for $W = 0$ for reasons of continuity. The value of $\tilde{\rho}_t$ where $W(\tilde{\rho}_t) = 0$ obeys:

$$\tilde{\rho}_t = 2\left(\sqrt{c_U - u} \pm \sqrt{\frac{N_S}{128\pi^2}}\right)^2.$$ 
(A63)

A switch of sign of the $\sqrt{W}$-term is only possible in a region where $u \leq c_U$. Continuity of the quartic coupling $\lambda$ at $\tilde{\rho}_t$ requires further:

$$\frac{\partial W}{\partial \tilde{\rho}}(\tilde{\rho}_t) = 0.$$ 
(A64)

Combining Equation (A64) with $W(\tilde{\rho}_t) = 0$ yields the condition"

$$\tilde{\rho}_t = \frac{N_S}{64\pi^2}.$$ 
(A65)

Comparing further Equations (A65) and (A63) requires at $\tilde{\rho}_t$:

$$c_U - u = \left\{0, \frac{N_S}{32\pi^2}\right\},$$ 
(A66)

corresponding to:

$$\tilde{m}^2(\tilde{\rho}_t) = \{0, -2\}. \tag{A67}$$

The second value violates the condition $\tilde{m}^2 > -1$ needed for a stable scalar propagator, and we conclude that scaling solutions with a change of the sign of the term $\pm\sqrt{W}$ in Equation (A56) require:

$$\tilde{m}^2\left(\tilde{\rho} = \frac{N_S}{64\pi^2}\right) = 0. \tag{A68}$$

Generic local scaling solutions with arbitrary $\tilde{m}_0^2$ do not obey Equation (A68), as we have checked by a numerical solution of Equation (A56).

The two flat scaling solutions discussed in Section 3,

$$u(\tilde{\rho}) = c_U(v_\pm), \tag{A69}$$

with $v_\pm$ given by Equation (30), obey Equation (A68). For these particular solutions there is actually a change of sign in the term $\pm\sqrt{W}$ in Equation (A56) at $\tilde{\rho}_t = N_S/(64\pi^2)$, since $\sqrt{W} = \tilde{\rho} - \tilde{\rho}_t$ for $\tilde{\rho} > \tilde{\rho}_t$ and $\sqrt{W} = \tilde{\rho}_t - \tilde{\rho}$ for $\tilde{\rho} < \tilde{\rho}_t$. It is not obvious if there exist other scaling solutions obeying the condition (A68). If not, and if the inclusion of nonzero $\lambda$ does not change in an important way the possible scaling solutions, the inclusion of the scalar mass term reduces the family of scaling solution for matter freedom discussed in Section 3 to only two scaling solutions, both with a flat potential.

If we do not impose the condition (A64), the matching of solutions with different signs $\pm\sqrt{W}$ at $\tilde{\rho}_t$ typically induces a discontinuity of $\partial\tilde{m}^2/\partial\tilde{\rho}$ at $\tilde{\rho}_t$. Higher order couplings will then diverge and the approximation of neglecting them remains no longer valid. So far, it is not known if the inclusion of $\lambda$ (and higher order couplings as $u^{(3)}$) can smoothen the discontinuity or not. At the present stage it is therefore not known if a continuous family of scaling solutions exists, or if this is reduced to a discrete subset.

As a general lesson, we conclude that the issue of the existence of a whole family of scaling solutions, or only a discrete subset, is typically decided in the transition region. Both limiting cases $\tilde{\rho} \to 0$ and $\tilde{\rho} \to \infty$ admit families of scaling solutions characterized by a continuous parameter. The question is if the families of scaling solutions in the two limiting cases can be matched to each other continuously in the transition region. For pure scalar models coupled to gravity with constant $w$, it seems most likely that only a discrete subset of overall scaling solutions remains. We will see that non-vanishing gauge or Yukawa couplings, as well as non-trivial couplings to gravity encoded in the $\tilde{\rho}$-dependence of $w(\tilde{\rho})$, modify the properties of the transition region considerably.

**Appendix D. Yukawa Couplings**

In this appendix we discuss the influence of a non-zero Yukawa coupling of the scalar to fermions. In this case the fermion fluctuations stabilize a minimum of the potential at $\rho = 0$, preventing spontaneous symmetry breaking in the scaling solution.

*Appendix D.1. Flow Equations with Yukawa Coupling*

A fermion with a Yukawa coupling $y$ to a scalar field $\phi$ acquires a mass $m = y\phi$. Similar to massive gauge bosons, the mass suppresses its contribution to the flow of $u$. This induces a modification of the flow generator:

$$\frac{\Delta\tilde{\pi}_f}{k^4} = -\frac{1}{16\pi^2}\sum_{j=1}^{N_F}\left(\frac{1}{1+w_j} - 1\right). \tag{A70}$$

Here $j$ labels the mass eigenstates with mass $m_j$, and

$$w_j = \frac{m_j^2}{k^2} = y^2 a_j^{(F)}(\phi_a)/k^2 \tag{A71}$$

is the dimensionless squared mass of the fermion $j$. The sum is over Majorana fermions, with Dirac fermions counting as two Majorana fermions with equal $m_j^2$. The minus sign reflects Fermi statistics and is the main difference as compared to the gauge boson contribution. In case of several independent Yukawa couplings, the mass eigenvalues $m_j$ depend on a linear combination of Yukawa couplings and fields.

We may again investigate a simple scenario where $\bar{N}_F$ fermions have an equal mass,

$$m_j^2 = c_f \, y^2 \rho \, . \tag{A72}$$

Similar to Equation (57) this results in:

$$\frac{\Delta \tilde{\pi}_f}{k^4} = \frac{c_f \bar{N}_F \, y^2 \tilde{\rho}}{16\pi^2 \left(1 + c_f \, y^2 \tilde{\rho}\right)} \, . \tag{A73}$$

We can therefore take over the computations for gauge couplings with the replacements:

$$c_g \, g^2 \to c_f \, y^2 \, , \quad \bar{N}_V \to -\frac{2}{3} \bar{N}_F \, . \tag{A74}$$

*Appendix D.2. Flow Away from the Scaling Solution*

The essential new feature is the overall change of sign of the fermion contribution as compared to the gauge boson contribution. Combining with the gravitational contribution and contributions from other massless fields one obtains:

$$\partial_t u = -4u + 2\tilde{\rho} \frac{\partial u}{\partial \tilde{\rho}} + 4c_\text{U} + 4c_{\text{U},f} \, , \tag{A75}$$

with

$$c_{\text{U},f} = \frac{\Delta \tilde{\pi}_f}{4k^4} = -\frac{\bar{N}_F}{64\pi^2} \left(\frac{1}{1 + y^2 \tilde{\rho}} - 1\right) = \frac{\bar{N}_F \, y^2 \tilde{\rho}}{64\pi^2 (1 + y^2 \tilde{\rho})} \, . \tag{A76}$$

Here we take $c_f = 1$. We observe that $c_{\text{U},f}$ adds a positive contribution to $c_\text{U}$.

The flow of the mass term $\tilde{m}^2 = \partial u / \partial \tilde{\rho}$ is found by taking a $\tilde{\rho}$-derivative of Equation (A75),

$$\partial_t \tilde{m}^2 = -2\tilde{m}^2 + 2\tilde{\rho} \frac{\partial \tilde{m}^2}{\partial \tilde{\rho}} + A\tilde{m}^2 + \frac{\bar{N}_F y^2}{16\pi^2 (1 + y^2 \tilde{\rho})^2} \, , \tag{A77}$$

where we have omitted contributions from $\Delta \tilde{\pi}_s$ and possible contributions from $\Delta \tilde{\pi}_\text{gauge}$. If $\lambda(\tilde{\rho}) = \partial \tilde{m}^2 / \partial \tilde{\rho}$ remains finite for $\tilde{\rho} \to 0$ the flow of the scalar mass term at the origin reads:

$$\partial_t \tilde{m}_0^2 = (A_0 - 2)\tilde{m}_0^2 + \frac{\bar{N}_F y^2}{16\pi^2} \, . \tag{A78}$$

For $A_0 < 2$ the fixed point occurs for positive $\tilde{m}_{0,*}^2$, such that the origin is a local minimum for the scaling solution:

$$\tilde{m}_{0,*}^2 = \frac{\bar{N}_F y^2}{16\pi^2 (2 - A_0)} \, . \tag{A79}$$

In contrast, for $A_0 > 2$ one has $\tilde{m}_{0,*}^2 < 0$ and a local maximum at the origin for the scaling solution (this holds provided the term $\sim \lambda_0$ from $\Delta \tilde{\pi}_s$ is small).

The flow of the quartic coupling $\lambda(\tilde{\rho})$ is obtained by taking a further $\tilde{\rho}$-derivative of Equation (A77):

$$\partial_t \lambda = A\lambda + 2\tilde{\rho} \frac{\partial \lambda}{\partial \tilde{\rho}} - \frac{\bar{N}_F y^4}{8\pi^2 (1 + y^2 \tilde{\rho})^3} + B\tilde{m}^4 \, . \tag{A80}$$

For $\tilde{\rho} \to 0$ one recognizes the well known negative contribution from the Yukawa coupling $\sim y^4$. The fixed point for $\lambda_0 = \lambda(\tilde{\rho} = 0)$ occurs for:

$$\begin{aligned}
\lambda_{0,*} &= \frac{\bar{N}_F\, y^4}{8\pi^2 A_0} - \frac{B_0}{A_0}\tilde{m}_{0,*}^4 \\
&= \frac{\bar{N}_F\, y^4}{8\pi^2 A_0}\left(1 - \frac{\bar{N}_F B_0}{32\pi^2(2 - A_0)^2}\right).
\end{aligned} \tag{A81}$$

The relative size of the second contribution $\sim B_0$ is typically small, such that $\lambda_{0,*} > 0$. For the flow away from the fixed point we can keep $\lambda$ close to the fixed point value:

$$\lambda_{0,*} \approx \frac{\bar{N}_F y^4}{8\pi^2 A_0}, \tag{A82}$$

since it is an irrelevant parameter. For $A_0 < 2$ the mass term is relevant, however, and the deviation from the critical surface $\gamma = \tilde{m}_0^2 - \tilde{m}_{0,*}^2$ increases according to Equation (A36). Indeed, the last term in Equation (A78) shifts the value of $\tilde{m}_{0,*}^2$, but does not contribute to the flow of $\gamma$. The same holds for the flow of $\delta u$. The only difference to the discussion in Appendix C.3 are the different fixed point values for $\tilde{m}_{0,*}^2$ and $\lambda_{0,*}$. Omitting the irrelevant coupling $\delta\lambda$ the stability matrix for $\alpha_i = (\delta u, \gamma)$ reads, see Equation (A46),

$$-T = \begin{pmatrix} A - 4 & -d \\ B\tilde{m}_{0,*}^2 & A - 2 + 3d\lambda_{0,*} \end{pmatrix}. \tag{A83}$$

Neglecting the small off-diagonal elements we can follow the discussion of Equations (A38)–(A40), resulting in:

$$\tilde{m}^2(k) = \frac{\bar{N}_F\, y^2 k^2}{16\pi^2(2 - A_0)} + \gamma_\Lambda \Lambda^{\theta_\gamma} k^{2 - \theta_\gamma}, \tag{A84}$$

with

$$\theta_\gamma = 2 - A_0 - \frac{3\bar{N}_F\, y^4}{(16\pi^2)^2 A_0\left(1 + \tilde{m}_{0,*}^2\right)^2}. \tag{A85}$$

The last term in Equation (A85) is small and may be neglected, such that:

$$m^2(k) = \frac{\bar{N}_F\, y^2 k^2}{16\pi^2(2 - A_0)} + \gamma_\Lambda \Lambda^{2 - A_0} k^{A_0}. \tag{A86}$$

The scaling solution with constant $A_0$ holds only as long as the gravitational fluctuations are effective. The running coupling $w_0(k)$ corresponds to a relevant parameter, with qualitative behavior:

$$w_0(k) = w_{0,*} + \frac{M^2}{2k^2}, \tag{A87}$$

where $M$ is the observed reduced Planck mass. Once $k^2$ drops below $k_c^2 = M^2/(2w_{0,*})$, the function $w_0(k)$ starts to increase rapidly. As a consequence, $A_0(k)$ decreases rapidly to zero for $k \ll k_c$. For $k^2 \ll k_c^2$ one has approximately:

$$m_0^2(k) = m_0^2(k_c) - \frac{\bar{N}_F\, y^2\left(k_c^2 - k^2\right)}{32\pi^2}. \tag{A88}$$

For a suitable value of $\gamma_\Lambda$, one can obtain $m_0(k_c) = \bar{N}_F\, y^2\, k_c^2/(32\pi^2)$, such that $m_0^2(k = 0) = 0$. More generally, by suitable initial conditions for the relevant parameter $\gamma$ one

can achieve arbitrary values of $m_0^2(k=0)$. In turn, one can realize an arbitrary value of spontaneous symmetry breaking, e.g. an arbitrary location of the minimum of $U$ at $k=0$,

$$\rho_0(k) = -\frac{m_0^2(k)}{\lambda_0(k)} . \tag{A89}$$

If $\rho_0(0) \neq 0$ is associated with a spontaneous breaking of a symmetry, the scale of this symmetry breaking is a free parameter. What is predicted, however, is the value of the quartic coupling, since it is associated to an irrelevant parameter. With "initial values":

$$\lambda_0(k_c) = \lambda_{0,*}, \quad y(k_c) = y, \tag{A90}$$

one can follow for $k < k_c$ the "low energy flow" of $\lambda$ and $y$ to $k = 0$. The gravitational degrees of freedom do not contribute to the low energy flow.

*Appendix D.3. Global Scaling Solution*

So far we only have discussed the vicinity of $\tilde{\rho} = 0$. As long as the minimum $\rho_0(k)/k^2$ stays small, this is a reasonable local approximation. In order to be sure that no other minimum of $u$ occurs for large $\tilde{\rho}$ one needs the global solution for the whole range of $\tilde{\rho}$. We have numerically solved the differential equation for the scaling solution:

$$2\tilde{\rho}\,\frac{\partial u}{\partial \tilde{\rho}} = 4\Big(u - c_U - c_{U,f}\Big). \tag{A91}$$

The result is shown in Figure A1 for different values of $w_0$. The only minimum occurs for $\tilde{\rho} = 0$. The mass term $\tilde{m}^2(\tilde{\rho})$ is positive and small for the whole range of $\tilde{\rho}$, as can be seen from Figure A2. This justifies the omission of $c_{U,s}$ for the whole range of $\tilde{\rho}$, in contrast to the situation with vanishing Yukawa coupling. The qualitative situation does not depend sensitively on the initial conditions, as demonstrated in Figure A3. It seems that the differential Equation (A91) admits a whole family of scaling solutions extended over the whole range of $\tilde{\rho}$. In view of the small value of $\tilde{m}^2$ we do not expect that this changes if $c_{U,s}$ is included.

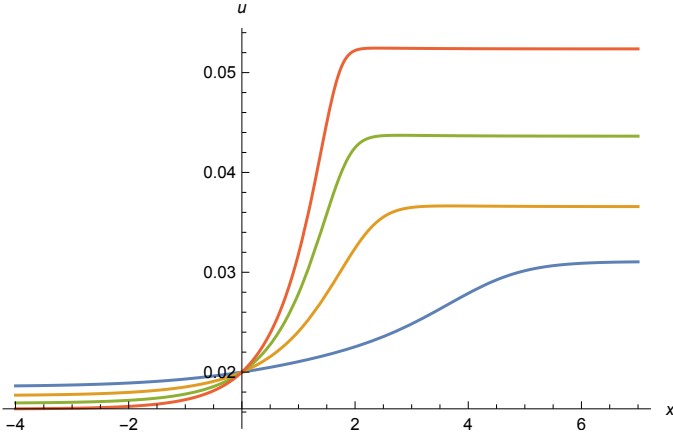

**Figure A1.** Effective potential $u(x)$ as function of $x = \ln(\tilde{\rho})$ in presence of a Yukawa coupling, $y^2/4\pi = 1/40$. The curves (from bottom to top on the right part) for $w_0 = 0.042, 0.046, 0.052$, and $0.06$. Other parameters are $N = 10$, $\bar{N}_f = 1$, and initial values are set as $u(x = 0) = 0.02$.

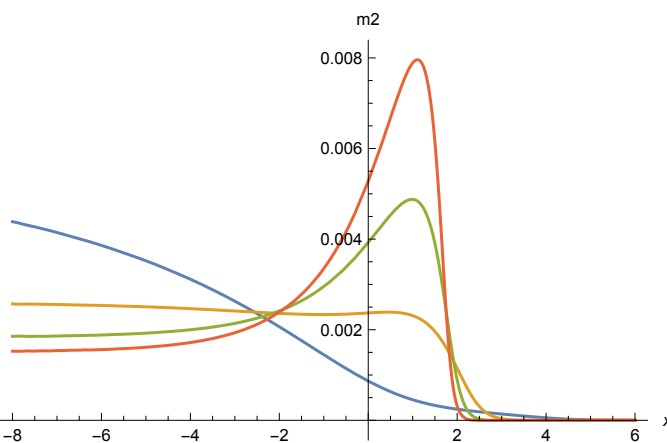

**Figure A2.** Mass term $\tilde{m}^2$ as function of $x = \ln(\tilde{\rho})$ for nonvanishing Yukawa couplings. Parameters are as for Figure 9, with largest value at $x = 0$ for $w_0 = 0.06$ and smallest for $w_0 = 0.042$.

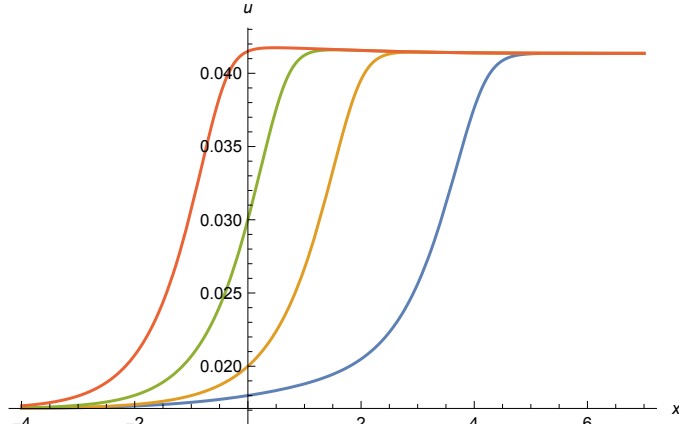

**Figure A3.** Effective potential $u(x)$ as function of $x = \ln(\tilde{\rho})$ for a Yukawa coupling $y^2/4\pi = 1/40$. Parameters are $N = 10$, $\bar{N}_f = 1$, $w_0 = 0.05$ and we show different initial conditions at $x = 0$, namely $u(x = 0) = 0.018, 0.02, 0.03$, and $0.0415$ from right to left.

### Appendix E. Vicinity of the Constant Scaling Solution

In this appendix we investigate possible scaling solutions that remain in the vicinity of the constant scaling for a certain range in $\tilde{\rho}$. We work in the truncation of two free functions $u(\tilde{\rho})$ and $w(\tilde{\rho})$.

In the vicinity of the constant scaling solution we may linearize the differential Equation (98) for small:

$$\Delta u(\tilde{\rho}) = u(\tilde{\rho}) - u_* , \quad \Delta w(\tilde{\rho}) = w(\tilde{\rho}) - w_* , \tag{A92}$$

with $u_*$, $w_*$ the constant scaling solutions according to Equation (106). The linearized equations read:

$$2\tilde{\rho} \, \partial_{\tilde{\rho}} \Delta u = 4\Delta u - 4\Delta c_{\mathrm{U}} , \quad 2\tilde{\rho} \, \partial_{\tilde{\rho}} \Delta w = 2\Delta w - 2\Delta c_M , \tag{A93}$$

with:

$$\Delta c_{\mathrm{U}} = c_{\mathrm{U}}(u_* + \Delta u, \, w_* + \Delta w) - c_{\mathrm{U}}(u_*, w_*) ,$$
$$\Delta c_M = c_M(u + \Delta u, \, w_* + \Delta w) - c_M(u_*, w_*) . \tag{A94}$$

For $g^2 = y^2 = 0$ one finds:

$$\Delta c_{\rm U} = \frac{5}{96\pi^2}\tilde{\Delta} - \frac{N_{\rm S}}{128\pi^2}\Delta u',$$

$$\Delta c_M = \frac{25}{128\pi^2}\tilde{\Delta} + \frac{N_{\rm S}}{192\pi^2}\Delta u' - \frac{N_{\rm S}}{64\pi^2}\Delta w', \tag{A95}$$

where

$$\tilde{\Delta} = \frac{1}{1-v} - \frac{1}{1-v_*} = \frac{1}{(1-v_*)^2}\frac{w_*\Delta u - u_*\Delta w}{w_*^2}. \tag{A96}$$

Equation (A95) is a coupled system of two linear differential equations,

$$\left(2\tilde{\rho} - \frac{N_{\rm S}}{32\pi^2}\right)\Delta u' = 4\Delta u - \frac{5}{24\pi^2}\tilde{\Delta},$$

$$\left(2\tilde{\rho} - \frac{N_{\rm S}}{32\pi^2}\right)\Delta w' + \frac{N_{\rm S}}{96\pi^2}\Delta u' = 2\Delta w - \frac{25}{64\pi^2}\tilde{\Delta}. \tag{A97}$$

We can write $\tilde{\Delta}$ in terms of the graviton-induced anomalous dimension $A$ for which the second term in Equation (37) is neglected and $v$, $w$ taken as $v_*$, $w_*$,

$$A = \frac{5}{24\pi^2 w_*\,(1-v_*)^2}, \tag{A98}$$

namely,

$$\frac{5}{24\pi^2}\tilde{\Delta} = A\,(\Delta u - v_*\Delta w). \tag{A99}$$

This yields:

$$\left(2\tilde{\rho} - \frac{N_{\rm S}}{32\pi^2}\right)\Delta u' = (4 - A)\,\Delta u + A\,v_*\Delta w,$$

$$\left(2\tilde{\rho} - \frac{N_{\rm S}}{32\pi^2}\right)\Delta w' + \frac{N_{\rm S}}{96\pi^2}\Delta u' = 2\Delta w + \frac{15A}{8}\,(\Delta u - v_*\Delta w), \tag{A100}$$

to be compared with Equation (A19) for $\lambda = 0$.

We may discuss separately three characteristic regions in $\tilde{\rho}$. For $\tilde{\rho} \ll N_{\rm S}/(64\pi^2)$ one has the approximate equations:

$$\Delta y' = -M_0\Delta y, \quad \Delta y = \begin{pmatrix}\Delta u \\ \Delta w\end{pmatrix}, \tag{A101}$$

with

$$M_0 = \frac{32\pi^2}{N_{\rm S}}\begin{pmatrix} 4 - A & A\,v_* \\ \frac{4}{3} + \frac{37}{24}A & 2 - \frac{37}{24}A\,v_* \end{pmatrix}. \tag{A102}$$

The two eigenvalues $\lambda_\pm$ of $M_0$ are the solutions of the quadratic equation:

$$\lambda^2 - \lambda\left(6 - A\left(1 + \frac{37}{24}v_*\right)\right) + 8 - A\left(2 + \frac{15}{2}v_*\right) = 0. \tag{A103}$$

For $A = 0$, one finds two possible eigenvalues $\lambda_+ = 4$, $\lambda_- = 2$. With respect to the flow with increasing $\tilde{\rho}$ both $\Delta u$ and $\Delta w$ are relevant parameters. This extends to a range $A > 0$ with eigenvalues depending continuously on A. The solution approaches for $\tilde{\rho} \to 0$ constant values, $\Delta u(\tilde{\rho} = 0) = \Delta u_0$, $\Delta w(\tilde{\rho} = 0) = \Delta w_0$. They are related to the derivatives by Equations (101) and (102). This behavior is similar to the discussion in Appendix C.2.

A second region for $\tilde{\rho} \gg N_{\rm S}/(64\pi^2)$ obeys the approximate equations:

$$\tilde{\rho}\,\partial_{\tilde{\rho}}\,\Delta y = M_\infty\Delta y \tag{A104}$$

with

$$M_\infty = \frac{1}{2}\begin{pmatrix} 4-A & A\,v_* \\ 15A/8 & 2-\frac{15\,A\,v_*}{8} \end{pmatrix}. \tag{A105}$$

If the largest eigenvalue $\lambda_+$ of $M_\infty$ is positive, the solution diverges for $\tilde{\rho} \to \infty$ as:

$$\Delta y_* = c_+ \tilde{\rho}^{\lambda_+} + c_- \tilde{\rho}^{\lambda_-} \tag{A106}$$

with $c_\pm$ eigenvectors of $\lambda_\pm$ specifying the particular solutions. We conclude that the constant scaling solution is "unstable" in the sense that generic local solutions do not approach the scaling solutions for $\tilde{\rho} \to \infty$. If a whole family of scaling solutions exists, the constant scaling solution does not correspond to the generic solution of this family. In the other direction, many possible scaling solutions may be attracted towards the vicinity of the constant scaling solution for $\tilde{\rho} \to 0$. This holds, in particular, if both $\lambda_+$ and $\lambda_-$ are positive.

The third region is the "transition region" for $\tilde{\rho} \approx N_S/(64\pi^2)$. One expects that the neglection of second derivatives $u''$ and $w''$ may be no longer justified. The situation is similar to the discussion in Appendix C.5. If we continue to neglect $u''$ and $w''$, the scaling solution can cross the transition region only at the price of strong variations of $u'$ and $w'$. In the close vicinity of $\tilde{\rho} = N_S/(64\pi^2)$ Equation (A100) is approximated by:

$$\Delta w = -\frac{4-A}{A\,v_*}\Delta u\,, \tag{A107}$$

and

$$\Delta u' = \frac{48\pi^2}{N_S}\left(15 - \frac{4\,(4-A)}{A\,v_*}\right)\Delta u\,. \tag{A108}$$

We could specify the initial conditions for the solution of the linear differential equations by specifying $\Delta u$ and $\Delta w$ at $\tilde{\rho} = N_S/(64\pi^2)$. The condition (A107) implies that only a one-parameter family of scaling solutions in the close vicinity of the constant scaling solutions is possible.

For any smooth scaling solution we can write:

$$\Delta u = a\,\Delta w\,, \quad \Delta u' = a\,\Delta w' + a'\,\Delta w\,, \tag{A109}$$

with $a(\rho)$ a function without very rapid variation with $\tilde{\rho}$. The second Equation (A100) becomes:

$$\left[2\tilde{\rho} - \frac{N_S}{32\pi^2}\left(1 - \frac{a}{3}\right)\right]\Delta w' = B\,\Delta w\,, \tag{A110}$$

with

$$B = 2 + \frac{15\,A}{8}(a - v_*) - \frac{N_S}{96\pi^2}f\,a\,, \quad f = \frac{a'}{a}\,. \tag{A111}$$

The coefficient of $\Delta w'$ vanishes for:

$$\tilde{\rho}_w = \frac{N_S}{64\pi^2}\left(1 - \frac{a}{3}\right), \tag{A112}$$

which is for $a < 0$ somewhat smaller than the location of the vanishing coefficient of $\Delta u'$ in the first Equation (A100) at $\tilde{\rho}_u = N_S/(64\pi^2)$. This is compatible with a smooth behavior only if $B(\tilde{\rho}_w) = 0$. For a smooth solution the term $\sim f$ is typically small as compared to other terms in $B$. One infers different values of $a$ at $\tilde{\rho}_w$ and $\tilde{\rho}_u$,

$$a(\tilde{\rho}_w) = v_* - \frac{16}{15A}\,, \quad a(\tilde{\rho}_u) = -\frac{A\,v_*}{4-A}\,. \tag{A113}$$

In turn, the first Equation (A100) yields at $\tilde{\rho}_w$:

$$\Delta u'(\tilde{\rho}_w) = -\frac{96\pi^2}{N_S}\left(4 - A + \frac{A\,v_*}{a(\tilde{\rho}_w)}\right)\Delta w$$

$$\approx -\frac{384\pi^2}{N_S}\left(1 + \frac{4A}{15A\,v_* - 16}\right)\Delta w\,. \tag{A114}$$

The value $\Delta u'/\Delta w$ becomes typically very large at $\tilde{\rho}_w$, contradicting the assumption of slow variation of the scaling solution.

Numerical solutions in the transition region are indeed characterized by strong variations of $\Delta u'$ and $\Delta w'$ in the transition region. Those solutions typically diverge outside the transition region before $\tilde{\rho} = 0$ and $\tilde{\rho} \to \infty$ are reached, such that the linear approximation does not remain valid. Similarly to Appendix C.5 we remain with two possibilities. Either the inclusion of $\Delta u''$ and $\Delta w''$ changes the strong variations such that a family of slowly varying scaling solutions exists in the vicinity of the constant scaling solution. Or only a discrete number of scaling solutions remains close to the scaling solution over the whole range of $\tilde{\rho}$. It could turn out that the constant scaling solution is the only possible scaling solution for $g^2 = y^2 = 0$.

A possible scenario for a continuous family of solutions could be that all solutions that are close to the constant scaling solution for $\tilde{\rho} \to 0$ always deviate substantially from the scaling solution for large enough $\tilde{\rho}$, such that the linear approximation no longer holds. This happens, in particular, if the asymptotic behavior of $w(\tilde{\rho} \to \infty)$ is not a constant, but rather involves the non-minimal coupling $\tilde{\xi}$. This type of scaling solution is discussed in Section 7.5.

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
