# Peer review of "Effective Scalar Potential in Asymptotically Safe Quantum Gravity"

_universe, doi:10.3390/universe7020045_

Round 1

Reviewer 1 Report

The paper is concerned with the computation of effective scalar potentials in presence of quantum gravitational effects. As quantum gravitational input the author adopts the notion of ultraviolet completion known as asymptotically safe quantum gravity, in which it is conjectured that metric gravity admits an ultraviolet fixed point of the renormalization group. The most natural physical application of such potential is of course the Higgs sector of the standard model, and the computations are expected to be relevant in the context of Higgs inflation, which is in fact the topic that the author addresses thereafter.

The paper is very interesting and shows a well structured discussion on the compatibility, or lack thereof, of Higgs inflation with asymptotic safety. The main discussion is based on a few assumptions, for example one is the validity of the method of Shaposhnikov and the author to determine the Higgs mass. Personally I have nothing against the idea, which is actually very interesting, but it is important to remember that obtaining the number 126 might be a circumstantial lucky accident, and therefore the entire perspective on the topic could be subjective.

In these respects, some sentences of the paper are a bit "strong" in that they validate the view of the author. An early example could be the sentence after (1) in which it is said that the anomalous dimension induced by gravity to the running of lambda is universal. This might be true in d=2, but I find it difficult to believe in d=4. Nevertheless, I would be more afraid of the potential dangers of gauge dependence, rather than cutoff dependence.

Having said that, I think that the paper excels in explaining and delivering the author's point of view on the topic, which, as I already said, is interesting and, if further validated, potentially groundbreaking. For this reason I recommend the paper for publication.

Author Response

Thanks for the positive report!

The paper serves to argue that the prediction is not an accident.

It depends, of course, on the absence of much new physics below the Planck scale.

The wording "universal" was not meant in the sense of cutoff independence, but rather in the sense explained in the following sentence. I will choose as a clearer formulation:

is universal in the sense that...

Reviewer 2 Report

The paper is devoted to the analysis of effective scalar potentials in asymptotically safe quantum gravity, scaling solutions and spontaneous symmetry breakings. It presents a wide and deep analysis on the matter. Hence, I think the paper should be accepted for publication in Universe in its present form.

Author Response

Thanks for the positive report!

Reviewer 3 Report

This is an extensive review of what we know about the effect of gravitational fluctuations on the form of the scalar potential. Such effects are generally neglected in particle physics applications, but they are expected to be present at high energy and could play an important role in the early universe. The paper is based mainly on previous work by the same author, but due reference is made to work by others. It also contains some new material.

Author Response

Thanks for the positive report!